# Systematic epigenome editing captures the context-dependent instructive function of chromatin modifications

Cristina Policarpi ●[1], Marzia Munafò ●[1], Stylianos Tsagkris ●[1], Valentina Carlini[1,2] & Jamie A. Hackett ●[1,3] ✉

Chromatin modifications are linked with regulating patterns of gene expression, but their causal role and context-dependent impact on transcription remains unresolved. Here we develop a modular epigenome editing platform that programs nine key chromatin modifications, or combinations thereof, to precise loci in living cells. We couple this with single-cell readouts to systematically quantitate the magnitude and heterogeneity of transcriptional responses elicited by each specific chromatin modification. Among these, we show that installing histone H3 lysine 4 trimethylation (H3K4me3) at promoters can causally instruct transcription by hierarchically remodeling the chromatin landscape. We further dissect how DNA sequence motifs influence the transcriptional impact of chromatin marks, identifying switch-like and attenuative effects within distinct *cis* contexts. Finally, we examine the interplay of combinatorial modifications, revealing that co-targeted H3K27 trimethylation (H3K27me3) and H2AK119 monoubiquitination (H2AK119ub) maximizes silencing penetrance across single cells. Our precision-perturbation strategy unveils the causal principles of how chromatin modification(s) influence transcription and dissects how quantitative responses are calibrated by contextual interactions.

Regulation of eukaryotic transcription is guided by a complex interplay between transcription factors (TFs), *cis* regulatory elements and epigenetic mechanisms. The latter includes chromatin-based systems, most prominently post-translational histone and DNA modifications. Such 'chromatin modifications' influence transcription activity by directly altering chromatin compaction, by acting as specific docking sites for 'reader' proteins and/or by influencing TF access to cognate motifs[1–3]. As a result, chromatin marks are thought to play a central regulatory role in deploying and propagating gene expression programs during development, while, conversely, aberrant chromatin profiles are linked with gene mis-expression and pathology[4–6].

Major initiatives have mapped genome-wide chromatin modifications across healthy and disease cell types, revealing correlations with genomic features and transcription activity[7–12]. For example, H3K4me3 is enriched at active gene promoters, and H3K9 dimethylation (H3K9me2), H3K9me3, H3K27me3 and H2AK119ub are correlated with transcription repression, while active enhancers are comarked by H3K4 monomethylation (H3K4me1) and H3K27 acetylation (H3K27ac)[13]. Whether the observed correlations indicate causation remains unresolved however[14–17]. To interrogate the nature of functional relationships, perturbation strategies have been widely deployed, often by manipulating chromatin-modifying enzymes or histone residues[5,18–20]. While insightful, such approaches affect the entire (epi)genome

[1]Epigenetics and Neurobiology Unit, European Molecular Biology Laboratory (EMBL), Rome, Italy. [2]Faculty of Biosciences, EMBL and Heidelberg University, Heidelberg, Germany. [3]Genome Biology Unit, EMBL, Heidelberg, Germany. ✉e-mail: jamie.hackett@embl.it

**Fig. 1 | A modular toolkit for precisely programming chromatin states.**
**a**, Schematic of the modular epigenetic editing platform. Upon DOX induction, dCas9$^{GCN4}$ recruits five copies of the CD of chromatin-modifying effector(s) or control GFP$^{scFV}$ to target loci via a specific gRNA. DNAme, DNA methylation.
**b**, Relative abundance of the indicated histone modification at *Hbb-y* assayed by either CUT&RUN–qPCR or by chromatin immunoprecipitation followed by qPCR (ChIP–qPCR) (H3K36me3, H3K79me2), following epigenetic editing or control GFP$^{scFV}$ recruitment in ESCs for 7 d. Shown is the mean of three biologically independent experiments; error bars indicate s.d. Norm., normalized.
**c**, Histogram showing mean DNA methylation installed at the unmethylated *Col16a1* promoter, determined by bisulfite pyrosequencing in three biologically independent experiments; error bars indicate s.d. **d–i**, Relative abundance of the indicated histone modification (H3K4me3 (**d**), H3K27me3 (**e**), H2AK119ub (**f**),

H3K27ac (**g**), H3K9me3 (**h**), H3K36me3 (**i**)) across the *Hbb-y* locus after epigenetic programming with a specific CD$^{scFV}$ (Prdm9 (**d**), Ezh2 (**e**), Ring1b (**f**), p300 (**g**), G9a (**h**), Setd2 (**i**); red line) or control GFP$^{scFV}$ (gray line), assayed by CUT&RUN–qPCR. Mean enrichment across a ~14-kb region centered on the gRNA-binding site is shown for editing in biological triplicates as well as for endogenous positive (Pos1 and Pos2) and negative (Neg1 and Neg2) loci for each mark. NS, not significant. ND, not determined. **j**, Percentage of DNA methylation at CpG dinucleotides across the *Col16a1* and *Hand1* promoters in triplicate experiments. **k**, Scatterplots showing limited OFF-target gene expression changes following induction of the indicated epigenetic mark at *Hbb-y* for 7 d, relative to that of control GFP$^{scFV}$. Differentially expressed genes are indicated in green or orange. Gray dots indicate unaffected genes. *p300*, *ep300*; *G9a*, *Ehmt2*; *Ring1b*, *Rnf2*. *P* values in all panels were calculated by one-tailed unpaired *t*-test. *$P < 0.05$, **$P < 0.01$, ***$P < 0.001$.

simultaneously and thus render it challenging to distinguish direct from indirect effects. Indeed, chromatin-modifying enzymes also have multiple non-histone substrates[21,22] and non-catalytic roles[23,24], which further complicates interpretation of their loss of function. Thus, the extent to which chromatin modifications per se causally instruct gene expression states remains unresolved.

A deeper understanding of the functional role of epigenetic modifications on DNA-templated processes would be facilitated by the development of tools for precision chromatin perturbations. Epigenome editing technologies that enable manipulation of specific chromatin states at target loci have recently emerged, primarily based around programmable dead Cas9 (dCas9)-fusion systems[25,26]. For example, p300 and histone deacetylase 3 (HDAC3) have been fused to dCas9 to reciprocally modulate histone acetylation, while other systems aimed to edit DNA methylation, H3K27me3, H3K4me3 and H3K79me2 (refs. [27–36]). Such pioneering studies revealed proof of principle that altering the epigenome can induce at least some changes in gene expression. However, the transcriptional responses to specific marks are generally modest, if at all, and register at only a restricted set of target genes. This may partly reflect technical limitations of current approaches in depositing physiological levels of chromatin modifications, but also implies that their functional impact varies depending on context-dependent influences. Indeed, there is increasing appreciation that factors such as underlying DNA motifs and variants, and the cell type-specific repertoire of TFs, will all modulate the precise impact of a chromatin modification at a given locus[37,38]. Thus, beyond the principle of causality, it is also important to deconvolve the degree to which each chromatin mark affects transcription levels quantitatively (as opposed to an ON–OFF toggle), how DNA sequence context influences this and the hierarchical relationships involved.

Here, we develop a suite of modular epigenome editing tools to systematically program nine biologically important chromatin modifications to target loci at physiological levels. By coupling this with single-cell readouts, we capture the causal and quantitative impact of specific modification(s) on transcription. We further show that epigenetic marks are linked to each other by hierarchical interplays, act combinatorially, and are functionally influenced by underlying sequence motifs.

## Results

### A toolkit for precision epigenome editing at endogenous loci

We sought to engineer a modular epigenome editing system that can program de novo chromatin modification(s) to target loci at physiological levels. To achieve this, we exploited a catalytically inactive dCas9 fused with an optimized tail array of GCN4 motifs (dCas9[GCN4])[39,40]. This tethers five scFV-tagged epigenetic 'effectors' to genomic targets, thereby amplifying editing activity (Fig. 1a). To program a broad range of chromatin modifications, we built a library of effectors, each comprising the catalytic domain (CD) of a DNA- or histone-modifying enzyme linked with scFV (collectively, CD[scFV]). By isolating the CD, we can exclude confounding effects of tethering entire chromatin-modifying proteins, which can exert non-catalytic regulatory activity. The toolkit includes catalytic cores that deposit H3K4me3 (Prdm9-CD[scFV]), H3K27ac

(p300-CD[scFV]), H3K79me2 (Dot1l-CD[scFV]), H3K9me2 (G9a-CD[scFV]), H3K36me3 (Setd2-CD[scFV]), DNA methylation (Dnmt3a3l-CD[scFV]), H2AK119ub (Ring1b-CD[scFV]) and full-length (FL) enzymes that write H3K27me3 (Ezh2-FL[scFV]) and H4K20me3 (Kmt5c-FL[scFV]) (Fig. 1a). As further controls, we generated catalytic point mutants for each CD[scFV] effector (mut-CD[scFV]) that specifically abrogate their enzymatic activity (Extended Data Fig. 1a). Our strategy therefore enables direct assessment of the functional role of the deposited chromatin mark per se.

We engineered the system to be doxycycline (DOX) inducible for dynamic epigenetic editing and used an enhanced guide RNA (gRNA) scaffold for targeting[41]. Moreover, all CD[scFV] effectors were tagged with superfolder green fluorescent protein (GFP) to monitor protein stability, to track dynamics and to isolate epigenetically edited populations (Extended Data Fig. 1b–d). Finally, up to three nuclear localization sequences were incorporated into effectors, as fewer often precluded nuclear accumulation, for example, for Dot1l-CD[scFV] (Extended Data Fig. 1e).

To test for epigenome editing, we introduced dCas9[GCN4] and each CD[scFV] into mouse embryonic stem cells (ESCs) with the piggy-Bac system and targeted the endogenous Hbb-y locus with a single gRNA. Following DOX induction, each effector directed significant deposition of its chromatin modification relative to recruitment of GFP[scFV], judged by quantitative cleavage under targets and release using nuclease (CUT&RUN–quantitative PCR (qPCR)). This includes de novo establishment of H3K27ac ($P = 0.0003$), H3K4me3 ($P = 0.011$), H3K79me2 ($P = 0.029$), H4K20me3 ($P = 0.001$), H3K27me3 ($P = 0.0006$), H2AK119ub ($P = 0.0002$), H3K36me3 ($P = 0.001$), H3K9me2/3 ($P = 0.0002$) (Fig. 1b) and DNA methylation ($P < 0.0001$) (Fig. 1c).

To determine the quantitative level and genomic spreading of installed chromatin marks, we independently assessed enrichment across the entire Hbb-y locus. We observed a peak around the gRNA-binding site, with programmed domains extending >2 kb on either side. Enrichment of targeted histone modifications ranged from sevenfold to >20-fold over background (Fig. 1d–i) and, importantly, was quantitatively comparable to strong positive peaks in most cases. For example, H3K4me3 installation at Hbb-y was equivalent to that at highly marked Pou5f1 (Oct4) and Nanog promoters (Fig. 1d), while de novo H3K27me3 and H2AK119ub were similar to those at Polycomb targets Zic4 and Wnt10a (Fig. 1e,f). Moreover, de novo H3K36me3, H3K79me2 and H4K20me3 were equivalent to endogenous peaks, while H3K9me2/3 and H3K27ac were deposited at moderately lower levels (Fig. 1g–i and Extended Data Fig. 1f). Finally, up to 60% DNA methylation was installed at previously unmethylated promoters (Fig. 1j).

We did not detect OFF-target chromatin mark deposition at negative (nontargeted) loci with most effectors (Fig. 1d–i and Extended Data Fig. 1f). Indeed, analysis of the highly active Prdm9-CD[scFV] effector revealed robust H3K4me3 installation at ON-target Hbb-y but only six other de novo sites genome wide, implying that our recruitment strategy largely facilitates ON-target chromatin editing (Extended Data Fig. 2a,b). We further tested for indirect and OFF-target effects at the functional level by performing RNA-seq following induction of each epigenome editing system. We observed no toxicity and only minor

---

**Fig. 2 | Distinct chromatin modifications causally instruct transcriptional responses. a**, Schematic depicting the structure of the REF reporter and its targeted integration into either a transcriptionally permissive (chr9, ON) or nonpermissive (chr13, OFF) locus. Asterisks indicate gRNA target sites within the neutral DNA context. UTR, untranslated region.; pA, poly-A tail; TE, transposable element. **b**, Representative fluorescence images (left) and expression from quantitative flow cytometry (right) showing activity of the REF reporter when integrated into either the permissive or nonpermissive locus. $n = 1,000$ individual cells; reading was performed for three independent experiments. Bars denote the geometric mean. The P value was determined by two-tailed unpaired t-test. Scale bars, 100 μm. **c–k**, Programming of a specific chromatin modification (left) and transcriptional responses in single cells (right) for H2AK119ub (**c**),

H3K9me2/3 (**d**), DNA methylation (**e**), H3K4me3 (**f**), H3K27ac (**g**), H3K79me2 (**h**), H4K20me3 (**i**), H3K36me3 (**j**) and H3K27me3 (**k**). Left: histogram showing relative (rel.) enrichment of the indicated chromatin modification after targeting control GFP[scFV] (gray bar), wild-type CD[scFV] (red bar) or catalytically inactive mut-CD[scFV] (blue bar) for 7 d. Displayed is the mean of at least two independent quantitations by CUT&RUN–qPCR or ChIP–qPCR. Error bars represent s.d. Rep, reporter. Right: dot plot showing $\log_{10}$ (mCherry expression) in response to epigenetic editing of the indicated chromatin mark. $n = 250$ individual cells; bars denote geometric mean of the population; gray shading indicates control geometric mean. Reading was performed for four independent experiments. P values were calculated by one-way ANOVA with Tukey's multiple-test correction.

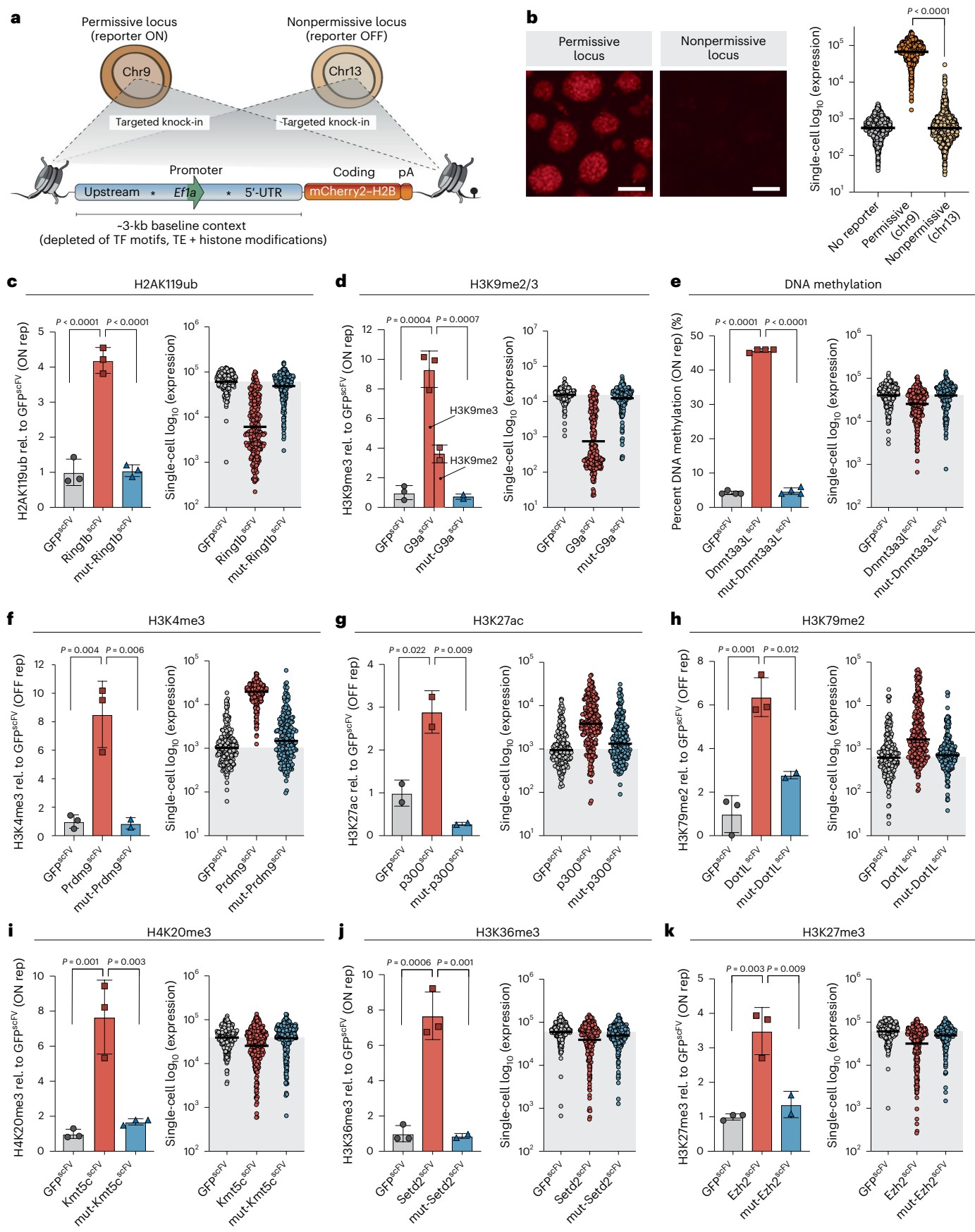

changes in global gene expression (Fig. 1k). An exception is p300-CD$^{scFV}$, which elicited indirect expression changes and reduced cell viability. To mitigate this, we limited p300-CD$^{scFV}$ induction by using DOX at a concentration 20-fold lower (Extended Data Fig. 2c,d). Overall, the data suggest that OFF-target and/or indirect effects are minimized with our modular CD$^{scFV}$ recruitment design.

Thus, we developed a flexible epigenome editing toolkit capable of programming high levels of nine key chromatin modifications to specific endogenous loci. The system includes multiple controls to isolate the causal function of chromatin modifications per se, is compatible with combinatorial targeting, and can track temporally resolved responses and epigenetic memory.

## Chromatin modifications can instruct transcriptional outputs

To investigate the direct regulatory role of chromatin modifications on transcription, we initially engineered a reporter system that facilitates quantitative single-cell readouts. We embedded the endogenous *Ef1a* (*Eef1a1*) core promoter (212 bp) into a contextual DNA sequence (~3 kb) selected from the human genome to be feature neutral: it carries no transposable elements, has ~50% GC content and has minimal TF motifs (Fig. 2a). We inserted the sequences for this 'reference' (REF) reporter into two genomic locations, chosen to be either permissive (chromosome (chr)9) or nonpermissive (chr13) for transcriptional activity (Fig. 2a). Consistently, knock-in to the permissive locus supported strong expression (ON), whereas the nonpermissive landing site resulted in minimal activity (OFF), which partially reflects acquisition of Polycomb silencing (Fig. 2b and Extended Data Fig. 2e,f). These identical reporters residing within distinct genomic locations thus enable assessment of both activating and repressive activity of induced chromatin modifications on the same underlying DNA sequence.

We targeted each CD$^{scFV}$ to each reporter and confirmed significant programming of the expected chromatin modification (Fig. 2c–k, left). Importantly, catalytic mutant effectors (mut-CD$^{scFV}$) did not change the chromatin state (Fig. 2c–k). We therefore moved to assess the functional impact of each programmed mark on transcription quantitatively and in single cells by flow cytometry. Using this sensitive strategy, we grouped chromatin marks into three functional categories: (1) modifications that instruct transcriptional repression, with penetrance across the majority fraction of cells, (2) modifications that trigger transcription activation, with majority penetrance and (3) modifications that have subtle and/or partially penetrant transcriptional effects.

The first group is characterized by the Polycomb repressive complex 1 (PRC1) modification H2AK119ub and heterochromatic H3K9me2, which is endogenously converted to H3K9me3. De novo deposition of either H2AK119ub or H3K9me2/3 is sufficient to drive silencing of the permissive (ON) reporter >100-fold in some cells, with average repression exceeding tenfold (geometric mean) (Fig. 2c,d, right). Moreover, while there was heterogeneity, >98% of cells shifted expression below the average level of control GFP$^{scFV}$. DNA methylation is also included here, as it elicited penetrant albeit modest effects, averaging 1.9-fold (±0.1 s.d.) repression (Fig. 2e and Extended Data Fig. 3a). Targeting mut-Ring1B-CD$^{scFV}$, mut-G9a-CD$^{scFV}$ or mut-Dnmt3a3l-CD$^{scFV}$ had no significant impact on expression (Fig. 2c–e). This indicates that H2AK119ub and H3K9me2/3 marks per se are sufficient to causally instruct silencing, while partial (~50%) DNA methylation causes moderate repression.

The second group induced quantitative transcriptional activation when deposited at a repressed promoter and comprised H3K4me3, H3K27ac and H3K79me2. Programming each mark triggered a reproducible population shift leading to 18.1-fold (±3.8 (s.d.)), 3.5-fold (±0.2) and 2.4-fold (±0.4) increased expression, respectively, with some cells activating >50-fold over the GFP$^{scFV}$ control (Fig. 2f–h). Moreover, programming H3K4me3 to the active (ON) locus shifted cells into a homogenous state of maximal expression (Extended Data Fig. 3b). Targeting catalytically inactive mut-Prdm9-CD$^{scFV}$, mut-p300-CD$^{scFV}$ or mut-Dot1l-CD$^{scFV}$ did not affect transcription, indicating that the marks per se are responsible.

The third functional group elicited variable or weak repressive responses and comprised H4K20me3, H3K36me3 and H3K27me3. Repression amounted to 1.6-fold (±0.3 (s.d.)), 1.2-fold (±0.1) and 1.5-fold (±0.1) (geometric mean) at the population level, respectively, with the relevant catalytic mutant CD$^{scFV}$ controls bearing no effect (Fig. 2i–k). Notably, these marks triggered repression in a highly heterogeneous manner, >50-fold in some cells, but with the majority of cells remaining within the original expression range (Fig. 2i–k and Extended Data Fig. 3c). Because other equivalently enriched modifications provoked more penetrant impacts, these heterogeneous responses likely reflect biological rather than technical outcomes.

We next assessed other response parameters to programming each modification. We first captured the temporal dynamics of transcriptional changes, noting that, while the majority of the response occurred by day 2, differences between marks arose. For example, H3K9me2 elicits its repressive activity faster than H2AK119ub (Extended Data Fig. 4a). We also found that promoter accessibility correlated well with the directionality of gene expression change induced by epigenetic editing, supporting an impact of modifications on transcriptional levels rather than post-transcriptional levels (Extended Data Fig. 4b). Finally, we observed a dose-dependent correlation between the induction level of the epigenetic editing machinery and target expression changes, suggesting that gene activity can be tuned with chromatin modifications (Extended Data Fig. 4c).

**Fig. 3 | De novo H3K4me3 triggers transcription upregulation. a**, H3K4me3 enrichment over the transcriptional start site (TSS) ±5 kb in wild-type and *Mll2*$^{CM/CM}$ ESCs, stratified according to H3K4me3 changes in *Mll2*$^{CM/CM}$ ESCs. **b**, MA plot of expression change for each gene in *Mll2*$^{CM/CM}$ ESCs, colored by whether the promoter loses H3K4me3 (green) or retains H3K4me3 (red). WT, wild type. **c**, Bar plots showing expression of the indicated genes in wild-type, *Mll2*$^{CM/CM}$ and *Mll2*$^{CM/CM}$ + Prdm9$^{scFV}$ ESCs, in which H3K4me3 has been programmed back to a repressed promoter that previously lost H3K4me3. Shown is the mean of three biological replicates assayed by qPCR with reverse transcription (RT–qPCR). Error bars represent s.d., and significance of rescue was calculated by two-tailed unpaired *t*-test. **d**, Bar plots of endogenous gene expression in wild-type ESCs and upon programming H3K4me3 with Prdm9$^{scFV}$ or control mut-Prdm9$^{scFV}$. Data are the mean of biological triplicates; error bars represent s.d. Significance was calculated by one-way ANOVA with Tukey's correction. *Oct6* (*Pou3f1*). **e**, Dot plots showing single-cell expression of the OFF reporter after targeting with different H3K4me3 effectors: Prdm9$^{scFV}$ (left) or Setd1a$^{scFV}$ (right). *n* = 500 individual cells; bars denote the geometric mean. Reading was performed for three independent experiments. **f**, Bar plots of mean gene expression in wild-type ESCs targeted with Setd1a$^{scFV}$ or untargeted (−DOX), assayed by RT–qPCR from biological triplicates. Error bars, s.d. with significance calculated by two-tailed unpaired *t*-test. **g**, Epigenetic landscape response at the OFF reporter before (−DOX) and after (+DOX) targeted H3K4me3 programming. Histone modification enrichment is indicated across ~2 kb. *n* = 3 independent experiments with significance calculated by two-tailed unpaired *t*-test. **h**, Left: bar plots showing that the mean percentage of mCherry-positive cells is restricted after (+DOX) H3K4me3 installation by Prdm9$^{scFV}$ in the presence or absence of the p300 inhibitor (inh) A485. Con, control. Data are biological triplicates; error bars represent s.d. *P* values were calculated by two-way ANOVA with Tukey's correction. Right: relative abundance of the indicated histone modifications after programming H3K4me3 (+DOX) in the presence of A485. *n* = 3 independent experiments, with significance calculated by two-tailed unpaired *t*-test. **i**, Schematic of the strategy and scatterplot showing genes that depend on MLL2-mediated promoter H3K4me3 for upregulation (up) during the ESC transition to EpiLCs. Significant genes are colored. **j**, Dot plots showing normalized log expression of each gene (*n* = 498) that is normally activated in wild-type EpiLCs but fails to be upregulated in *Mll2*$^{CM/CM}$ cells. Where indicated, \**P* < 0.05, \*\**P* < 0.01, \*\*\**P* < 0.001.

In summary, by exploiting a sensitive single-cell readout and precision epigenome editing, we capture that de novo epigenetic marks can causally instigate quantitative changes in gene expression. We report the magnitude and nature of these changes, which vary from robust, to subtle and/or heterogeneous, to nonfunctional, depending on the identity of the mark and the genomic context. These data thus support the principle that each chromatin modification tested here has the potential to directly influence transcription

output when measured at an appropriate quantitative and single-cell resolution.

## H3K4me3 can trigger transcription upregulation

Among the salient impacts of epigenome editing was robust reporter activation by H3K4me3 deposition (Fig. 2f). H3K4me3 is universally correlated with transcriptional activity, yet whether it can instruct expression or is merely a consequential marker is intensely debated[42,43].

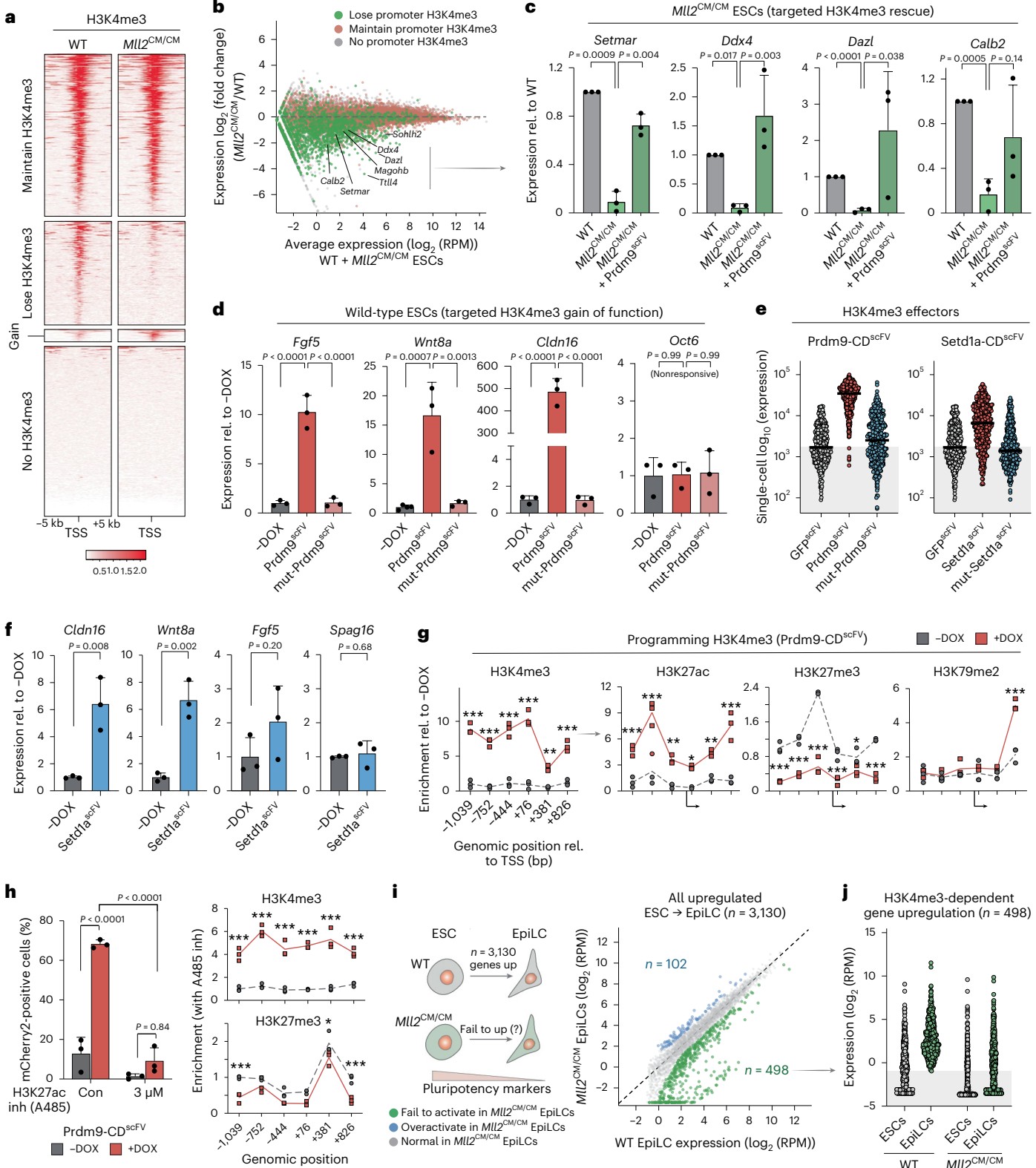

To probe this further, we generated ESCs with homozygous knock-in Y2602A catalytic point mutations (CM) in the H3K4 methylase gene *Mll2* (*Kmt2b*), which specifically disrupts its enzymatic activity (*Mll2*^CM/CM). This enables loss of H3K4me3 per se to be assessed without confounding issues associated with deletion of MLL2 protein and complexes. CUT&RUN-seq identified 3,102 H3K4me3 promoter peaks that are lost in *Mll2*^CM/CM ESCs, while 15,244 promoters retained H3K4me3 due to redundant H3K4me3 methylases (Fig. 3a). Among promoters depleted of H3K4me3, almost all exhibited reduced expression as a consequence ($P < 0.0001$), while promoter clusters that maintained H3K4me3 showed no change ($P = 0.53$) (Fig. 3b and Extended Data Fig. 5a). Indeed, 98% (347) of significantly differential genes (adjusted $P$ ($P_{adj}$ < 0.05, fold change > 2) within the H3K4me3-loss cluster were downregulated, with just 2% (six) upregulated (Extended Data Fig. 5b). Profiling the chromatin landscape in *Mll2*^CM/CM ESCs revealed that elimination of promoter H3K4me3 triggered a secondary depletion of H3K27ac and gain of H3K27me3 domains (Extended Data Fig. 5c,d). Thus, specifically removing H3K4me3 unmasks the potential for silencing a subset of genes that were previously active.

To distinguish whether H3K4me3 simply safeguards against silencing versus whether H3K4me3 is capable of instigating transcriptional upregulation, we next programmed H3K4me3 back to genes that became repressed due to H3K4me3 loss in *Mll2*^CM/CM cells. Upon DOX induction of Prdm9-CD^scFV to restore H3K4me3, all targeted genes showed a trend of reactivation, with five out of seven reaching significant transcriptional rescue, including *Setmar*, *Ttll4* and *Ddx4* (Fig. 3c and Extended Data Fig. 6a,b). By contrast, the control *Pldn* (*Bloc1s6*) gene, which was downregulated without H3K4me3 loss, exhibited no reactivation (Extended Data Fig. 6b). Thus, (re)acquisition of H3K4me3 can activate endogenous genes that were previously expressed before genetically-induced depletion of H3K4me3. To examine whether H3K4me3 can also instigate expression of genes that were never active in a given cell type, we targeted H3K4me3 to eight silent promoters in naive ESCs. Installation of H3K4me3 resulted in significant activation at three out of eight of these genes, with maximal upregulation reaching >400-fold at *Cldn16* (Fig. 3d and Extended Data Fig. 6c). Importantly, targeting the catalytically inactive mut-Prdm9-CD^scFV had no detectable impact. Forced H3K4me3 programming at promoters can therefore overcome silencing to instigate transcription, at least at some genes, and this reflects activity of the H3K4me3 mark itself.

To validate this further, we generated a second H3K4me3 effector based on the catalytic core of SET domain-containing protein 1A (Setd1a-CD^scFV). We targeted compound Setd1a-CD^scFV to the OFF reporter, which triggered robust activation (Fig. 3e). Indeed, >85% of cells expressed above the control average in response to Setd1a-CD^scFV-mediated H3K4me3, with 3.3-fold (±0.3 s.d.) increased transcription across the population. The catalytically inactive mut-Setd1a-CD^scFV effector had no impact (Fig. 3e). We also targeted endogenous genes with Setd1a-CD^scFV and again observed significant transcription activation of some (two of four) (Fig. 3f). Of note, the relative activation induced by each effector (Prdm9-CD^scFV > Setd1a-CD^scFV) correlated with the amount of H3K4me3 they respectively deposited (Extended Data Fig. 6d), suggesting a dose-dependent impact of H3K4me3. Consistently, responding cells within a population acquire more H3K4me3 than less-responsive cells (Extended Data Fig. 6e). In sum, independent targeted gain-of-function approaches support the principle that sufficient H3K4me3 can trigger transcription at otherwise silent promoters. Furthermore, the data show that, in some instances, de novo H3K4me3 is not sufficient to activate transcription.

### Functional implications of promoter H3K4me3

We next investigated the mechanisms through which H3K4me3 operates by initially asking whether de novo H3K4me3 remodels the local chromatin landscape. Installing H3K4me3 to the OFF reporter caused a highly significant secondary depletion of the Polycomb mark H3K27me3

(Fig. 3g), which is the reciprocal response to removing H3K4me3 (Extended Data Fig. 5c,d). Programming H3K4me3 also triggers a major gain of H3K27 acetylation (Fig. 3g). Because histone acetylation is linked with active transcription, we tested the functional implications of this by installing H3K4me3 with or without the p300 and CREB-binding protein (CBP) inhibitor A485, which blocks acetyltransferase activity[44]. A485 did not affect efficient programming of H3K4me3 but did restrict downstream activation to <10% of cells, compared to ~70% in no-inhibitor controls (Fig. 3h and Extended Data Fig. 6f). Programming H3K4me3 in the presence of A485 also largely blocked displacement of H3K27me3. This supports a hierarchical model by which de novo H3K4me3 functionally operates, at least partially, by facilitating promoter acetylation and evicting epigenetic silencing systems such as Polycomb.

To examine whether H3K4me3 contributes to gene activation programs during development, we induced differentiation of naive *Mll2*^CM/CM ESCs into formative epiblast-like cells (EpiLCs). This entails activation of 3,130 genes ($P_{adj}$ < 0.05, $\log_2$ (fold change) > 2) in wild-type cells. The majority of these activated normally in *Mll2*^CM/CM EpiLCs, while naive and formative markers also exhibited dynamics indistinguishable from those of the wild type, suggesting that mutant EpiLCs acquire appropriate cell identity (Fig. 3i and Extended Data Fig. 7a–c). Nevertheless, among the 3,130 genes that normally undergo upregulation, 498 exhibited significant failure in *Mll2*^CM/CM EpiLCs (Fig. 3i,j). Most (63%) were either silent or lowly expressed in precursor ESCs ($\log_2$ (reads per million (RPM)) < 0.1), suggesting that MLL2-mediated H3K4me3 participates in timely de novo activation of genes during cell fate transition (Extended Data Fig. 7d). For example, *Col1a2* and *Spon1* normally acquire promoter H3K4me3 and evict H3K27me3 coincident with activation in EpiLCs but fail to be upregulated in *Mll2*^CM/CM EpiLCs (Extended Data Fig. 7e).

In summary, our complementary precision gain-of-function and loss-of-function strategies demonstrate that de novo H3K4me3 installation is sufficient to remodel the local chromatin landscape and instigate transcription upregulation, at least at some genes, rather than only reflecting a consequence of activity.

### Epigenetic–genetic interactions modulate transcription

The precise functional impact of a given histone modification is likely dependent on contextual interactions, including with the underlying DNA sequence features. To investigate this interplay, we generated an allelic series of reporters in which each comprises an identical ~3-kb REF sequence but is distinguished by insertion of short DNA motifs (8–14 bp) (Fig. 4a). We employed motifs corresponding to binding sites of TFs (OCT4, OTX, EBOX, GATA), or that impact chromatin architecture by recruitment of proteins (TFs CTCF, YY1) or by forming G-quadruplexes (G4-U, G4-D)[45,46] (Fig. 4a and Extended Data Fig. 8a). We knocked in the sequence for each reporter to the permissive (ON) and nonpermissive (OFF) genomic landing sites (Fig. 2a). Most motifs did not impact baseline expression, albeit the inclusion of CTCF, G4-U or YY1 motifs subtly altered activity (Fig. 4b). Overall, we generated a reporter series that carries specific DNA sequence variants within highly controlled genomic environment(s).

To systematically explore *cis* genetic–epigenetic functional interplays, we installed each chromatin modification to each reporter, within each genomic context. We first focused on the 'ON' reporter(s), where repressive modifications generally exhibited coherent effects across the series. For instance, H3K9me2/3 and H2AK119ub manifested strong silencing irrespective of most underlying motifs (Fig. 4c). Nevertheless, we did observe striking interactions between specific marks and *cis* genetics (Fig. 4c and Extended Data Fig. 8b,c). For example, the presence of YY1 motifs within an otherwise identical sequence effectively blocked H2AK119ub- and H3K27me3-mediated transcriptional repression. Such YY1 sites also dampened the quantitative impact of DNA methylation and H3K9me2/3 (Fig. 4c). Conversely, OTX motifs rendered the reporter more amenable to repression by

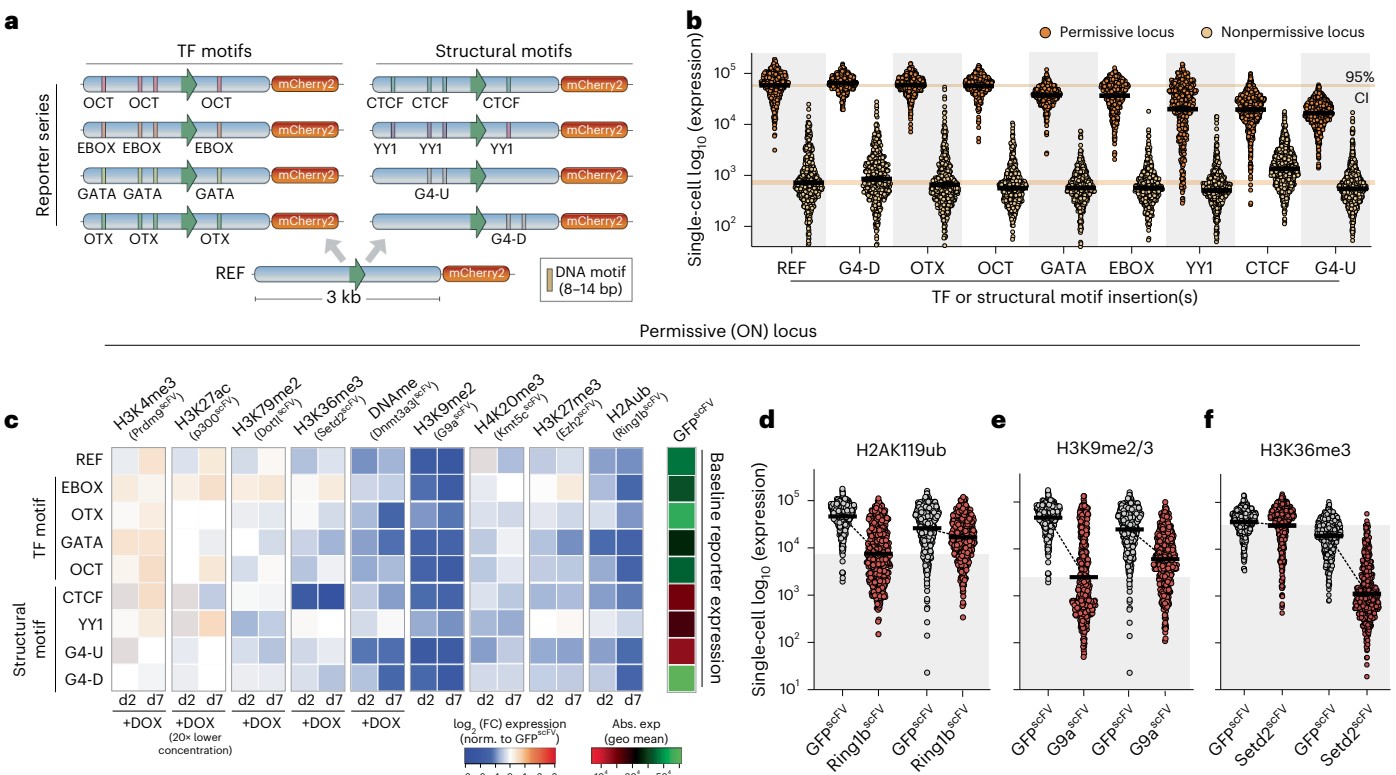

**Fig. 4 | Functional interplay between chromatin marks and TF motifs.**
**a**, Schematic of the reporter series in which each is identical apart from the insertion of specific short sequence motifs. **b**, Dot plots of mCherry2 expression from the indicated reporter type, integrated in either the permissive or the nonpermissive locus. Each data point represents a single cell ($n = 500$), and bars denote the geometric mean. Reading was performed for four independent experiments. CI, confidence interval. **c**, Heatmap showing the $\log_2$ (fold change (FC)) in transcription at the ON locus upon programming the indicated chromatin mark ($x$ axis) to the indicated $cis$ motif reporter ($y$ axis),

relative to control GFP$^{scFV}$ targeting. Data are shown after 2 d (d2) and 7 d (d7) of DOX-induced epigenetic editing and correspond to the average of four technical replicates. Abs., absolute; exp, expression; geo, geometric. **d**–**f**, Dot plots showing independent validations of functional interactions between programmed epigenetic marks (H2AK119ub (**d**), H3K9me3 (**e**), H3K36me3 (**f**)) and the underlying sequence motifs (REF versus +YY1 motif (**d**,**e**), REF versus +CTCF motif (**f**)). Each data point is $\log_{10}$ (expression) in a single cell ($n = 500$) carrying the indicated reporter, and bars denote geometric mean.

DNA methylation. The most striking observation related to switch-like behavior of H3K36me3. Here, programming H3K36me3 specifically on the +CTCF reporter imposed highly significant gene silencing beyond levels observed for any other modification (Fig. 4c).

To validate these contextual relationships, we generated independent knock-in reporter lines. We confirmed that inclusion of $cis$ YY1 motifs buffered the repressive activity of H2AK119ub and H3K9me2/3 (Fig. 4d,e). Quantitatively, this meant that expression was diminished by only 1.5-fold and 4.3-fold by H2AK119ub and H3K9me2/3, respectively, rather than 6.1-fold and 18.5-fold repression on the REF reporter lacking 12-bp YY1 sites. While the link between DNA methylation and OTX motifs was variable (Extended Data Fig. 8c), we reproducibly observed that inclusion of CTCF motifs licensed H3K36me3 to instruct transcriptional silencing exceeding 20-fold at the population level, with >98% of cells responding (Fig. 4f and Extended Data Fig. 8b). By contrast, there was almost no effect of programming H3K36me3 on the REF reporter.

Taken together, these data exploit a controlled system to reveal that underlying genetic motifs or variants mediate complex regulatory interactions with epigenetic modifications that quantitatively influence the transcriptional response. This implies that the precise function of a chromatin modification 'peak' is not unequivocal but highly context-dependent.

**Context-dependent impact of H3K36me3**

To explore context dependency further, we focused on the H3K36me3 interaction with CTCF. We first confirmed that transcription responses

are driven by H3K36me3 itself, as targeting mut-Setd2-CD$^{scFV}$ to the +CTCF reporter had no impact (Fig. 5a). Moreover, H3K36me3 is programmed to comparable levels on both REF and +CTCF reporters, ruling out technical disparities in epigenome editing (Fig. 5b). We therefore investigated the nature of CTCF motif dependency by first knocking-in reporters with CTCF motifs in varied orientations, which influences their ability to form chromatin loop structures[47]. Programming H3K36me3 was sufficient to repress all CTCF-containing sequences, albeit with some quantitative differences between arrangements (Fig. 5c), implying that the functional interaction between H3K36me3 and CTCF motifs is mostly independent of orientation.

We next assessed the hierarchical impact of installing H3K36me3 on other epigenomic features. We found that H3K4me3 sharply decreased upon programming H3K36me3 at the +CTCF reporter but remained unaffected in the REF context (Fig. 5d). While H3K27me3 and H3K9me3 were unaltered, DNA methylation was also specifically increased on the +CTCF reporter by H3K36me3 installation (Fig. 5d and Extended Data Fig. 9a,b). Thus, equivalent levels of H3K36me3 induce different epigenetic cascades depending on the underlying genetic sequence or motifs. To test the importance of this epigenomic cascade, we targeted Setd2-CD$^{scFV}$ to the +CTCF reporter coincident with 5-azacytidine (AZA), a potent DNA methylation inhibitor. AZA reduced the fraction of cells that switch OFF the +CTCF reporter in response to H3K36me3, implying a partial downstream role for DNA methylation (Fig. 5e). We conclude that the functional output of H3K36me3 is sensitive to the $cis$ genomic sequence and its susceptibility to epigenomic remodeling.

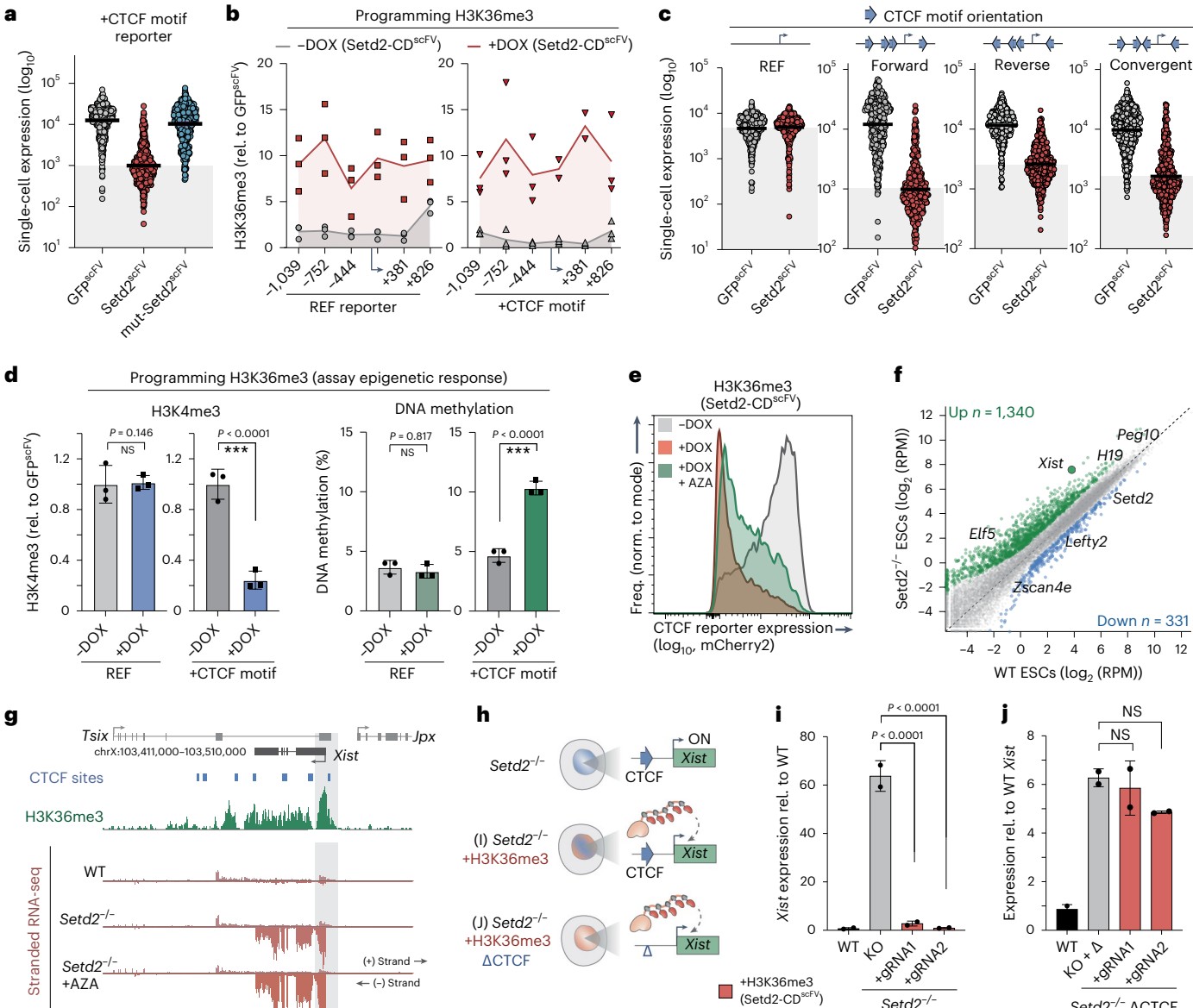

**Fig. 5 | Context-dependent influence on H3K36me3 activity. a**, Dot plots showing single-cell $\log_{10}$ (expression) of the +CTCF reporter after GFP[scFV], Setd2[scFV] (H3K36me3) or mut-Setd2[scFV] targeting for 7 d. $n = 250$ individual cells; bars denote the geometric mean. Reading was performed for four independent experiments. **b**, Relative abundance of H3K36me3 at the REF (left) or +CTCF (right) reporter assayed by ChIP–qPCR before (−DOX) or after (+DOX) Setd2[scFV] induction, across a ~2-kb region. Lines denote the mean of three replicates. **c**, $\log_{10}$ (expression) of knock-in reporters harboring +CTCF motif(s) in the indicated orientations following programming of H3K36me3 or control. Each data point represents a single cell ($n = 250$), and bars denote the geometric mean. Reading was performed for three independent experiments. **d**, Bar plots showing the enrichment of H3K4me3 (left) and percentage of DNA methylation (right) on either the REF or +CTCF reporter following programming of H3K36me3. Shown is the mean of three independent experiments. Error bars represent s.d., with significance calculated by two-tailed unpaired $t$-test. **e**, Representative flow

cytometry plot showing expression of the +CTCF reporter before (−DOX) or after (+DOX) programming of H3K36me3 with or without the DNA methylation inhibitor AZA. Freq., frequency. **f**, Scatterplot of gene expression changes in $Setd2^{-/-}$ ESCs versus wild-type ESCs, highlighting differentially expressed genes. Down, downregulated; up, upregulated. **g**, Genome view of the $Xist$ locus, showing a promoter H3K36me3 peak and expression in wild-type and $Setd2^{-/-}$ ESCs with or without AZA. **h**, Schematic of the triple (epi)genomic perturbation strategy. **i**, Mean expression level of $Xist$ in $Setd2^{-/-}$ ESCs before and after targeted programming of H3K36me3 to the promoter with an independent gRNA. Error bars represent s.d. Significance was calculated by two-tailed unpaired $t$-test. KO, knockout. **j**, $Xist$ expression in $Setd2^{-/-}$ ESCs with the promoter-proximal CTCF motif deleted, before and after programming of H3K36me3. Shown is the mean of three independent experiments. Error bars represent s.d. Significance was calculated by two-tailed unpaired $t$-test. ***$P < 0.001$.

To investigate whether H3K36me3 sequence dependency is relevant for endogenous gene regulation, we derived $Setd2$-knockout ESCs that lack H3K36me3. While H3K36me3 is rarely enriched at promoters, several of the most derepressed genes were modified by promoter H3K36me3 (Extended Data Fig. 9c–e). In particular, the X-inactivation regulator $Xist$ is associated with both promoter H3K36me3 and CTCF motifs and was highly upregulated in $Setd2^{-/-}$ cells (Fig. 5f,g).

To dissect the functional relevance of these (epi)genetic features, we programmed H3K36me3 back to the $Xist$ promoter in $Setd2^{-/-}$ female ESCs (Fig. 5h). This resulted in re-imposition of transcriptional silencing in independent $Setd2^{-/-}$ lines (>50-fold), supporting the principle that H3K36me3 can function at endogenous promoters (Fig. 5i and Extended Data Fig. 9f). To test the role of underlying CTCF motifs for this effect, we deleted the $Xist$-adjacent CTCF sequence in $Setd2^{-/-}$ ESCs and then

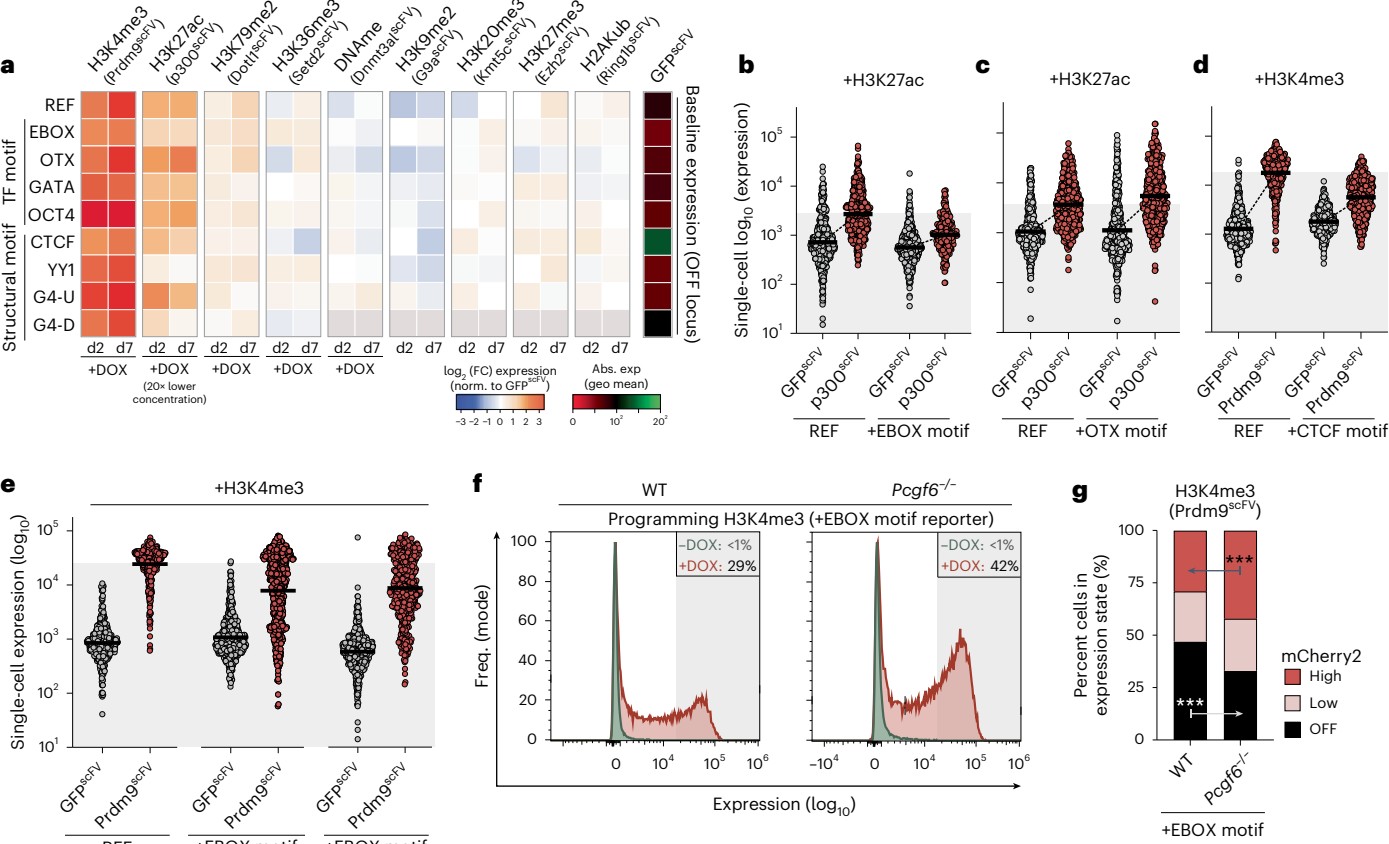

**Fig. 6 | Instructive activity of chromatin modifications is throttled by *cis* genetics. a**, Heatmap showing log₂ (fold change) in transcription at the OFF locus upon programming the indicated chromatin mark (*x* axis) to the indicated *cis* motif reporter (*y* axis), relative to control GFP^scFV targeting. Data are shown after 2 d and 7 d of DOX-induced epigenetic editing and correspond to the average of four technical replicates. **b**–**d**, Dot plots showing independent validations of functional interactions between programmed epigenetic marks (+H3K27ac (**b**,**c**), +H3K4me3 (**d**)) and underlying sequence motifs (REF versus +EBOX motif (**b**), REF versus +OTX motif (**c**), REF versus +CTCF motif (**d**)). Each data point is log₁₀ (expression) of the indicated reporter variant in a single cell ($n = 500$) after control GFP^scFV or specific CD^scFV epigenetic editing for 7 d. Bars denote the geometric mean. **e**, Dot plots showing that single-cell expression of +EBOX reporters in independent lines is restricted after induction of H3K4me3, relative to the control REF reporter. $n = 500$ individual cells; bars denote the geometric mean. **f**, Representative flow cytometry plot showing +EBOX reporter expression before (−DOX) or after (+DOX) Prdm9^scFV targeting for 5 d in either a wild-type or a *Pcgf6*^−/− genetic background. **g**, Contingency plot indicating that an elevated fraction of cells acquire the 'high' expression state following H3K4me3 programming in *Pcgf6*^−/− ESCs. Significance was calculated by two-way ANOVA with Tukey's correction. ***$P < 0.001$.

again re-installed H3K36me3 by epigenome editing. The absence of this CTCF motif resulted in failure of H3K36me3 to reimpose silencing at *Xist* (Fig. 5j). This suggests that the interplays between *cis* sequence and epigenome function we identified are physiologically relevant.

### Functional interaction between activating marks and TF motifs

To examine genetic–epigenetic interplays further, we tested interactions at the nonpermissive locus. We found that H3K27ac is reciprocally modulated by short motifs, with EBOX and YY1 attenuating and OTX enhancing H3K27ac output (Fig. 6a–c). We also reproducibly confirmed that H3K4me3 is quantitatively impacted by underlying OCT4, CTCF and EBOX *cis* contexts, with the latter attenuating H3K4me3 activity (Fig. 6d,e and Extended Data Fig. 8c). Because EBOX can recruit repressive PRC1.6 complexes[48], we hypothesized that this counteracts H3K4me3. To test this, we generated *Pcgf6*^−/− cells that lack PRC1.6 and installed H3K4me3 to the +EBOX reporter. This rescued H3K4me3 functional attenuation relative to wild type (Fig. 6f,g), suggesting that PRC1.6 recruitment via EBOX motifs provides a genetically encoded mechanism to threshold maximal induction. These data further underscore the relevance of genomic context for quantitative epigenome function.

### Epigenetic memory of chromatin marks in ESCs

We next deployed our editing toolkit to interrogate other regulatory questions. We first asked whether epigenetically programmed transcriptional states are inherited through mitotic divisions and whether DNA context impacts this. We targeted each CD^scFV to each reporter in each genomic context to install the panel of epigenetic modifications and then withdrew DOX to remove the inducing signal. Despite robust initial transcriptional responses, upon a 7-d washout of the editing machinery, we observed no significant long-term memory of either activated or repressed reporter activity (Fig. 7a,b). This was evident for all tested genetic contexts and regardless of genomic location, implying that transcriptional changes instigated by de novo chromatin marks are robustly reset to baseline in naive ESCs. Such lack of 'epigenetic memory' may reflect the unique ESC cell type, as acquired heterochromatin domains also do not propagate in naive pluripotent cells but do so in differentiated cellular contexts[40].

### Functional synergy of H3K27me3 and H2AK119ub

We finally asked whether and to what extent combinatorial chromatin marks interact with one another to synergize or antagonize their quantitative effects on transcription. We exploited our modular system to induce pairs of CD^scFV, focusing on marks that co-occur on

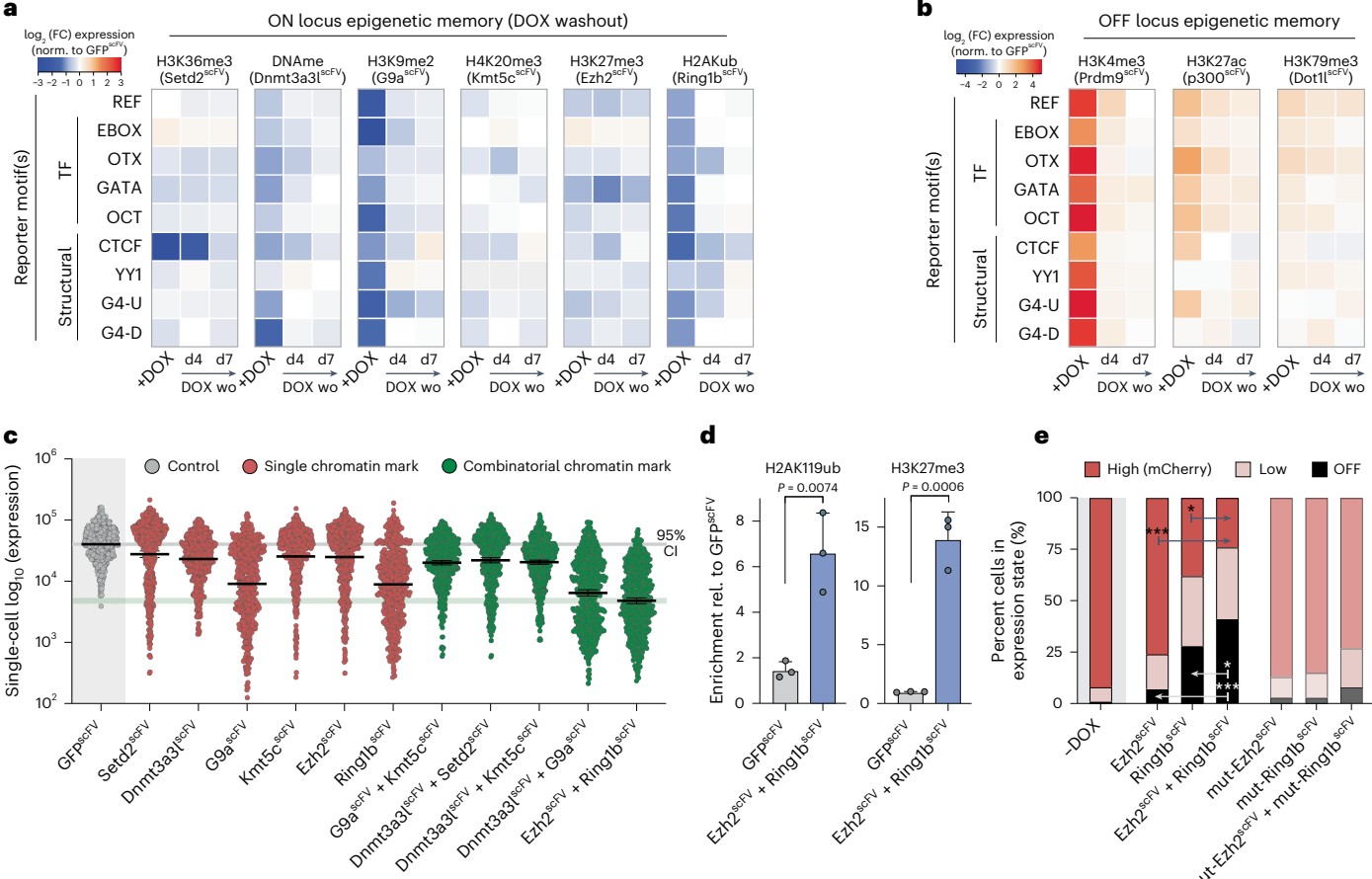

**Fig. 7 | Functional synergy between H3K27me3 and H2AK119ub.**
**a,b**, Heatmaps showing $\log_2$ (fold change) in transcription upon programming the indicated chromatin mark (*x* axis) to the indicated motif reporter (*y* axis) and then upon washout (DOX wo) for 4 d (d4) or 7 d (d7) to assay epigenetic memory. Shown are transcriptional persistence effects at the ON locus (**a**) and the OFF locus (**b**). **c**, Representative dot plots indicating $\log_{10}$ (expression) after control GFP$^{scFV}$, single CD$^{scFV}$ or multiplex CD$^{scFV}$ targeting for 7 d to program combinatorial marks. Each data point represents a single cell (*n* = 500), and bars denote the geometric mean. **d**, Bar plots showing enrichment of H2AK119ub (left) and H3K27me3 (right) on the ON REF reporter assayed by CUT&RUN–qPCR following control GFP$^{scFV}$ or combinatorial Ezh2$^{scFV}$ and Ring1b$^{scFV}$ targeting. Shown is the mean of three biological replicates; error bars represent s.d.; significance was determined by two-tailed unpaired *t*-test. **e**, Contingency plot indicating that an elevated fraction of cells acquire the 'OFF' expression state following combinatorial H3K27me3–H2AK119ub programming. Significance was calculated by two-way ANOVA with Tukey's correction. *$P < 0.05$, ***$P < 0.001$.

chromatin. Among functional interactions, we noted that co-deposition of H3K9me2/3 and DNA methylation (G9a-CD$^{scFV}$ and Dnmt3a3l-CD$^{scFV}$) increased the transcriptional response, relative to each mark singularly (Fig. 7c). Specifically, while the maximal level of repression among single cells was similar to that of H3K9me2/3 alone, there was an increase in the fraction of cells that fully silenced expression when DNA methylation was co-targeted (35% ± 6% versus 41% ± 4%), indicating that these marks may cooperate to confer robustness (Fig. 7c and Extended Data Fig. 10a). Accordingly, when DNA methylation was inhibited following H3K9me2/3 deposition using AZA (Extended Data Fig. 10b), an elevated percentage of cells did not fully silence reporter activity (Extended Data Fig. 10c).

The most striking synergy, however, came from co-targeting H3K27me3 and H2AK119ub (Ezh2-CD$^{scFV}$ and Ring1b-CD$^{scFV}$), which instigated a significant increase in the single-cell penetrance of silencing, relative to installing either mark individually (Fig. 7c–e and Extended Data Fig. 10d,e). We confirmed that significant levels of both H3K27me3 and H2AK119ub were programmed by combinatorial targeting (Fig. 7d). Moreover, independent ESC lines supported the notion that multiplex epigenetic editing led to functional synergism, with 41% (±7% s.d.) of cells reaching the fully OFF state, relative to deposition of H2AK119ub (28% ± 7%, *P* = 0.029) or H3K27me3 (7% ± 3%, *P* < 0.001) alone (Fig. 7e and Extended Data Fig. 10e). Importantly, catalytic mutant effectors

registered only a subtle negative effect on reporter activity. Overall, these data suggest that combinatorial chromatin modifications can increase the single-cell penetrance of transcriptional responses, with H3K27me3 and H2AK119ub together exemplifying effects that are at least additive and potentially synergistic. Such functional interactions between marks provides an additional layer of context dependency and further uncovers the parameters that modulate the quantitative effects of chromatin modifications.

## Discussion
The extent to which specific chromatin modifications are causative or consequential of DNA-templated processes and in which contexts is an area of intense debate[37,42]. To address this, we developed a comprehensive epigenome editing toolkit that enables de novo installation of nine key chromatin marks at precise genomic loci with high efficiency. We leverage this platform to capture that acquisition of each tested modification is sufficient to trigger at least some transcriptional response, in at least some contexts. The precise quantitative impact and single-cell penetrance of a mark is contingent on multiple contextual factors, however, and we provide direct evidence that the underlying DNA sequence, genomic location and combinatorial modifications interact to modulate the overall expression output. This is likely further complicated by cell type context. Thus, while

chromatin marks have the potential to causally instruct transcription programs, they represent one regulatory layer within multiple nonlinear governing mechanisms.

Among our findings, we charted a function for H3K4me3, which is an evolutionary conserved marker of transcriptionally active promoters[7,49]. Nevertheless, loss-of-function studies across model systems suggest that H3K4me3 is not required for the majority of gene expression[43,50,51]. Using an array of H3K4me3 programming tools, catalytic mutant controls and *Mll2*[CM/CM] ESCs that specifically lack H3K4me3, we uncover that H3K4me3 per se can directly impact transcription. The cumulative studies point toward a dual-feedback relationship in which transcription itself promotes downstream accumulation of H3K4me3, but, reciprocally, de novo acquisition of H3K4me3 can trigger transcription. Mechanistically, H3K4me3 acquisition initiates an epigenetic cascade including loss of H3K27me3 and gain of promoter acetylation, which is necessary for H3K4me3-mediated effects. This is likely reinforced by H3K4me3 promoting RNA polymerase II pause release[52] and by the transcription machinery having affinity for the mark[53,54]. However, H3K4me3 activity is ultimately contingent on the appropriate TF in the cellular milieu, and, indeed, only a fraction (~35%) of silent genes responded to de novo deposition. In this respect, acquisition of H3K4me3 may instruct transcriptional upregulation primarily by antagonizing repression systems[55], thereby establishing a permissive environment for the relevant TF. This may require a threshold level of H3K4me3, with our optimized toolkit amplifying both the magnitude and genomic breadth of de novo H3K4me3 domains, thus unmasking functionality.

Understanding the regulatory relationship(s) between the genome and the epigenome is key toward deciphering how DNA sequence variants influence molecular outputs and phenotypic traits[56]. By quantifying the instructive potential of multiple marks, we were subsequently able to dissect how underlying TF motifs interact with chromatin functionalities to tune expression. For example, EBOX motifs act as genetically encoded signals to threshold epigenetic activation by de novo H3K4me3 or H3K27ac. More strikingly, H3K36me3 exhibits switch-like behavior in the context of *cis* CTCF motifs, a relationship relevant to endogenous *Xist* regulation. The interplay between overlapping chromatin modifications represents a further contextual parameter for genome regulation. Indeed, combinatorial H3K27me3 and H2AK119ub enhances the fraction of responsive cells but not absolute repression capacity. Such epigenetic 'penetrance' effects imply an equilibrium of regulatory forces, where programming more influential (or combinatorial) marks has greater probability of overcoming the governing status quo in each cell. Importantly, however, while our data imply that chromatin marks can be instructive, they emphasize that their impacts are context-dependent. This argues against a hard-wired 'histone code' where specific patterns of chromatin marks elicit a specific output and instead points toward a nonlinear regulatory network that produces quantitative outputs depending on myriad inputs including TF binding, chromatin architecture, *cis* genetics, cell type and indeed epigenetic modifications themselves.

In summary, our study captures the principles of how de novo chromatin modifications can causally influence gene expression across contexts. Moreover, the modular epigenetic editing toolkit provides a framework to explore regulatory mechanisms across DNA-templated processes and to precisely manipulate chromatin for desirable responses in disease models.

## Online content

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

## Methods

### Cell culture

Wild-type mouse ESCs (mESCs) were derived freshly (mixed 129/B6, XY) and cultured on gelatin-coated cell culture plates under naive conditions (2i/leukemia inhibitory factor (LIF)), in accordance with the approved protocol by the laboratory animal management and ethics committee of the EMBL under license 20191001MBJH. Routine passaging was performed in N2B27 basal culture medium (NDIFF, Takara, y40002), supplemented with 1 µM PD0325901 and 3 µM CHIR99021 (both from Axon Medchem), 1,000 U ml⁻¹ LIF (in-house production), 1% FBS (Millipore) and 1% penicillin–streptomycin (Gibco). All culture media were filtered through a 0.22-µm pore Stericup vacuum filtration system (Millipore). Cells were maintained at 37 °C in a humidified atmosphere with 5% $CO_2$ and were passaged every 2 d by dissociation with TrypLE (Thermo Fisher Scientific). Culture medium was replaced with fresh stocks daily. Mycoplasma contamination was tested routinely by the ultrasensitive qPCR assay (Eurofins).

### Generation of reporter cell lines

We designed a REF reporter to provide a baseline context and to enable the influence of subsequently inserting sequence motifs or variants to be assessed. We used the endogenous EF1α core promoter (~200 bp) embedded into a DNA sequence context selected from human chr7:41,344,065–41,346,105 (GRCh38/hg38) to be neutral in respect of genomic features, including depleted of TF motifs, GC percentage (50%), lacking retrotransposons and without epigenetic enrichments. The resulting cassette (~3 kb) was designed as a gBlock gene fragment from Integrated DNA Technologies and amplified by PCR using Q5 hot start high-fidelity polymerase (NEB, M0494S) and primers with appropriate overhangs. This was inserted by In-fusion HD Cloning into a recipient vector upstream of a Kozak sequence, the mCherry2–H2B fluorescence coding sequence and a polyA motif. The assembled reporter construct (DNA::EF1αPr::DNA::mCherry2-H2B::pA) was verified by sequencing and then amplified by PCR with Q5 polymerase, using ultramer DNA oligonucleotides (Eurofins) carrying 200-bp-long overhangs homologous to DNA sequences flanking the desired genomic insertion site(s). Specifically, we chose two intergenic genomic insertion sites that differentially support transcription. First, a permissive landing site (chr9:21,545,329, ON locus, TIGRE) and second, a nonpermissive landing site that only supports weak transcription (chr13:45,253,722, OFF locus), albeit within a euchromatic domain[57].

To insert the cassettes into each locus, we transfected 1 µg of PCR-amplified dsDNA reporter sequence into naive mESCs together with the spCas9 plasmid pX459 (Addgene, 62988), carrying a single gRNA complementary to the genomic integration site. After puromycin selection (1.2 µg ml⁻¹) for transient px459 transfection (2 d), mCherry2-positive cells that were candidates for correct insertion were purified by fluorescence-activated cell sorting. Single clones were expanded, and correct mono-allelic (hemizygous) integration of the reporter was verified by PCR genotyping and Sanger sequencing (Azenta). The full allelic series of reporter variants, which each comprised the same baseline sequence as the REF, but with insertion of several discrete TF or structural motifs (Supplementary Information) were also ordered as gBlock Gene Fragments from Integrated DNA Technologies. Generation of the complete reporter cassette and genomic integration was carried out as described above for the REF to generate a total of 18 independent reporter lines (nine reporter variants in two genomic locations), each with independent clones. We validated independent insertions of each reporter to confirm reproducibility.

### Generation of epigenetic editing toolkit constructs

Epigenetic editing tools comprising sequences for a nuclease dCas9[GCN4] and the catalytic cores of chromatin-modifying enzymes were cloned into piggyBac recipient plasmids by homology arm recombination using In-Fusion HD Cloning (Takara, 639650). Specifically, the sequence

for *Streptococcus pyogenes* dCas9[GCN4] was amplified by PCR from the PlatTET-gRNA2 plasmid[39] (Addgene, 82559) and subcloned under the control of a DOX-inducible TRE-3G promoter into a piggyBac backbone. The vector also carries sequences for the Tet-On 3G transactivator and hygromycin resistance.

For all chromatin-modifying 'effector' plasmids, the sequence for the scFV domain and an sfGFP coding sequence were amplified from the PlatTET-gRNA2 plasmid (Addgene, 82559) and fused in frame with the CD or FL mouse *Prdm9*, *p300*, *Dot1l*, *G9a*, *Kmt5c*, *Setd2*, *Ezh2* and *Ring1b*, all amplified from early-passage ESC cDNA. Sequences for the *Dnmt3a* CD and the C-terminal part of mouse *Dnmt3a* (3a3l) were amplified from pET28-Dnmt3a3l-sc27 (Addgene, 71827). The resulting constructs (collectively, CD[scFV]) were cloned in piggyBac recipient vectors under the control of the TRE-3G promoter. These vectors are also designed for constitutive expression of a neomycin resistance gene. The control GFP[scFV] effector was cloned as described above but lacks any chromatin-modifying domain. Finally, catalytic mutant (mut-CD[scFV]) effectors were also cloned as described above. Specific mutations that abolish the catalytic activity of each CD[scFV] but that retain protein stability were introduced during PCR amplification with oligonucleotide primers designed with precisely mismatched nucleotides. The catalytically inactivating point mutations introduced in each CD[scFV] are p300, D1398Y; Dot1l, GS163–164RC; Prdm9, G282A; Setd2, R1599C; Dnmt3a, C706S; G9a, Y1207del; Kmt5c, NHDC182–185AAAG; Ezh2, Y726D; Ring1b, I53S; Set1a, S1631I[29,58–64].

The gRNA plasmid, carrying an enhanced gRNA scaffold, was amplified from Addgene plasmid 60955 and cloned into a piggyBac recipient vector, which are also designed for constitutively expression of a puromycin resistance gene and TagBFP. All gRNA species used to target the epigenetic editing system were designed using the GPP Web Portal (Broad Institute). gRNA forward and reverse strands carrying appropriate overhangs (final concentration of 10 µM) were annealed in buffer containing 10 mM Tris, pH 7.5–8.0, 60 mM NaCl and 1 mM EDTA at 95 °C for 3 min and allowed to cool down at room temperature for >30 min. Annealed gRNA was ligated with T4 DNA ligase (NEB, M0202S) for 1 h at 37 °C into the piggyBac recipient vector previously digested with BlpI (NEB, R0585S) and BstXI (NEB, R0113S) restriction enzymes. Final plasmids were amplified by bacterial transformation and purified by endotoxin-free midi preparation (Zymo Research, D4200). Correct assembly and sequences were confirmed by Sanger sequencing (Azenta). All gRNA species used in this study are listed in Supplementary Table 1.

### Epigenetic editing assays

For stable integration of the epigenetic editing system, mESC lines were co-transfected to express dCas9[GCN4] and one or more CD[scFV] constructs (or control GFP[scFV]) and with gRNA plasmids in addition to the piggyBac transposase vector using a molar ratio of 10:20:2:1, respectively. Cells with successful integration of all three constructs were enriched by successive antibiotic selection with hygromycin (250 µg ml⁻¹) for 5 d, neomycin (300 µg ml⁻¹) for 5 d and puromycin (1.2 µg ml⁻¹) for 2 d. After allowing cells to recover and expand, expression of dCas9[GCN4] and CD[scFV] was induced by supplementing the culture medium with DOX (100 ng ml⁻¹) for either 2 or 7 d, with the exception of p300-CD[scFV], for which we used 5 ng ml⁻¹ DOX to mitigate against OFF targeting and toxicity. Correct induction of all epigenetic editing components results in double GFP- and BFP-positive cells (GFP⁺BFP⁺). Activity of endogenous target genes or reporter (mCherry2) was analyzed by qPCR or quantitative flow cytometry by sorting and gating to analyze only GFP⁺BFP⁺ cells that had correctly induced the editing system (typically >75% of cells). For experiments employing the p300 inhibitor A485, cells were stimulated with 100 ng ml⁻¹ DOX for 3 d and, in parallel, treated with 3 µM A485 (Cayman Chemical, 24119). When indicated, 1 µM AZA (Sigma-Aldrich) was included in media and replaced daily for 3 d in a row.

For epigenetic memory experiments, cells were washed thoroughly with PBS and subsequently cultured in the absence of DOX, which led to rapid downregulation of the epigenetic editing machinery (GFP−). Memory of reporter expression changes was quantified by flow cytometry after 4 or 7 d of DOX washout in cells that were confirmed to have fully switched off the epigenetic editing tool (BFP+GFP− cells, typically >99%).

## Transfection

DNA transfection was performed with Lipofectamine 3000 (Thermo Fisher Scientific, L30000015). Cells were seeded 1 d in advance to reach ~60% confluency on the day of transfection. Appropriate amounts of DNA were calculated according to the manufacturer's instructions. The medium was changed after 8 h and replaced with fresh antibiotic-containing medium.

## Generation of genetically edited embryonic stem cell lines

Knockout cell lines for *Pcgf6* were generated by means of CRISPR–Cas9 genome editing. Specifically, for each target gene, two plasmids (pX459) were transiently transfected into low-passage wild-type ESCs that had previously been engineered to carry a specific knock-in reporter. Each plasmid encoded one of two gRNA species targeting the flanking introns of a critical coding exon in the gene of interest (*Pcgf6*) (Supplementary Table 3) and wild-type Cas9. The critical exon was present within all known isoforms, and gRNA species were designed with the goal of specifically deleting the entire exon. After transfection, cells were selected with puromycin (1.2 µg ml−1) for 3 d and subsequently seeded at low density (1,000 cells per 10 cm²) for single-clone isolation. Following expansion, single clones were screened for homozygous genetic editing by PCR genotyping (Supplementary Table 2), and dual loss-of-function (frame-shifted) alleles were confirmed by Sanger sequencing (Genewiz).

For generation of precision-edited catalytic mutant *Mll2* (*Mll2*CM/CM) and *Setd2*−/− lines, homozygous ESCs were derived freshly from heterozygous FVB crosses of mice carrying either an *Mll2* (Y2602A) or a *Setd2*-null allele, under Italian Ministry of Health authorization code 101/2024-PR. To generate the *Setd2*−/− ΔCTCF lines, *Setd2*−/− ESCs were transiently transfected with a plasmid (pX459) expressing a gRNA targeting a CTCF site (identified using ChIP–seq data from Nora et al.[65] and by manual inspection of the CTCF consensus sequence) upstream of the *Xist* promoter. After transfection, cells were selected with puromycin (1.2 µg ml−1) for 3 d and subsequently seeded at low density (1,000 cells per 10 cm²) for single-clone isolation. Following expansion, single clones were screened for genetic editing by PCR genotyping followed by Sanger sequencing (Genewiz). Homozygosity was confirmed by Sanger sequencing and restriction digest with BbsI, the cut site of which is absent in deletion mutants.

## Flow cytometry

Cells were washed with PBS and gently dissociated into a single-cell suspension using TrypLE, followed by resuspension in FACS buffer composed of PBS with 1% FBS, and filtered through a 40-µm cell strainer (BD, cup-Filcons, 340632). A FACSAria III (Becton Dickinson) and the Attune NxT Flow Cytometer (Thermo Fisher Scientific) were used for sorting and analysis, respectively. Ninety-six-well plates containing the different combinations of reporter × epigenetic effector cell lines were analyzed using the Attune NxT Flow Cytometer Autosampler, and resulting data were used to generate the heatmaps shown in Figs. 4c and 5a. Alternatively, specific reporter × epigenetic effector cell lines were generated and cultured in 12-well plates, and samples were analyzed one by one using the single-sample line of the Attune NxT Flow Cytometer. Flow cytometry data analysis was performed with FlowJo version 10.5.3 (Tree Star).

To generate the dot plots shown in this study, FlowJo software was used first to gate for live cells and then for cells expressing all epigenetic

editing components (GFP+BFP+). The resulting population was randomly downsampled to 1,000 cells. The mCherry2 scaled fluorescent values corresponding to the relative expression intensities for each cell were exported and imported into GraphPad Prism statistical software. Dot plots were constructed with the geometric mean of the raw data shown (black bar). For dot plots representative of individual reporter expression, before transfection of the editing machinery (Fig. 4b), analysis was performed as described above, except that no GFP+BFP+ gating was performed and mCherry2 single-cell values were obtained from the whole population of live cells. To generate histograms, the parental GFP+BFP+ cell population was selected as above and the frequency distribution of the flow data was plotted versus mCherry2 fluorescence intensity using a log10 scale. The bisector gating tool was then used to split histograms into two sectors corresponding to the mCherry2 ON expression state and the mCherry2 OFF expression state, based on negative and positive controls. Alternatively, the ranged gate tool was used to split the histogram into three sectors corresponding to mCherry2 'high', mCherry2 'low' and mCherry2 'OFF' expression states. Identical gates were applied to all samples within an experiment.

Finally, to generate heatmaps, mCherry2 scaled fluorescent values for 1,000 GFP+BFP+ cells were obtained, and the geometric mean for each sample (indicating reporter expression after GFPscFV or specific CDscFV effector targeting) was calculated. The geometric mean of each CDscFV effector was normalized to the corresponding geometric mean of GFPscFV to obtain the fold change of reporter expression following epigenetic editing (geometric mean CDscFV effector/geometric mean GFPscFV). The normalized geometric mean values coming from four technical replicates of the experiments were averaged and log2 transformed. log2 (fold change) values were plotted in R statistical software (version 3.6.2) using Bioconductor packages.

## RNA extraction, library preparation and sequencing

Total RNA was extracted from cells using the Monarch Total RNA Miniprep Kit (NEB, T2010), following manufacturer instructions. Purified RNA was quantitated with a Qubit Fluorometer (Thermo Fisher Scientific) and checked for quality with an automated electrophoresis system (Agilent TapeStation System) to ensure RNA integrity (RIN > 9). Precisely 1 µg of each RNA sample was used to prepare sequencing libraries using the NEBNext Ultra II Directional RNA Library kit by the EMBL Genomics facility. Libraries were sequenced on the NextSeq Illumina sequencing system (paired-end 40 sequencing). Raw FastQ reads were trimmed to remove adaptor sequences with Trim Galore (0.4.3.1, '-phred33–quality 20–stringency 1 -e 0.1–length 20'), checked for quality and aligned to the mouse mm10 (GRCm38) genome using RNA Star (2.5.2b-0, default parameters except for '−outFilterMultimapNmax 1000'). Analysis of the mapped sequences was performed using SeqMonk software (Babraham Bioinformatics, version 1.47.0) to generate log2 (RPM) or gene length-adjusted (reads per kilobase per million mapped reads) gene expression values, and data were plotted with R statistical software (version 3.6.2). Differentially expressed genes were determined using the DESeq2 package (version 1.24.0), inputting raw strand-specific mapping counts and applying a multiple-testing-adjusted (FDR) significance threshold of $P < 0.05$ and log2 (fold change) filter where indicated.

## Quantitative PCR with reverse transcription

Total RNA was extracted from cells using the Monarch Total RNA Miniprep Kit (NEB, T2010), following manufacturer instructions. After quantification using a Qubit Fluorometer (Thermo Fisher Scientific), 1 µg of each sample was treated with DNase and used as input for cDNA synthesis by incubation with a mixture of random hexamers and reverse transcriptase (Takara PrimeScript RT Reagent Kit with gDNA Eraser, Takara Bio, RR047A). The resulting cDNA was diluted 1:10, and 2 µl of each sample was amplified using a QuantStudio 5 (Applied Biosystems) thermal cycler, employing the SyGreen Blue Mix

(PCR Biosystems) and prevalidated gene-specific primers that span exon–exon junctions. Results were analyzed using the $2^{-\Delta\Delta Ct}$ method (relative quantitation) with QuantStudio 5 software and normalized to the housekeeping gene *Rplp0*. All primers used for qPCR analysis are listed in Supplementary Table 2.

## Bisulfite pyrosequencing

DNA bisulfite conversion was performed starting from a maximum of $1 \times 10^5$ pelleted cells per sample using the EZ DNA Methylation-Direct Kit (Zymo Research, D5021) and following the manufacturer's instructions. Target genomic regions were amplified by PCR using 1 μl of bisulfite-converted DNA and specific primer pairs, one of which was conjugated to biotin, using the PyroMark PCR kit (Qiagen, 978703). Ten microliters of the PCR reaction was used for sequencing using the dispensation orders (below) generated by PyroMark Q24 Advanced 3.0 software, along with PyroMark Q24 advanced reagents (Qiagen, 970902) according to the manufacturer's instructions. Briefly, the PCR reaction was mixed with streptavidin beads (GE Healthcare, 17-5113-01) and binding buffer, denatured with denaturation buffer using a PyroMark workstation (Qiagen) and released into a PyroMark Q24 plate (Qiagen) preloaded with 0.3 μM sequencing primer. Annealing of the sequencing primer to the single-strand PCR template was achieved by heating at 80 °C for 2 min and cooling down at room temperature for 5 min. Pyrosequencing was run on the PyroMark Q24 advanced pyrosequencer (Qiagen). Results were analyzed with PyroMark Q24 Advanced 3.0 software. Primers used for PCR amplification are listed in Supplementary Table 2.

## Cleavage under targets and release using nuclease

The CUT&RUN protocol[66] was used to detect genomic enrichment of histone modifications. Cells ($2.5 \times 10^5$ to $3 \times 10^6$, depending on the selected antibody) were pelleted at 300*g* for 3 min following flow sorting. Cells were washed twice with wash buffer (1 ml of 1 M HEPES, pH 7.5, 1.5 ml of 5 M NaCl, 12.5 μl of 2 M spermidine, final volume brought to 50 ml with dH$_2$O, complemented with one Roche cOmplete Protease Inhibitor EDTA-free tablet). Pellets were then resuspended in 1 ml wash buffer and 10 μl concanavalin beads (Bangs Laboratories, BP531-3ml) in 1.5-ml Eppendorf tubes and allowed to rotate at room temperature for 10 min. The supernatant was removed by placing the samples on a magnet stand, and 300 μl antibody buffer (wash buffer supplemented with 0.02% digitonin and 2 mM EDTA) containing 0.5–3 μg of target-specific antibody was added. Samples were left to rotate overnight at 4 °C. Antibodies used were as follows: rabbit anti-H3K4me3 (Diagenode, C15410003, 0.5 μg for $2.5 \times 10^5$ cells), rabbit anti-H3K27me3 (Millipore, 07-449, 0.5 μg for $2.5 \times 10^5$ cells), rabbit anti-H3K9me3 (Abcam, ab8898, 2 μg for $3 \times 10^6$ cells), rabbit anti-H2Aub (Lys119) (CST, 8240, 3 μg for $3 \times 10^6$ cells), rabbit anti-H3K36me3 (Active Motif, 61101, 3 μg for $3 \times 10^6$ cells), rabbit anti-H3K27ac (Active Motif, 39133, 3 μg for $3 \times 10^6$ cells), rabbit anti-H4K20me3 (Abcam, ab9053, 0.5 μg for $2.5 \times 10^5$ cells).

The following day, each tube was placed on a magnetic stand, and cell–bead complexes were washed twice with cold Dig-wash buffer (wash buffer containing 0.02% digitonin) and then resuspended in 300 μl of cold Dig-wash buffer supplemented with 700 ng ml$^{-1}$ of purified protein A–MNase fusion (pA–MNase). Samples were left to rotate on a rotor at 4 °C for 1 h. After two washes with cold Dig-wash buffer, cell–bead complexes were resuspended gently in 50 μl Dig-wash buffer and placed on an aluminum cooling rack on ice to precool to 0 °C. To initiate pA–MNase digestion, 2 μl of 100 mM CaCl$_2$ was added, and samples were flicked to mix and immediately returned to the cooling rack. Digestion was allowed to proceed for 30 min and was then stopped by adding 50 μl 2× stop buffer (340 mM NaCl, 20 mM EDTA, 4 mM EGTA, 0.02% digitonin, 250 μg RNase A, 250 μg glycogen). Samples were incubated at 37 °C for 10 min to release CUT&RUN fragments from insoluble nuclear chromatin and centrifuged at 16,000*g* for 5 min at 4 °C. The supernatant was isolated by means of a magnet stand and

transferred into a new tube while cell–bead complexes were discarded. Two microliters of 10% SDS and 2.5 μl proteinase K were added, and the samples were incubated for 10 min at 70 °C. Purification and size selection of DNA were performed using SPRI beads (Beckman Coulter, B23318) following the manufacturer's instructions for double size selection with bead volume-to-sample volume ratios of 0.5× and 1.3×. Purified DNA was eluted in 30 μl ultrapure water.

For analysis of specific genomic targets, CUT&RUN DNA fragments were subjected to qPCR analysis. A 1:10 dilution was performed, and 2 μl of diluted DNA was amplified by means of a QuantStudio 5 (Applied Biosystems) thermal cycler using the SyGreen Blue Mix (PCR Biosystems) and specific primers for both targeted and control genomic regions. Relative abundance of histone marks was determined by calculating the $2^{-Ct}$ value for each genomic region of interest and normalizing it to the $2^{-Ct}$ value of a positive control genomic locus ($2^{-Ct}$ targeted region/$2^{-Ct}$ positive control region). Data were then shown as relative fold change between experimental samples and control samples (for example, CD$^{scFV}$ over GFP$^{scFV}$) with a randomly selected control replicate set as the baseline (=1). Primers used for CUT&RUN–qPCR are listed in Supplementary Table 2.

For genome-wide analysis, CUT&RUN was performed as described above, followed by library preparation. Specifically, eluted DNA fragments were purified and subjected to DNA size selection using SPRI beads (Beckman Coulter, B23318) following the manufacturer's instructions for double size selection with bead volume-to-sample volume ratios of 0.5× and 1.3×. Purified DNA was eluted in 30 μl ultrapure water, and 10 ng was input into the NEBNext Ultra II DNA Library Prep Kit for Illumina (NEB, E7645S) using the following PCR program: 98 °C for 30 s, 98 °C for 10 s, 65 °C for 10 s and 65 °C for 5 min, steps 2 and 3 repeated for 12–14 cycles. After quantification and checking for quality with an automated electrophoresis system (Agilent TapeStation System), library samples were sequenced on the NextSeq Illumina sequencing system (paired-end 40 sequencing). Raw FastQ sequences were trimmed to remove adaptors with Trim Galore (version 0.4.3.1, '-phred33 –quality 20 –stringency 1 -e 0.1 –length 20'), checked for quality and aligned to the mouse mm10 genome with the inserted mCherry reporter using Bowtie 2 (version 2.3.4.2, '-I 50 -X 800 –fr -N 0 -L 22 -i 'S,1,1.15' –n-ceil 'L,0,0.15' –dpad 15 –gbar 4 –end-to-end –score-min 'L,-0.6,-0.6''). Analysis of the mapped sequences was performed using SeqMonk software (Babraham Bioinformatics, version 1.47.0) by enrichment quantification of the normalized reads. To identify promoters with H3K4me3 changes in *Mll2*$^{CM/CM}$ cells, a 1-kb window centered on the transcription start site was quantified among replicates, and a normalized log (fold change) filter was applied between samples. Meta-plots over genomic features were constructed by quantifying 100-bp bins centered on the features of interest, and normalized cumulative enrichments were plotted.

## Chromatin immunoprecipitation followed by quantitative PCR

A total of $3 \times 10^6$ cells were dissociated with TrypLE, resuspended in PBS and pelleted at 200*g* for 4 min at room temperature. After, PBS was removed, and the cell pellet was fixed in 1 ml of 1% PFA for 10 min at room temperature, followed by centrifugation at 200*g* for 4 min. The supernatant was discarded, and fixation was quenched by adding 1 ml of 0.125 M glycine for 5 min at room temperature. Glycine was removed, and pellets were washed twice with cold PBS. Samples were kept on ice from this stage onward. Cells were resuspended in 1 ml of cold lysis buffer (50 mM HEPES, pH 8.0, 140 mM NaCl, 1 mM EDTA, 10% glycerol, 0.5% NP-40, 0.25% Triton X-100), incubated on ice for 5 min and subsequently centrifuged at 1,200*g* for 5 min at 4 °C. One wash in rinse buffer (10 mM Tris, pH 8.0, 1 mM EDTA, 0.5 mM EGTA, 200 mM NaCl) was performed, followed by another centrifugation at 1,200*g* for 5 min at 4 °C. Cell nuclei were then resuspended in 900 μl shearing buffer (0.1% SDS, 1 mM EDTA, pH 8.0 and 10 mM Tris, pH 8.0), transferred into

a Covaris milliTUBE 1 ml AFA Fiber (Covaris, 520135) and sonicated for 12 min using a Covaris ultrasonicator at 5% duty cycle, 140 PIP and 200 cycles per burst. The sonication cycle was repeated twice. Sonicated chromatin was centrifuged at 10,000$g$ for 5 min at 4 °C, and the supernatant was collected and moved to a new tube. Twenty microliters of chromatin was taken to analyze appropriate chromatin shearing on a 1% agarose gel, while 1/10 of the total volume (~90 µl) was topped up with 5× IP buffer (250 mM HEPES, 1.5 M NaCl, 5 mM EDTA, pH 8.0, 5% Triton X-100, 0.5% DOC and 0.5% SDS) and frozen at −20 °C for total input analysis. The remaining chromatin was topped up to 1 ml with 5× IP buffer, and then 30 µl Protein A/G Magnetic Beads (Thermo Fisher Scientific, 88802) and 3 µg antibody were added to each tube, and samples were left to rotate overnight at 4 °C. Antibodies used were as follows: rabbit anti-H3K36me3 (Diagenode, C15410192, 3 µg for 3 × 10$^6$ cells) and rabbit anti-H3K79me2 (Abcam, ab3594, 2 µg for 3 × 10$^6$ cells), rabbit anti-H3K9me2 (Active Motif, 39041, 3 µg for 3 × 10$^6$ cells).

The following day, beads were washed with 1 ml of 1× IP buffer by constant rotation at 4 °C for 10 min. This step was repeated twice. Two more washes were performed: the first one with DOC buffer (10 mM Tris, pH 8, 0.25 M LiCl, 0.5% NP-40, 0.5% DOC, 1 mM EDTA) and the second one with 1× TE buffer. Next, beads were resuspended in 100 µl freshly prepared elution buffer (1% SDS, 0.1 M NaHCO$_3$) and agitated constantly on a vortex for 15 min at room temperature. The eluted chromatin was transferred to a new tube, and the elution was repeated again as before by adding 50 µl elution buffer to the beads. The eluted chromatin was combined. Finally, 10 µl of 5 M NaCl was added to the eluted chromatin as well as to the thawed total input tubes. Samples were incubated overnight at 65 °C in a water bath. The next day, DNA was purified using the Zymo Genomic DNA Clean & Concentrator Kit (Zymo Research, D4011) and eluted with 30 µl ultrapure water. For qPCR analysis, samples were handled as described above for CUT&RUN–qPCR. Specifically, a 1:10 dilution was performed, and 2 µl of diluted DNA was amplified by means of a QuantStudio 5 (Applied Biosystems) thermal cycler using the SyGreen Blue Mix (PCR Biosystems) and specific primers for both targeted and control genomic regions. Relative abundance of histone marks was determined by using the 'percent input' method (the 2$^{-Ct}$ values obtained from ChIP samples were divided by the 2$^{-Ct}$ values of the input samples). Data are then shown as relative fold change between experimental samples and control samples (for example, CD$^{scFV}$ over GFP$^{scFV}$). Primers are listed in Supplementary Table 2.

**Assay for transposase-accessible chromatin with sequencing**
Cells were initially treated in culture medium with 200 U ml$^{-1}$ DNase I for 30 min at 37 °C to digest degraded DNA released from dead cells and then collected. Cells were then washed five times with PBS, dissociated with TrypLE and counted. A total of 5 × 10$^4$ cells were pelleted at 500$g$ and 4 °C for 5 min. The supernatant was removed, and the cell pellet was resuspended in 50 µl of cold ATAC resuspension buffer (10 mM Tris-HCl, pH 7.4, 10 mM NaCl, 3 mM MgCl$_2$, supplemented with 0.1% NP-40, 0.1% Tween-20 and 0.01% digitonin), followed by incubation on ice for 3 min. Lysis was stopped by washing with 1 ml of cold ATAC resuspension buffer supplemented with 0.1% Tween-20 only. Nuclei were pelleted at 500$g$ for 10 min at 4 °C. The supernatant was removed, and the nuclei were resuspended in 50 µl transposition mixture (25 µl 2× TD buffer, 2.5 µl transposase from the Illumina Tagment DNA Enzyme and Buffer Kit (20034197), 16.5 µl of 1× PBS, 0.5 µl of 1% digitonin, 0.5 µl of 10% Tween-20 and 5 µl water). Samples were incubated at 37 °C for 30 min in a thermomixer while shaking at 1,000 rpm. Next, DNA was purified using the Zymo Genomic DNA Clean & Concentrator Kit (Zymo Research, D4011) and eluted with 21 µl elution buffer. Twenty microliters was used for PCR amplification using Q5 hot start high-fidelity polymerase (NEB, M0494S) and a unique combination of the dual-barcoded primers P5 and P7 from the Nextera XT Index kit (Illumina, 15055293). The cycling conditions were

as follows: 98 °C for 30 s, 98 °C for 10 s, 63 °C for 30 s, 72 °C for 1 min, 72 °C for 5 min, repeated for five cycles. After, 5 µl of the pre-amplified mixture was used to determine additional cycles by qPCR amplification using SyGreen Blue Mix (PCR Biosystems) and the P5 and P7 primers selected above in a QuantStudio 5 (Applied Biosystems) thermal cycler. The number of additional PCR cycles to be performed was determined by plotting linear Rn (the value calculated by dividing the fluorescence of the reporter dye (SYBR Green) by the fluorescence of the passive reference dye (ROX)) versus cycle and by identifying the cycle number that corresponded to one-third of the maximum fluorescent intensity[67]. The determined extra PCR cycles were performed by placing the pre-amplified reaction back in the thermal cycler. Finally, cleanup of the amplified library was performed again using the DNA Clean & Concentration Kit (Zymo, D4014), and DNA was eluted with 20 µl water. After quantification and a quality check with an automated electrophoresis system (Agilent TapeStation System), library samples were pooled together and sequenced on the NextSeq Illumina sequencing system (paired-end 40 sequencing). Following sequencing, raw reads were first trimmed with Trim Galore (version 0.4.3.1, reads >20 bp and quality >30) and then checked for quality with FastQC (version 0.72). The resulting reads were aligned to the custom mouse mm10 genome containing the reporter using Bowtie 2 (version 2.3.4.3, paired-end settings, fragment size '0-1,000,–fr', allow mate dovetailing). Aligned sequences were then analyzed with SeqMonk (Babraham Bioinformatics, version 1.47.0) by performing enrichment quantification of the normalized reads.

### Statistical analysis
Details on all statistical analyses used in this paper, including the statistical tests used, the number of replicates and precision measures, are indicated in the corresponding figure legends. Statistical analysis of replicate data was performed using appropriate strategies in GraphPad Prism statistical software (version 8.4.3), with the following significance designations: not significant, $P > 0.05$; *$P ≤ 0.05$; **$P ≤ 0.01$; ***$P ≤ 0.001$.

### Reporting summary
Further information on research design is available in the Nature Portfolio Reporting Summary linked to this article.

## Data availability
All data derived from next-generation sequencing assays have been deposited in the ArrayExpress database under the accession codes E-MTAB-13466, E-MTAB-13467, E-MTAB-13468 and E-MTAB-12101. Additionally, previously published ChIP–seq data are used in this study[65]: GSE98671. All data are publicly available.

## Code availability
All analyses were performed using previously published or developed tools, as indicated in Methods. No custom code was developed or used.

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

## Acknowledgements

We are grateful to EMBL core facilities, particularly G. Pfister (FCF), N. Descostes (Informatics) and C. Giradot (GBCS) for experimental and logistic assistance. We thank L. Andrade for experimental support and M. Boulard and A. Boskovic for feedback. This study was funded by an EMBL program grant to J.A.H.

## Author contributions

C.P. performed experiments, data analysis and co-wrote the paper. M.M., S.T. and V.C. performed experiments. J.A.H. designed and supervised the study, performed data analysis and wrote the paper.

## Funding

## Competing interests

The authors declare no competing interests.

## Additional information

**Extended data** is available for this paper at https://doi.org/10.1038/s41588-024-01706-w.

**Correspondence and requests for materials** should be addressed to Jamie A. Hackett.

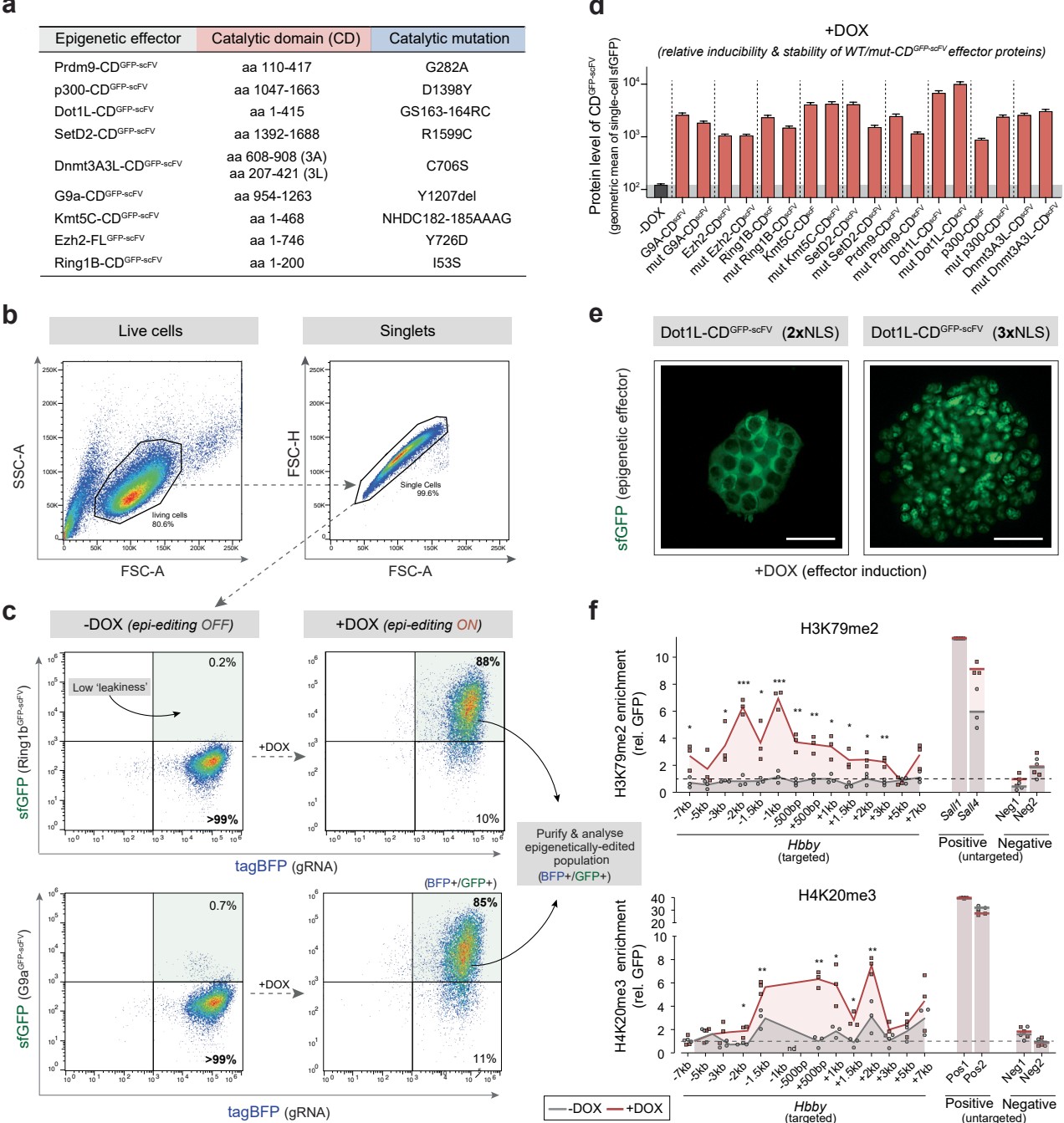

**Extended Data Fig. 1 | An optimised toolkit for precision & dynamic chromatin state perturbations.** (**a**) Table detailing the catalytic domains (CD) used as epigenetic 'effectors' in this study, and the precise point-mutant controls to specifically disrupt their catalytic activity. Each CD effector is tagged with superfolder GFP (sfGFP) and an scFV domain that specifically binds the GCN4 tail of dCas9[GCN4]. (**b-c**) Representative flow cytometry dot plots showing (**b**) the initial filtering and gating strategy, and (**c**) DOX-dependent induction of the epigenetic editing system, shown in upper panels for Ring1b-CD[scFV] (H2AK119ub) and lower panels for G9a-CD[scFV] (H3K9me2). The enhanced gRNA scaffold is constitutively expressed and marked by tagBFP (x-axis). dCas9[GCN4] and each CD[scFV] effector is activated by +DOX, leading to nuclear GFP signal (y-axis) and epigenetic editing. Note GFP signal confirms CD[scFV] or mut-CD[scFV] stability, enables dose-dependent responses to be ascertained, and is used to flow sort pure populations of cells that have appropriately activated the editing system

(GFP+). Note lack of GFP signal in -DOX conditions is consistent with minimal 'leaky' activity. (**d**) Protein levels of induced WT- and mut- CD[scFV] epigenetic effectors, confirming their comparable stability and relative expression level upon DOX induction relative to uninduced (-DOX). Shown is the geometric mean with 95% CI of individual cells (n = 500). (**e**) Representative GFP fluorescence image showing that CD[scFV] effectors often required additional (>2) nuclear localization sequences (NLS) for nuclear accumulation and efficient epigenetic editing. Scalebar=50μm. (**f**) High resolution enrichment of H3K79me2 (upper) and H4K20me3 (lower) across the entire *Hbby* locus after epigenetic editing, targeted with three gRNAs. Enrichment at positive control endogenous loci and negative control (untargeted) loci is shown. Error bars represent S.D. of three independent experiments. *P-values* are calculated by one-tailed unpaired t-test. *P < 0.05 **P < 0.01, ***P < 0.001.*

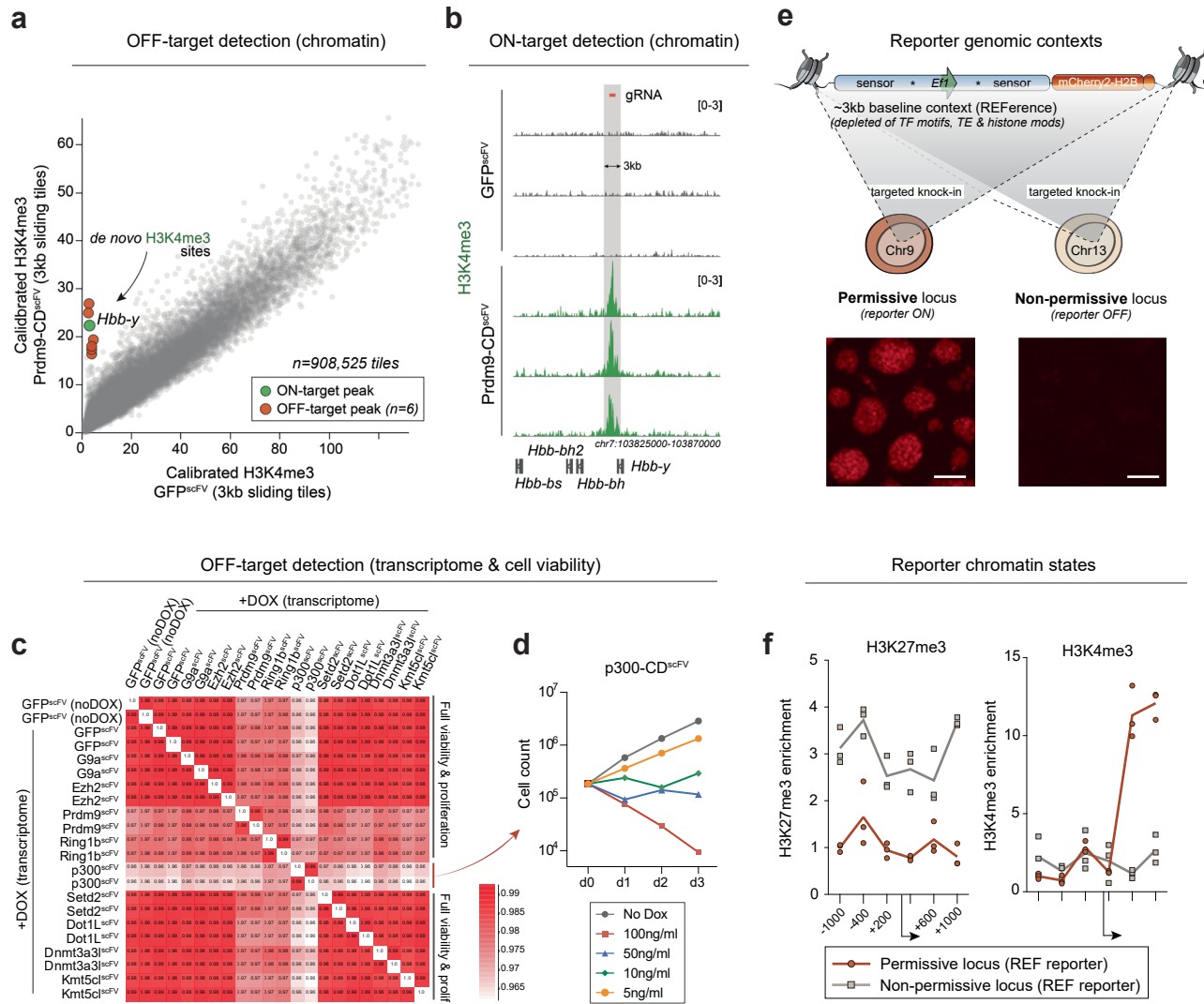

**Extended Data Fig. 2 | Minimal OFF-targeting from epigenetic editing & reporter (epi)genomic features. (a)** Dot plot of genome-wide H3K4me3 enrichment across sliding 3 kb tiles by calibrated Cut&Run. Shown is genome-wide H3K4me3 upon epigenetic editing at the *Hbby* locus with Prdm9-CD^scFV relative to control ESC with GFP^scFV, demonstrating the vast majority of the genome does not acquire H3K4me3 peaks (minimal OFF-targeting). **(b)** Genome tracks showing ON-target enrichment of programmed H3K4me3 across the *Hbby* locus by Prdm9-CD^scFV mediated epigenetic editing. **(c)** Correlation matrix of replicate transcriptomes (RNA-seq) following induction of the indicated epigenetic editing system with DOX. We routinely observed high correlation

(>0.98) between global gene expression, with few OFF-target genes mis-expressed, indicative of preferential ON-target activity. The exception is p300^scFV, and we therefore reduced the DOX concentration to mitigate indirect effects. **(d)** ESC proliferation following a titration of p300-CD^scFV induction levels with DOX. **(e)** Schematic and fluorescent images of ESC carrying the reference (REF) reporter knocked-in to distinct genomic locations; a permissive locus for transcription (left) and a non-permissive locus (right). Images were captured in two independent experiments with similar results. **(f)** Quantification of baseline H3K4me3 and H3K27me3 at identical reference reporters located within the each genomic context.

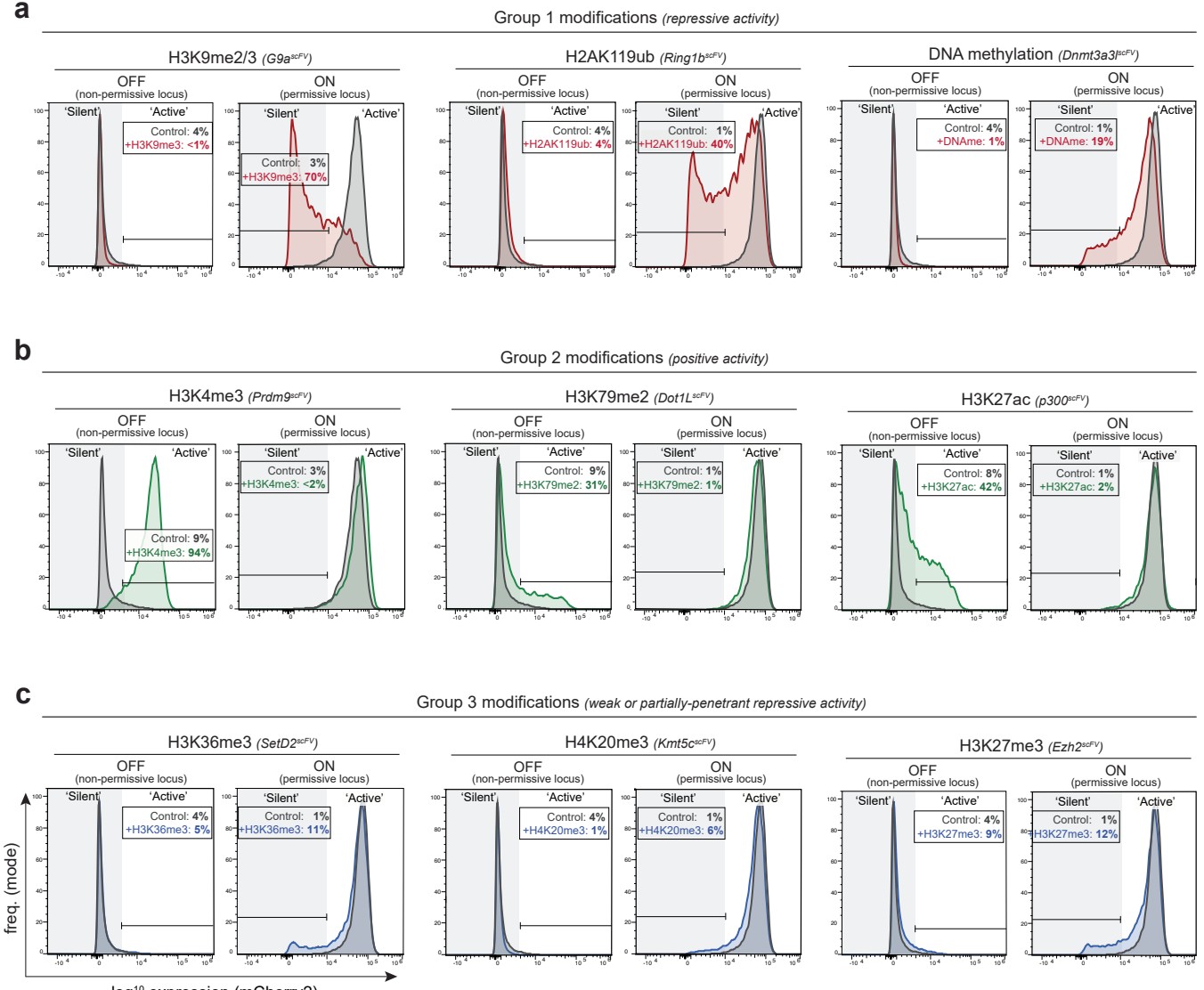

**Extended Data Fig. 3 | Transcriptional impact of programmed chromatin marks at active & inactive loci. (a-c)** Representative flow cytometry histograms of reporter gene expression following *de novo* programming of the indicated chromatin modification. For each modification, the transcriptional effect is shown from a inactive location (initial expression OFF; see left panels) and on an identical promoter in a permissive location (initial expression ON; see right panels). The percentage of cells that acquire a new expression state following precision chromatin editing in each context is indicated, along with control (GFP[scFV]) targeting. Based on reproducible transcriptional responses, we grouped chromatin modifications into functional cohorts whereby **(a)** deposition promotes significant gene repression amongst a major fraction of cells, from an active genomic location **(b)** deposition facilitates significant gene activation amongst a major fraction of cells, from a repressed location, and **(c)** *de novo* targeting has a subtle or highly partially-penetrant repressive effect.

**Extended Data Fig. 4 | See next page for caption.**

**Extended Data Fig. 4 | Temporal dynamics and dose-dependent responses to epigenetic editing.** (**a**) Dot plots showing log expression of the reference reporter in each cell following targeted epigenetic editing with the indicated chromatin modifications. Shown is the transcriptional response at day 2 (d2) and day 7 (d7) after programming each mark with its cognate CD$^{scFV}$ effector relative to control targeting of the GFP$^{scFV}$ effector. *N = 250* cells. Reading was performed in four independent experiments. (**b**) Promoter accessibility at the permissive reporter locus measured by ATAC-seq. Shown is the genome view of promoter accessibility following *de novo* programming of the indicated chromatin modification. (**c**) Dose-dependent transcriptional responses to the indicated chromatin modification effectors. A single population of +DOX cells was stratified based on the level of induced CD$^{scFV}$ expression, as determined by GFP. Shown is the transcriptional response of the reporter, which is directly correlated with the amount of epigenetic editing activity in the cell. Representative dose-dependent responses are displayed as boxplots of single-cell expression levels following programming of H3K4me3, H3K9me2/3, H2AK119ub and DNA methylation. Lines indicate median values and box 25th and 75th percentiles. Whiskers indicate 10$^{th}$ and 90$^{th}$ percentile.

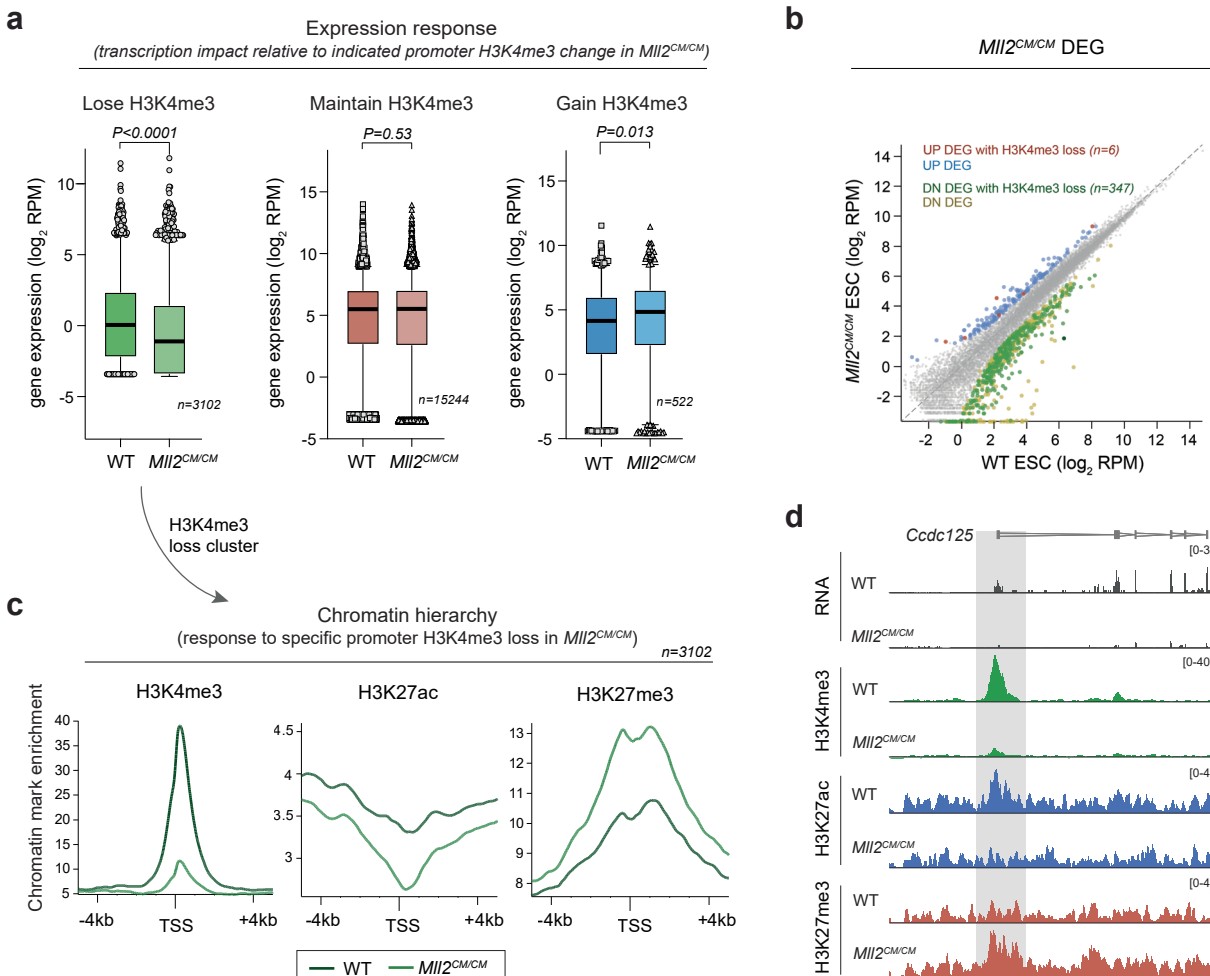

**Extended Data Fig. 5 | Analysis of *Mll2^{CM/CM}* ESC that specifically lack H3K4me3 methylase activity. (a)** Boxplots showing log expression change of genesets stratified according to their promoter H3K4me3 status in *Mll2^{CM/CM}* ESC. Specifically, genes that lose H3K4me3 are significantly downregulated whilst those that maintain H3K4me3 remain unaltered ($N = 3102$, left; $N = 15244$, center; $N = 522$, right). Lines indicate median values and box 25th and 75th percentiles. Whiskers indicate 10th and 90th percentile. **b)** Scatter plot showing all significant differentially expressed genes (DEG) in *Mll2^{CM/CM}* ESC. DEGs are highlighted by being up- or down- regulated and whether they lose promoter H3K4me3 in *Mll2^{CM/CM}* ESC relative to WT ESC. **(c)** Metaplots of H3K4me3, H3K27ac and H3K27me3 enrichment in WT and *Mll2^{CM/CM}* ESC, over MLL2-dependent promoters (TSS) that lose H3K4me3. Promoter H3K4me3 depletion is linked with parallel depletion of H3K27ac and gain of H3K27me3, indicating a chromatin hierarchy. **(d)** Representative genome view showing expression (RNA), and changes in chromatin marks H3K4me3, H3K27ac, and H3K27me3 upon specific loss of H3K4me3 in *Mll2^{CM/CM}* ESC.

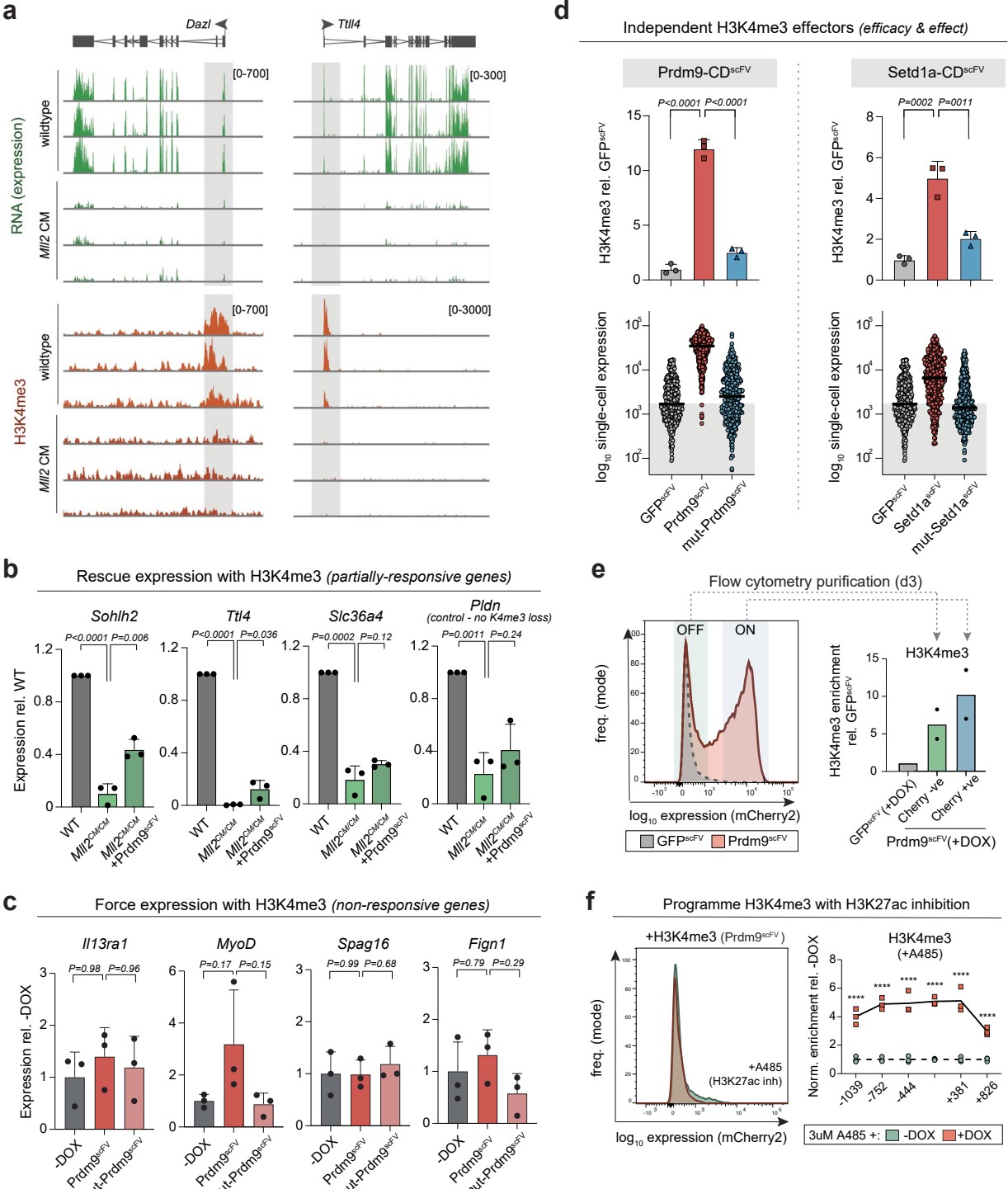

**Extended Data Fig. 6 | See next page for caption.**

**Extended Data Fig. 6 | Programming H3K4me3 activates gene expression via H3K27ac.** (**a**) Genome view of replicate assays showing genes that specifically lose promoter H3K4me3 in *Mll2*<sup>CM/CM</sup> ESC (red), which is linked with strong expression downregulation (green). (**b**) qRT-PCR showing re-targeting H3K4me3 back to endogenous promoters that have lost H3K4me3 in *Mll2*<sup>CM/CM</sup> ESC partially rescues their expression level (see also Fig. 3c). The control *Pldn* gene exhibits no initial loss of H3K4me3 (indirectly affected), and accordingly was not rescued by deposition of further H3K4me3. Bar plots show the mean of n = 3 biologically independent experiments. Error bars represent S.D. Significance of rescue is calculated by two-tailed unpaired t-test. (**c**) Silent endogenous genes targeted for H3K4me3 epigenetic editing that do not exhibit significant transcriptional responses. Bar plots show the mean of *N = 3* biologically independent experiments. Error bars represent S.D. Significance by one-way ANOVA with Tukey's multiple test correction. (**d**) Comparison of two independent H3K4me3 effectors for epigenetic editing (Prdm9-CD<sup>scFV</sup> and Setd1a-CD<sup>scFV</sup>). Upper: CUT&RUN-qPCR showing the level of H3K4me3 deposited at the OFF reporter promoter by each effector and their respective catalytic-mutant controls. Note Prdm9-CD<sup>scFV</sup> deposits significantly higher levels of H3K4me3 than Setd1a-CD<sup>scFV</sup>.

Lower: transcriptional impact of H3K4me3 programming in single cells to each effector reveals a dose-dependent response. Bar plots show the mean of *N = 3* biologically independent experiments. Error bars represent S.D. P-values are calculated by one-way ANOVA with Tukey's multiple test correction. (**e**) Flow cytometry plot at day 3 of Prdm9<sup>scFV</sup> induction, showing ~half the population have initiated a transcriptional response (activation). Active (ON) and inactive (OFF) populations were purified and the level of deposited H3K4me3 assayed by CUT&RUN-qPCR. Whilst all cells are enriched with H3K4me3, those with the higher levels are active, indicating a threshold level of H3K4me3 is necessary to trigger transcriptional activation. Bar plots show the mean of *N = 3* biologically independent experiments. Error bars represent S.D. (**f**) Representative flow cytometry histogram showing that programming H3K4me3 no longer activates expression in the presence of an acetylation inhibitor (A485) - compare with short-term induction plot above with no A485. Shown right is CUT&RUN-qPCR confirming H3K4me3 is programmed in the presence of A485 but cannot elicit downstream effects on transcription. Significance calculated by one-tailed unpaired t-test. *P < 0.05 **P < 0.01, ***P < 0.001.

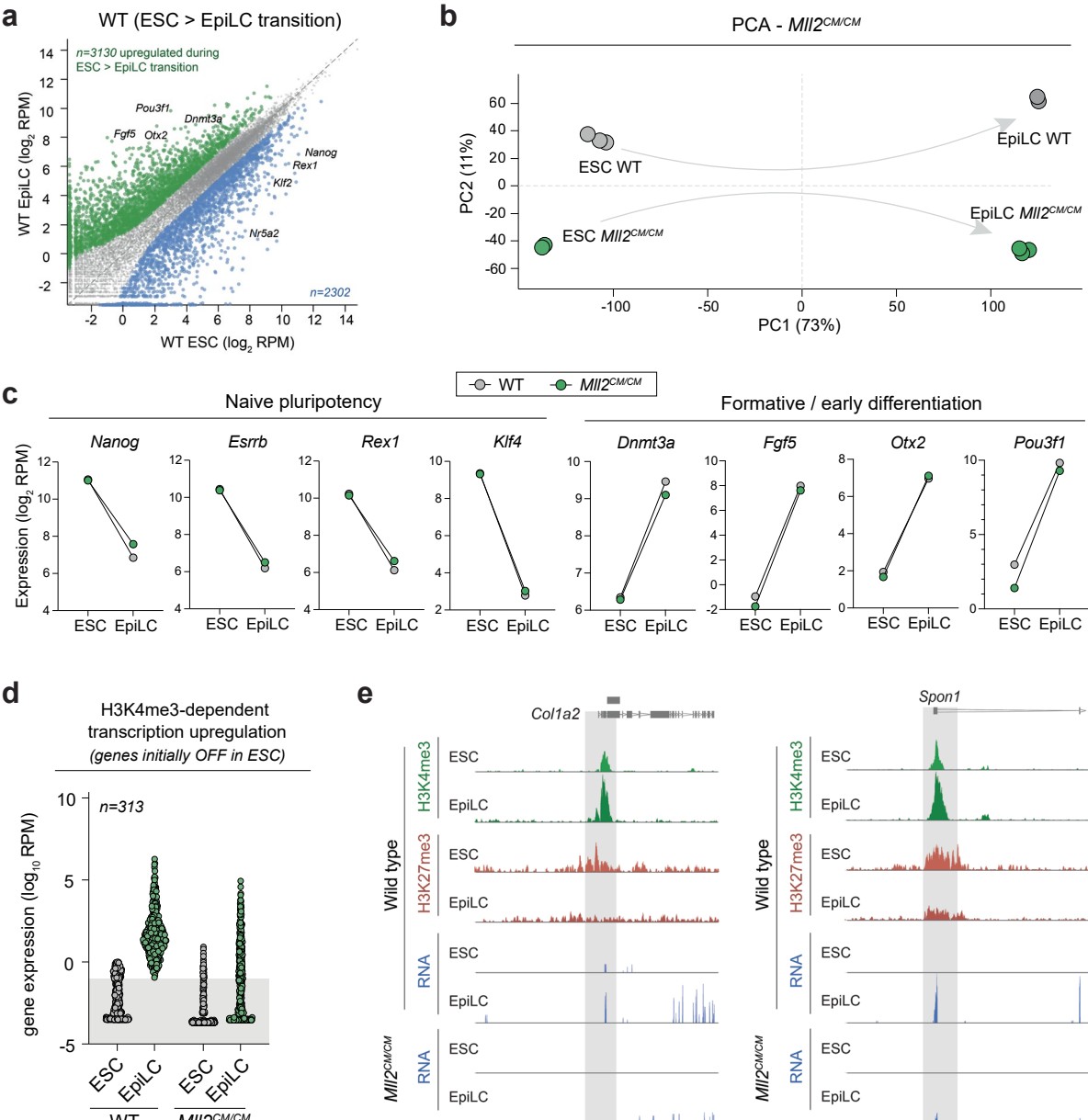

**Extended Data Fig. 7 | Role of H3K4me3 in *de novo* gene activation during cell fate transition.** (**a**) Scatter plot showing expression of genes in naive ESC (x-axis) and formative EpiLC (y-axis) in WT cells. This identified 3130 genes that exhibit significant transcriptional activation during cell fate transition to EpiLC under normal conditions (shown in green). (**b**) Principal component analysis (PCA) of all expressed genes (RPM > 1) in WT and *Mll2^CM/CM* ESC, and upon transition to EpiLC. (**c**) Expression of representative marker genes showing removal of H3K4me3 by *Mll2^CM/CM* does not impact the expression of pluripotency and formative (early differentiation) genes. This indicates that *Mll2^CM/CM* ESC

are fully competent to generate EpiLC, and that any expression changes are not indicative of impaired cell fate commitment. N = three independent experiments (**d**) Expression of the geneset that requires H3K4me3 for *de novo* activation in EpiLC. Shown are genes that are silent in ESC (RPM < 0.1) but fail to fully initiate expression (DEG) in *Mll2^CM/CM* EpiLC that lack H3K4me3, despite normal cell fate transition. Each datapoint represents a single gene (*N = 313*) (**e**) Representative genome view plots of genes that are normally activated in WT EpiLC but fail to initiate expression in *Mll2^CM/CM* EpiLC. These genes normally gain H3K4me3 and lose H3K27me in EpiLC during this transition.

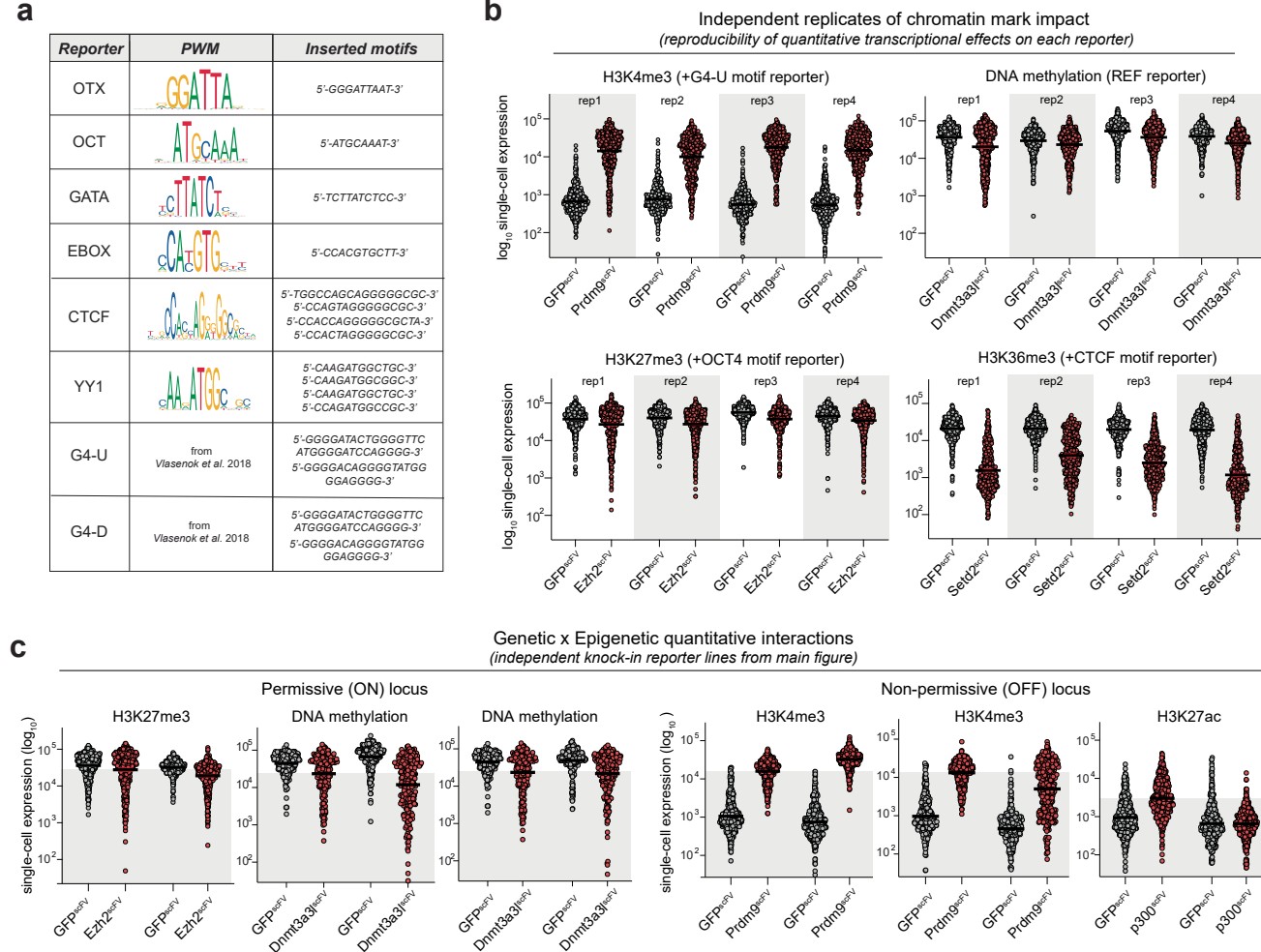

**Extended Data Fig. 8 | Reproducible genetic x epigenetic functional interactions.** (**a**) Table illustrating the position weight matrix for each motif deployed across the reporter series, along with the actual inserted motif(s). (**b**) Dot plots showing $\log_{10}$ expression in single cells (n = 500); bars denote the geometric mean in the population. Displayed within each plot are four independent replicates of programming the same specific epigenetic modification to the same specific reporter, relative to control GFP[scFV]. Different marks and reporter combinations are selected to illustrate the reproducibility of both subtle and major quantitative effects elicited by epigenetic editing.

(**c**) Examples of functional interplay between the quantitative impact of a programmed modification and the presence of an underlying TF motif in the reporter. For example, H3K27ac-mediated activation is attenuated in the context of YY1 motifs, H3K4me3 activation is strengthened in the presence of OCT4 motifs, whilst OTX motifs may enhance DNA methylation mediated repression, albeit this effect was not reproducible across all independent clones (representative samples shown). Dot plots show $\log_{10}$ expression in single cells (N = 500); bars denote the geometric mean in the population.

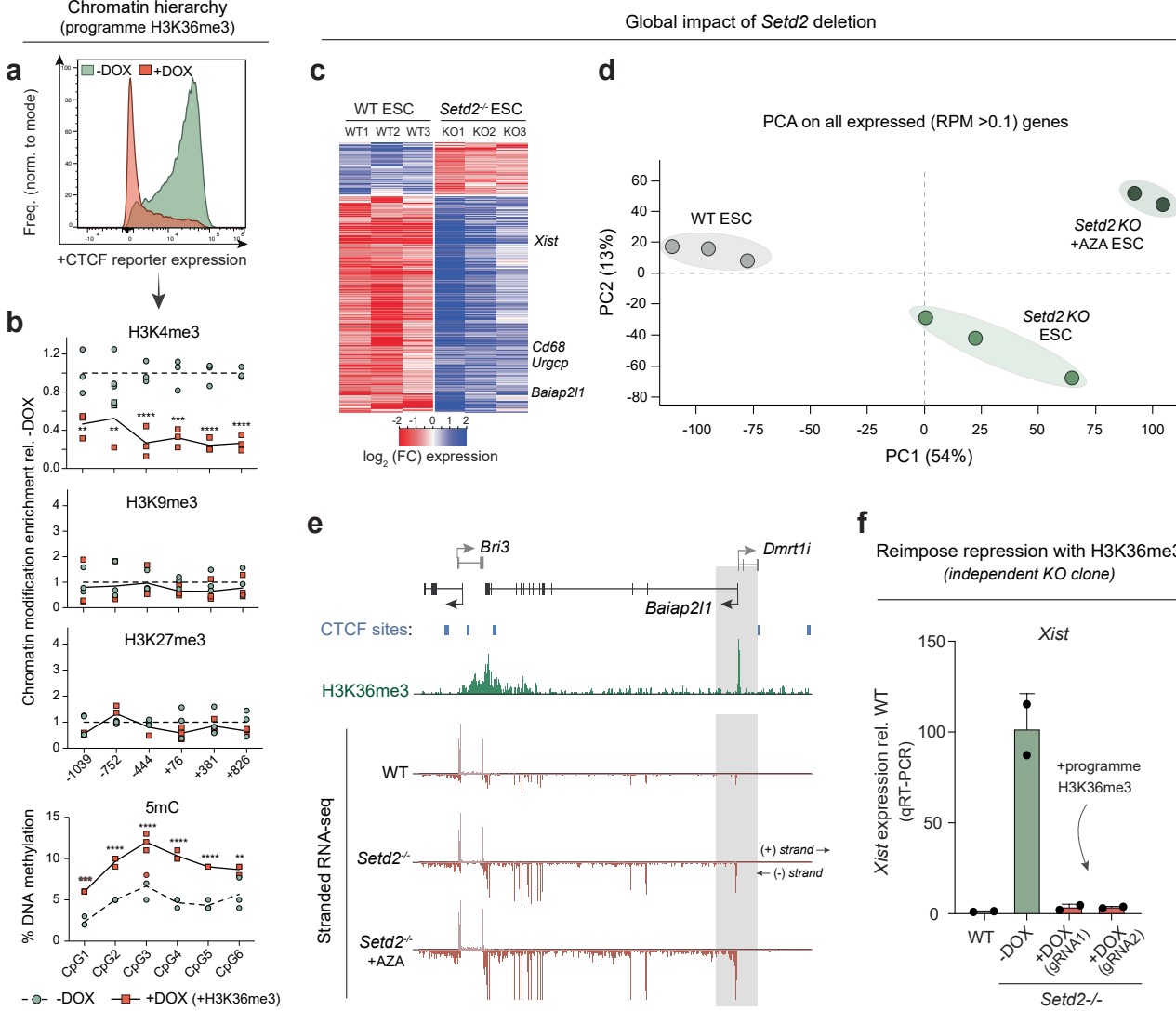

**Extended Data Fig. 9 | H3K36me3 interacts with *cis* CTCF motifs to induce silencing.** (**a**) Histogram showing the distribution of expression of the +CTCF motif reporter in control (-DOX) ESC and upon targeted installation of H3K36me3 at the promoter by Setd2-CD$^{scFV}$ (+DOX). (**b**) Hierarchy of chromatin modification changes following programming of H3K36me3 to the +CTCF reporter, assayed by CUT&RUN-qPCR or bisulfite pyrosequencing. Specific H3K36me3 deposition evicts H3K4me3 and promotes DNA methylation but has no downstream impact on H3K9me3 and H3K27me3. $N = 3$ independent experiments with significance calculated by two-tailed unpaired t-test. *$P < 0.05$ **$P < 0.01$, ***$P < 0.001$. (**c**) Heatmap showing all differentially expressed genes (DEG; $p(adj)<0.05$ and FC > 2)) in replicate *Setd2* knockout (KO) lines. The majority are upregulated, including genes marked by promoter H3K36me3

such as *Xist, Cd68, Urgcp* and *Baiap2l1*. (**d**) Principal component analysis (PCA) of global transcriptomes from WT, *Setd2* knockout or *Setd2* knockout + 5-Aza-Deoxycytidine (AZA: DNA methylation inhibitor) ESC. (**e**) Representative genome view of a gene (*Baiap2l1*) marked by promoter H3K36me3 in ESC, that is significantly upregulated following global loss of H3K36me3 in *Setd2* knockout cells. (**f**) qRT-PCR of de-repressed *Xist* expression in *Setd2* knockout ESC (-DOX), and after induction of epigenetic editing to install H3K36me3 specifically back to the *Xist* promoter (+DOX). Programming H3K36me3 to *Xist* promoter leads to almost complete re-imposition of silencing. $N =$ three biologically independent experiments. Error bars represent S.D. Significance was calculated by two-tailed unpaired t-test.

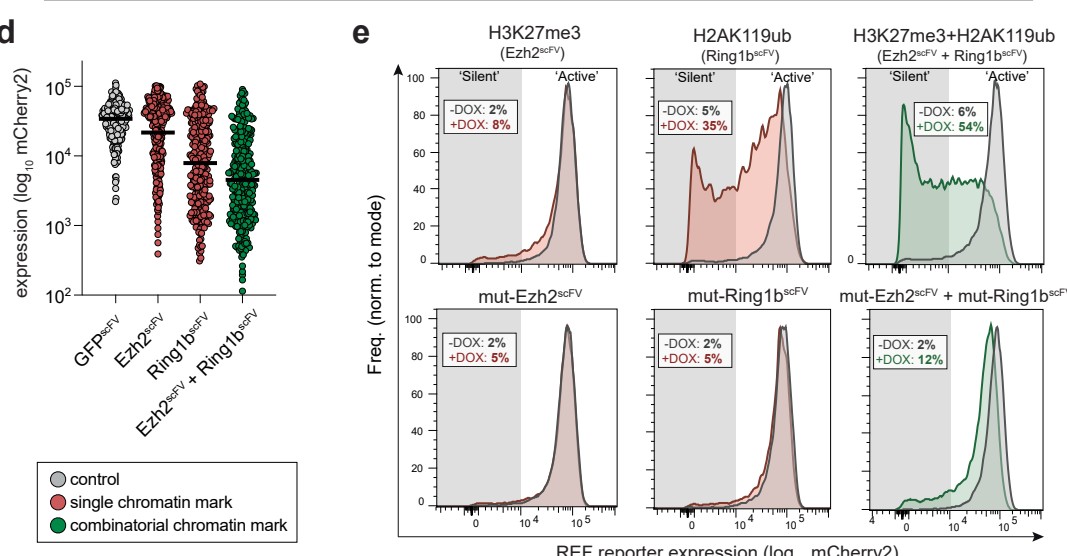

**Extended Data Fig. 10 | Combinatorial chromatin modifications enhance penetrance of single-cell silencing. a**) Dot plot showing log₁₀ single-cell expression (N = 500) upon specific programming of DNA methylation, H3K9me2/3 or both modifications together relative to control. Reading was performed for three independent experiments. (**b**) DNA methylation pyrosequencing confirming that treatment with 1 µM 5-azacytdine (AZA) impairs DNA methylation deposition at the reporter. (**c**) Fraction of cellular population that is in either a 'off', 'low' or 'high' expression state following epigenetic editing

+/− AZA. (**d**) Dot plot showing log₁₀ single-cell expression (N = 500) upon specific programming of H3K27me3, H2AK119ub or both polycomb modifications together relative to control. Reading was performed for three independent experiments. (**e**) Representative and independent flow cytometry plots showing the distribution of gene expression across the population upon single- or combinatorial- polycomb targeting (upper), or with catalytic mutant controls (below). Note that both polycomb marks together increase the penetrance of 'full' silencing.

# Reporting Summary

## Statistics

For all statistical analyses, confirm that the following items are present in the figure legend, table legend, main text, or Methods section.

| n/a | Confirmed | |
|---|---|---|
| ☐ | ☒ | The exact sample size (*n*) for each experimental group/condition, given as a discrete number and unit of measurement |
| ☐ | ☒ | A statement on whether measurements were taken from distinct samples or whether the same sample was measured repeatedly |
| ☐ | ☒ | The statistical test(s) used AND whether they are one- or two-sided *Only common tests should be described solely by name; describe more complex techniques in the Methods section.* |
| ☒ | ☐ | A description of all covariates tested |
| ☐ | ☒ | A description of any assumptions or corrections, such as tests of normality and adjustment for multiple comparisons |
| ☐ | ☒ | A full description of the statistical parameters including central tendency (e.g. means) or other basic estimates (e.g. regression coefficient) AND variation (e.g. standard deviation) or associated estimates of uncertainty (e.g. confidence intervals) |
| ☐ | ☒ | For null hypothesis testing, the test statistic (e.g. *F*, *t*, *r*) with confidence intervals, effect sizes, degrees of freedom and *P* value noted *Give P values as exact values whenever suitable.* |
| ☒ | ☐ | For Bayesian analysis, information on the choice of priors and Markov chain Monte Carlo settings |
| ☒ | ☐ | For hierarchical and complex designs, identification of the appropriate level for tests and full reporting of outcomes |
| ☒ | ☐ | Estimates of effect sizes (e.g. Cohen's *d*, Pearson's *r*), indicating how they were calculated |

*Our web collection on statistics for biologists contains articles on many of the points above.*

## Software and code

Policy information about availability of computer code

Data collection | Flow cytometry data was collected using the Attune NxT Flow Cytometer (Thermo Fisher Scientific). Sequencing data was collected using the Nextseq Illumina sequencing system. qPCR data was collected using QuantStudio 5 (Applied Biosystems) thermal cycler. Pyrosequencing was run on PyroMark Q24 advanced pyrosequencer (Qiagen). Cell images were taken using the Leica Application Suite X (v3.5.7.23225). No custom software was used in this study.

Data analysis | Data analysis was performed using Graphpad Prism version 8.4.3 graphical software, FlowJo (v10.5.3), PyroMark Q24 Advanced 3.0 software, the Galaxy workflow interface maintained by EMBL Genome Biology Computational Support (v 23.0.6.dev0 ), R statistical software (v3.6.2) using Bioconductor packages, and with Seqmonk (v1.47.0) mapped sequence data anaylser.

RNAseq: Raw Fastq reads were trimmed to remove adaptors with TrimGalore (0.4.3.1, -phred33– quality 20–stringency 1 -e 0.1–length 20), quality checked and aligned to the mouse mm10 (GRCm38) genome using RNA Star (2.5.2b-0, default parameters except for–outFilterMultimapNmax 1000). Analysis of the mapped sequences was performed using Seqmonk software (Babraham bioinformatics, v1.47.0) to generate log2 reads per million (RPM) or gene length-adjusted (reads per kilobase million, RPKM) gene expression values. Differentially expressed genes (DEG) were determined using the DESeq2 package (v.1.24.0), inputting raw strand-specific mapping counts and applying a multiple-testing adjusted (FDR) P< 0.05 significance threshold, and log2 fold-change filter where indicated.

CUT&RUNseq: Raw Fastq sequences were trimmed to remove adaptors with TrimGalore (v0.4.3.1, -phred33 --quality 20 --stringency 1 -e 0.1 --length 20), quality checked and aligned to the mouse mm10 genome with the inserted mCherry reporter using Bowtie2 (v2.3.4.2, -I 50 -X 800 --fr -N 0 -L 22 -i 'S,1,1.15' --n-ceil 'L,0,0.15' --dpad 15 --gbar 4 --end-to-end -- score-min 'L,-0.6,-0.6'). Analysis of the mapped sequences was performed using Seqmonk software (Babraham bioinformatics, v1.47.0) by enrichment quantification of the normalised reads. To identify

promoters with H3K4me3 change in Mll2CM/CM, a 1kb window centered on the TSS was quantified amongst replicates and a normalised log fold-change (FC) filter applied between samples. Metaplots over genomic features were constructed by quantifying 100bp bins centered on the features of interest and normalised cumulative enrichments plotted.

ATACseq: Following sequencing, raw reads were first trimmed with TrimGalore (v0.4.3.1, reads > 20 bp and quality > 30) and then quality checked with FastQC (v0.72). The resulting reads were aligned to custom mouse mm10 genome containing the reporter using Bowtie2 (v2.3.4.3, paired-end settings, fragment size 0-1,000, --fr, allow mate dovetailing). Aligned sequences were then analysed with seqmonk (Babraham bioinformatics, v1.47.0) by performing enrichment quantification of the normalised reads.

For manuscripts utilizing custom algorithms or software that are central to the research but not yet described in published literature, software must be made available to editors and reviewers. We strongly encourage code deposition in a community repository (e.g. GitHub). See the Nature Portfolio guidelines for submitting code & software for further information.

## Data

Policy information about availability of data

All manuscripts must include a data availability statement. This statement should provide the following information, where applicable:
- Accession codes, unique identifiers, or web links for publicly available datasets
- A description of any restrictions on data availability
- For clinical datasets or third party data, please ensure that the statement adheres to our policy

Manuscript includes a data availability statement. All data derived from next generation sequencing assays have been deposited in the publically available ArrayExpress database under the accession codes E-MTAB-12101, E-MTAB-13466, E-MTAB-13467, E-MTAB-13468.
https://www.ebi.ac.uk/biostudies/arrayexpress/studies/E-MTAB-12101
https://www.ebi.ac.uk/biostudies/arrayexpress/studies/E-MTAB-13466
https://www.ebi.ac.uk/biostudies/arrayexpress/studies/E-MTAB-13467
https://www.ebi.ac.uk/biostudies/arrayexpress/studies/E-MTAB-13468
Additionally, previously published ChiP-seq data from Nora et al. 2017 is used in this study: GSE98671, https://www.ncbi.nlm.nih.gov/geo/query/acc.cgi?acc=GSE98671
All data is publicly available.

## Human research participants

Policy information about studies involving human research participants and Sex and Gender in Research.

| Reporting on sex and gender | N/A |
|---|---|
| Population characteristics | N/A |
| Recruitment | N/A |
| Ethics oversight | N/A |

Note that full information on the approval of the study protocol must also be provided in the manuscript.

# Field-specific reporting

Please select the one below that is the best fit for your research. If you are not sure, read the appropriate sections before making your selection.

☒ Life sciences          ☐ Behavioural & social sciences          ☐ Ecological, evolutionary & environmental sciences

For a reference copy of the document with all sections, see nature.com/documents/nr-reporting-summary-flat.pdf

# Life sciences study design

All studies must disclose on these points even when the disclosure is negative.

| Sample size | Information on sample size is provided within each figure legend. Sample sizes were based on prior research experience for similar assays rather than power analysis. Biological replicates are defined as measurements of biologically distinct samples (e.g. independently derived clonal reporter lines each separately transected with epigenetic editing machinery; or independently (and freshly) derived genetically-modified ESC lines from blastocysts). Technical replicates are defined as repeated measurements of the same sample that show independent measures of the noise associated with the equipment and the protocols. |
|---|---|
| Data exclusions | To generate the heat maps shown in figures 4C and 6A, typically normalised geometric mean values coming from four technical replicates of the experiments were averaged and log2 transformed. Log2 fold-change values were plotted in R. In rare cases, outliers identified as extremely different values relative to all other values within the dataset were excluded. Data exclusion was also performed whenever cross-contamination between different samples was suspected.<br>No data was excluded from RNAseq, CUT&RUNseq, CUT&RUN-qPCR and bisulfite pyrosequencing experiments. |

| Replication | All reported findings are reliably reproducible. The impact of epigenetic marks on transcription in different genomic contexts was validated using different reporter clones across at least three independent experiments and by performing repeated measurements of the same samples over time. Reproducibility between independent RNAseq and CUT&RUNseq samples was assessed on binned and library-normalised files using multiple clustering approaches including PCA, correlation assessment, and unsupervised hierarchal clustering in R and Seqmonk software, with good reproducibility observed. All experiments with quantification data were repeated at least three times in biologically independent experiments (the only exception being epigenetic mark deposition assessment for catalytic mutant effectors in fig. 2c-k for which in some cases n = 2 biological replicates). All replications were successful, with coherent results. |
|---|---|
| Randomization | No randomization of data was performed as the study does not involve a clinical trial, human subjects or mice. All cell lines within each experimental paradigm/experiment were cultured concurrently to minimise batch variability which is not the result of treatment or genetic background. |
| Blinding | The investigators were not blinded during experiments because no subjectivity in assessment of experimental results was possible in our study. Indeed all data collection and quantification was performed via dedicated and quantitative instruments (Attune NxT Flow Cytometer, qPCR thermal cycler etc..) therefore avoiding biased outcomes. Additionally all results were checked and interpreted by two different individuals. |

# Reporting for specific materials, systems and methods

We require information from authors about some types of materials, experimental systems and methods used in many studies. Here, indicate whether each material, system or method listed is relevant to your study. If you are not sure if a list item applies to your research, read the appropriate section before selecting a response.

### Materials & experimental systems

| n/a | Involved in the study |
|---|---|
| ☐ | ☒ Antibodies |
| ☐ | ☒ Eukaryotic cell lines |
| ☒ | ☐ Palaeontology and archaeology |
| ☒ | ☐ Animals and other organisms |
| ☒ | ☐ Clinical data |
| ☒ | ☐ Dual use research of concern |

### Methods

| n/a | Involved in the study |
|---|---|
| ☐ | ☒ ChIP-seq |
| ☐ | ☒ Flow cytometry |
| ☒ | ☐ MRI-based neuroimaging |

## Antibodies

| Antibodies used | Rabbit anti-H3K4me3 (Diagenode Cat#C15410003), Rabbit anti-H3K27me3 (Millipore Cat#07-449), Rabbit anti-H3K9me3 (Abcam Cat#ab8898), Rabbit anti-H3K9me2 (Active Motif Cat#39041), Rabbit anti-H2Aub (Lys119) (CST Cat#8240), Rabbit anti-H3K36me3 (Diagenode Cat#C15410192), Rabbit anti-H3K36me3 (Active Motif Cat#61101), Rabbit anti-H3K27ac (Active Motif Cat#39133), Rabbit anti-H3K79me2 (Abcam Cat#ab3594), Rabbit anti-H4K20me3 (Abcam, Cat#ab9053) |
|---|---|
| Validation | All used antibodies are commercially available and have been validated by the manufacturer. Validations and detail product information are available on the manufacturer's websites.<br><br>Rabbit anti-H3K4me3 (Diagenode Cat#C15410003): the manufacturer has validated the antibody specificity by ChiP using the iDeal ChiP-seq kit and has determined the specificity by dot blot performed with peptides containing other histone modifications and the unmodified H3K4. The manufacturer states: "Figure 5A shows a high specificity of the antibody for the modification of interest". https://www.diagenode.com/en/p/h3k4me3-polyclonal-antibody-premium-50-ug-50-ul<br><br>Rabbit anti-H3K27me3 (Millipore Cat#07-449): the manufacturer has validated the antibody specificity by dot blot. The manufacturer states: "1 μg/mL of this antibody detected Trimethyl-Histone H3 (Lys27), but not unmethylated Histone H3 (Lys27) or other peptides corresponding to modified histones in an Absurance™ Histone H3 Antibody Specificity Array (Cat. No. 16-667) and in an Absurance™ Histone H2A, H2B, H4 Antibody Specificity Array (Cat. No. 16-665)". https://www.merckmillipore.com/IT/it/product/Anti-trimethyl-Histone-H3-Lys27-Antibody,MM_NF-07-449<br><br>Rabbit anti-H3K9me3 (Abcam Cat#ab8898): the manufacturer has validated the antibody specificity by western blot. The manufacturer states: "Histone H3 (tri methyl K9) antibody (ab8898) is specific for Histone H3 tri methyl Lysine 9. Shows slight cross-reactivity with tri methyl K27, which shares a similar epitope (please see Western blot image). Does not react with mono or di methylated K9". https://doc.abcam.com/datasheets/histone-h3-tri-methyl-k9-antibody-chip-grade-ab8898.pdf<br><br>Rabbit anti-H3K9me2 (Active Motif Cat#39041): the manufacturer has validated the antibody specificity by dot blot and peptide array analysis. https://www.activemotif.com/catalog/details/39239/histone-h3-dimethyl-lys9-antibody-pab<br><br>Rabbit anti-H2Aub (Lys119) (CST Cat#8240): the manufacturer states: "Ubiquityl-Histone H2A (Lys119) (D27C4) XP® Rabbit mAb recognizes endogenous levels of histone H2A protein only when ubiquitinated at Lys119. The antibody does not cross-react with other ubiquitinated proteins or free ubiquitin". https://www.cellsignal.com/products/primary-antibodies/ubiquityl-histone-h2a-lys119-d27c4-xp-rabbit-mab/8240 |

Rabbit anti-H3K36me3 (Diagenode Cat#C15410192): the manufacturer has validated the antibody specificity by dot blot and by peptide array analysis. The manufacturer states: "Figure 5A shows a high specificity of the antibody for the modification of interest. The peptide array analysis shows a slight cross reaction with H4K20me3 that was not observed in dot blot".
https://www.diagenode.com/en/p/h3k36me3-polyclonal-antibody-premium-50-mg

Rabbit anti-H3K36me3 (Active Motif Cat#61101): Dot blot analysis was used to confirm the specificity of Histone H3K36me3 antibody for trimethyl-lysine 36 of histone H3.
https://www.activemotif.com/catalog/details/61101/histone-h3-trimethyl-lys36-antibody-pab

Rabbit anti-H3K27ac (Active Motif Cat#39133): Dot blot analysis was used to confirm the specificity of Histone H3K27ac antibody.
https://www.activemotif.com/catalog/details/39133/histone-h3-acetyl-lys27-antibody-pab

Rabbit anti-H3K79me2 (Abcam Cat#ab3594): the manufacturer has validated the antibody specificity by western blot and by peptide array analysis. The manufacturer states: "ab3594 detects a 17 kDa band in single lane Western Blot. Peptide inhibition in Western Blot hasn't been processed. Modification specificity is determined by Peptide Array. ab3594 binds strongly to the Histone H3 dii methyl K79. In Peptide Array ab3594 also partially binds to mono methyl K79 and tri methyl K79 peptides".
https://www.abcam.com/en-it/products/primary-antibodies/histone-h3-di-methyl-k79-antibody-chip-grade-ab3594

Rabbit anti-H4K20me3 (Abcam, Cat#ab9053): the manufacturer has validated the antibody specificity by western blot. The manufacturer states: "ab9053 is specific for Histone H4 (tri-methyl K20). This is illustrated in lane 5 where the activity of ab9053 is specifically blocked by the addition of the immunizing peptide (ab17567)".
https://www.abcam.com/en-it/products/primary-antibodies/histone-h4-tri-methyl-k20-antibody-chip-grade-ab9053#all

# Eukaryotic cell lines

Policy information about cell lines and Sex and Gender in Research

| | |
|---|---|
| Cell line source(s) | Wildtype mouse embryonic stem cells (mESCs) were derived freshly (mixed 129/B6, XY) by the EMBL Rome Gene editing and Embryology facility. Catalytic-mutant Mll2 (Mll2CM/CM) and Setd2-/- homozygous ESCs were freshly derived from heterozygous FVB crosses carrying either an Mll2 Y2602A or a Setd2 KO allele. |
| Authentication | mESCs were authenticated by robust expression of pluripotency markers, morphology and by chimera formation and contribution to all embryonic tissues in vivo of the parental line (Carlini et al. 2022). |
| Mycoplasma contamination | mESCs were routinely tested for mycoplasma contamination by an independent commercial service using a highly-sensitive quantitative (q)PCR, with negative result each time. |
| Commonly misidentified lines (See ICLAC register) | N/A |

# ChIP-seq

## Data deposition

☒ Confirm that both raw and final processed data have been deposited in a public database such as GEO.

☒ Confirm that you have deposited or provided access to graph files (e.g. BED files) for the called peaks.

| | |
|---|---|
| Data access links *May remain private before publication.* | accession codes E-MTAB-12101, E-MTAB-13466, E-MTAB-13467, E-MTAB-13468. |
| Files in database submission | fastq |
| Genome browser session (e.g. UCSC) | No longer applicable. |

## Methodology

| | |
|---|---|
| Replicates | 3 replicates in each treatment setting for ChIP-seq and RNA-seq. |
| Sequencing depth | Libraries were sequenced on the Nextseq Illumina sequencing system (paired-end 40 sequencing). |
| Antibodies | Rabbit anti-H3K4me3 (Diagenode Cat#C15410003), Rabbit anti-H3K27me3 (Millipore Cat#07-449), Rabbit anti-H3K27ac (Active Motif Cat#39133) |
| Peak calling parameters | Analysis of the mapped sequences was performed using seqmonk software (Babraham bioinformatics, v1.47.0) by enrichment quantification of the normalised reads. To identify promoters with H3K4me3 change in Mll2 CM/CM, a 1kb window centered on the TSS was quantified amongst replicates and a normalised log fold-change (FC) filter applied between samples. Metaplots over genomic features were constructed by quantifying 100bp bins centered on the features of interest and normalised cumulative enrichments plotted. |
| Data quality | Raw Fastq sequences were trimmed to remove adaptors with TrimGalore (v0.4.3.1, -phred33 --quality 20 --stringency 1 -e 0.1 -- |

| | |
|---|---|
| Data quality | length 20), quality checked and aligned to the mouse mm10 genome with the inserted mCherry reporter using Bowtie2 (v2.3.4.2, -I 50 -X 800 --fr -N 0 -L 22 -i 'S,1,1.15' --n-ceil 'L,0,0.15' --dpad 15 --gbar 4 --end-to-end -- score-min 'L,-0.6,-0.6'). |
| Software | Seqmonk v1.46.0 Babraham bioinformatics https://www.bioinformatics.babraham.ac.uk/projects/seqmonk/<br>DESeq2 Love et al. 2014 https://bioconductor.org/packages/release/bioc/html/DESeq2.html<br>RNA Star Dobin et al. 2013 https://github.com/alexdobin/STAR<br>Trim Galore Krueger F. 2015 http://www.bioinformatics.babraham.ac.uk/projects/trim_galore/<br>FastQC Andrews S. 2010 https://www.bioinformatics.babraham.ac.uk/projects/fastqc/<br>FilterBAM Barnett et al. 2011 https://github.com/hammerlab/filter-bam<br>MarkDuplicates Tim Fennell https://broadinstitute.github.io/picard/<br>Bowtie2 Langmead and Salzberg 2012 https://github.com/BenLangmead/bowtie2 |

# Flow Cytometry

## Plots

Confirm that:

☒ The axis labels state the marker and fluorochrome used (e.g. CD4-FITC).

☒ The axis scales are clearly visible. Include numbers along axes only for bottom left plot of group (a 'group' is an analysis of identical markers).

☒ All plots are contour plots with outliers or pseudocolor plots.

☒ A numerical value for number of cells or percentage (with statistics) is provided.

## Methodology

| | |
|---|---|
| Sample preparation | Cells were washed with PBS and gently dissociated into single-cell suspension using TrypLE, followed by resuspension in FACS buffer comprised of PBS with 1% FBS, and filtered through a 40μm cell strainer (BD, cup-Filcons #340632). A FACS Aria III (Becton Dickinson) or Attune NxT Flow Cytometer (Thermo Fisher Scientific) were used for sorting or analysis, respectively. |
| Instrument | FACS Aria III (Becton Dickinson) or Attune NxT Flow Cytometer (Thermo Fisher Scientific) |
| Software | Flow cytometry data analysis was performed with FlowJo v10.5.3 (Tree Star, Inc.). Statistical analysis of flow cytometry data was performed using appropriate strategies in Prism GraphPad statistical software (v8.4.3). |
| Cell population abundance | For CUT&RUN experiments, from 1×105 to 1×106 cells (depending on the selected antibody) were flow sorted. For RNA extraction, 200.000 cells were flow sorted for each sample. A 4-Way Purity Precision Mode was used. A small portion of each sample was run again after each sort to verify that the intended cell population had been collected. |
| Gating strategy | Forward and side scatter density plots were used to distinguish between live cells and dead cells/debris. A side scatter height (SSC-H) vs side scatter area (SSC-A) plot was then used to exclude doublets. Next, two parameter density plots in which the X axis displays GFP fluorescence and the Y axis displays BFP fluorescence were used to gate for cells expressing all epigenetic editing components (GFP+; BFP+). Gates were pre-determined by running appropriate controls including GFP only and BFP only expressing cells and double negative cells (GFP-;BFP-). Gating strategy is also described in Extended data figure1. |

☒ Tick this box to confirm that a figure exemplifying the gating strategy is provided in the Supplementary Information.

