## [Peer Review File · Nature Genetics]

Peer Review Information

Manuscript Title: Systematic Epigenome Editing Captures the Context-dependent Instructive Function of Chromatin Modifications

Corresponding author name(s): Dr Jamie Hackett

Reviewer Comments & Decisions:

Decision Letter, initial version:

8th Nov 2022

Dear Dr Hackett,

First, please accept my apologies for the delay in returning this decision to you.

Your Article, "Systematic Epigenome Editing Captures the Context-dependent Instructive Function of Chromatin Modifications" has now been seen by 4 referees. Please note that Reviewers #3 and #4 reviewed the paper together and uploaded the same report.

You will see from their comments below that while they find your work of interest, some important points are raised. We are interested in the possibility of publishing your study in Nature Genetics, but would like to consider your response to these concerns in the form of a revised manuscript before we make a final decision on publication.

To guide the scope of the revisions, the editors discuss the referee reports in detail within the team, including with the chief editor, with a view to identifying key priorities that should be addressed in revision and sometimes overruling referee requests that are deemed beyond the scope of the current study. We note that Reviewer #2 considered the novel biology showcased by the tools to be modest. We considered their feedback carefully and while we think that the inclusion of more biology would fortify the paper, our editorial interest lies mainly in the suite of editing tools that you are presenting. As such, you are encouraged to fortify this aspect of the work, but the lack of these data will not necessarily preclude our interest providing that your response to all other points is sufficiently robust. Regarding this, we would like you to address all comments raised by your Reviewers, experimentally where possible, and /or textually where appropriate.

Please do not hesitate to get in touch if you would like to discuss these issues further.

We therefore invite you to revise your manuscript taking into account all reviewer and editor

comments. Please highlight all changes in the manuscript text file. At this stage we will need you to upload a copy of the manuscript in MS Word .docx or similar editable format.

*2) If you have not done so already please begin to revise your manuscript so that it conforms to our Article format instructions, available here.

*3) Include a revised version of any required Reporting Summary:

Please be aware of our guidelines on digital image standards.

[redacted]

We hope to receive your revised manuscript within four to eight weeks. If you cannot send it within this time, please let us know.

Sincerely,

Safia Danovi
Editor
Nature Genetics

Referee expertise:

Referee #1: epigenetics, single cell biology, methods

Referee #2: epigenetics, transcription

Referee #3: epigenomic editing (reviewed with #4)

Referee #4: epigenomic editing (reviewed with #3)

Reviewers' Comments:

Reviewer #1:

Remarks to the Author:

The paper by Policarpi et al is an impressive and interesting show case of a powerful set of epigenetic editing tools, which are applied toward resolving key challenges relating the functional hierarchy among epigenetic modulators and transcription control. While there is a growing body of literature on dCas9-based epigenetic editing, I find the current work is taking several key steps toward a reliable, integrated and potent scheme for resolving mechanisms and causal relationship between the transcriptionally promoting and repressive machineries. Beyond the technical contributions (the system itself), there are several truly important results. In that context I find of particular interest: a) experiments targeting H3k4me3 and positioning it within the feedback loop of transcriptional activation, b) some insightful results regarding polycomb-mediated repression and its complex interactions with other systems (PRC1.6, YY1, possible H2K119ub/H3K27me3 synergism), c) CTCF, H3k36me3 and linked effects.

I would like to stress that in my opinion, this paper is important not just because of any of these results individually – but because it is approaching the problem holistically and set up a system and a conceptual framework that can lead to much better quantitative understanding than the modest and in some cases non-existent models we have today. So even though the paper leaves many open questions, is restricted to work on mESC and is for the most part not expanding its conclusion to a epigenome-wide scale – I think it is of very high interest and quality as is, and support its timely publication.

I have very few comments for improvement and further consideration. The major comment relates to the uniqueness of the pluripotent/mESC state and the need to better acknowledge it. It is clear from

the text that the authors understand very well that mESCs represent a unique epigenetic state, but the text (intro, discussion) can (and should) make it much clearer to readers that the hierarchy of epigenetic factors that is studied here is likely to be specific to this state. There are many mechanisms contributing to epigenetic turnover and plasticity in mES, and while these are not truly understood yet, it is unlikely that the authors results would reproduce as is in differentiating embryonic lineages, homeostatic adult stem cells or other systems of interest. It is truly important to be open about this point and poise the paper as addressing questions of epigenetic function and hierarchy in the pluripotent state. The discussion can include a short analysis of the possible implications to other cellular states. For the epigenetic editing system itself, it is interesting to ask if its efficiency will be similar in less plastic cellular states. If there are preliminary results on this it will be important to include them. But I don't suggest this should delay a revision process.

A point in which some additional analysis may be insightful is around Fig 3H. Some attempt to characterize the H3K4me3 dependent genes is not described. Analysis of different genomic features can be helpful: CpG content on promoters, GC content, distance to nearby CTCF site, distance to nearby TAD boundary, distance to other active promoters. It also make sense to run a simple DNA motif screen on this set.

Another possible genome-wide follow up is relating the observation on YY1 and E-BOX motifs to context specific epigenetic states. Are YY1 motifs appears at lower frequencies in endogenous polycomb domains of mES? For MYC and PRC1.6 binding to E-BOX, are loci (promoters?) with a strong E-box more likely to be repressed in mES and transition to activation in MYC expression lineages? (e.g. hematoendothelial progenitors and primitive erythrocytes?). I would like to stress that I find the mechanistic and reductionist approach of this paper very useful and appropriate. If the above suggestion for follow up are leading to some useful insight it can only strengthen the paper (and if such results are difficult to derive, given the numerous confounders that are affecting such analyses, it will not undermine the relevance of the experiments reported here).

Typo: label in Fig S3A: H4K9me3 -> H3K9me3

Reviewer #2:

Remarks to the Author:

In this manuscript, Policarpi et al. use a catalytically dead Cas9-SunTag construct to target a set of 9 chromatin modifiers to a fluorescent reporter gene as well as selected endogenous loci and then measure the effects on gene expression. The authors make the conclusions that H3K4me3 is instructive for transcriptional activation, that DNA motifs influence the effects of chromatin modifications and that H2AK119ub (mediated by PRC1) and H3K27me3 (mediated by PRC2) function synergistically.

While this work reports a few interesting but albeit unexplained phenomena, overall, I found it to be preliminary, lacking in mechanistic detail and missing a large number of essential control experiments. I also don't see the potential for a broad and sustained impact for the field of genetics and feel this work would be better suited to a journal focused on chromatin biochemistry.

While the authors test multiple modifiers, this approach is not particularly novel as the dCas9-SunTag system has already been used to edit chromatin by a number of groups (For a review of this see:

Pulecio et al., 2018. "CRISPR/Cas9-based engineering of the epigenome" *Cell Stem Cell*). Additionally, the authors have not performed a comprehensive analysis of chromatin modifiers and in some cases their choices of which enzymes to examine seems arbitrary. For instance, they use G9a to implement H3K9me2, but do not examine the activity of Suv39 or SETDB1. Similarly, they primarily use the PR/SET domain of Prdm9 to implement H3K4me3 and perform a few validation experiments with the SET domain of Setd1a, however, they do not examine the well characterized KMT2/MLL class of H3K4 methyltransferases. I feel that extending this study to include all known histone methyltransferase and acetyltransferase domains would make it more broadly relevant, although it would still be best suited as a resource in a chromatin-centric specialty journal.

It is also worth considering that when histone modifications are implemented in cells, their activity is generally restricted by multiple histone "reader" domains present in histone modifiers and/or regulation of the enzymes at the level of catalytic activity. Thus, the system that the authors use is quite artificial and may be prone to artifacts that have no real relevance in vivo. This can be seen by the very broad distribution of H3K4me3 mediated by Prdm9, as this modification is generally observed as narrow peaks associated with gene promoters.

I found the experiments relating to H3K4me3 and transcription to be unconvincing. More problematically, this study lacks many essential control experiments, and therefore, I feel the data does not support the authors' conclusions.

Specific Comments

1. A major flaw of this work is that it fails to adequately address off-target effects caused by overexpression of histone modifier catalytic domains. Throughout the paper the authors use a GFP-scFV or catalytic dead enzyme as a control. In my opinion, these are not appropriate controls because they do not control for off-target effects mediated by overexpression of the active catalytic domains, which could implement their respective modifications at aberrant sites throughout the genome. The authors should perform experiments expressing all of their active modifier machinery (dCas9-SunTag + Catalytic Domain-scFV + Doxycycline) but using non-targeting sgRNAs and/or sgRNAs that target to an inert region of the genome (e.g. Rosa26). It is very clear from the data presented in Figure 1D through 1I and 1K that several of the modifiers used have off-target effects. For instance, in Fig. 1I it is apparent that the Setd2 construct implements H3K36me3 at the off target *Zic4* and *HoxD13* loci. It is also clear from figure 1K that the p300 construct induces massive transcriptional changes. The Ring1b and Prdm9 construct also result in widespread changes in gene expression, although the authors downplay point this in the manuscript.

In addition to performing experiments using non-target or off-target sgRNAs as controls, the authors must quantify genome-wide off-target activity of their constructs by performing normalized ChIP-seq using *Drosophila* chromatin spike in. Throughout the paper the authors use CUT&RUN followed by qPCR for selected loci. This is an atypical approach as ESCs can be easily proliferated to cell numbers that allow for ChIP-seq which provides much more reliable quantification. It should also be noted that the authors have not included any type of spike in normalization in their CUT&RUN experiments to control for efficiency and specificity. In addition, at least 2 of the antibodies that they use (H3K9me3 and H3K27ac) failed quality control testing in a large scale assay for antibody specificity (<https://chromatinantibodies.com/antibodies/e0531-5> and <https://chromatinantibodies.com/antibodies/e0512-1>). Performing a genome-wide quantitative ChIP-seq for the modifications implemented by each of the modifier constructs in the context of both on-target and off-target sgRNAs is of crucial importance to the interpretation to all of the authors'

experiments and is a standard in the chromatin field for measuring histone modification levels.

2. The experiments used to make the claim that H3K4me3 is instructive for transcription activation are not convincing. Many of the genes that the authors claim to be downregulated in the MLL2-CM/CM cells are expressed at extremely low levels in wildtype ESCs. The authors should provide RNA-sequencing tracks for the changes that they are observing as opposed to normalized qPCR data, which can be misleading when quantifying genes that are expressed at almost undetectable levels. Since these genes are barely expressed to begin with it's difficult to know how to interpret them being further downregulated. Given the extremely low expression of the genes shown in Figure 3B and Supplemental Figure 6A, I'm concerned that what they're looking at is random experimental variation due to either tissue culture, or during the multiple steps involved in RT qPCR. The authors should demonstrate they can rescue the expression of genes that are expressed at convincing levels (by RNA-seq) and downregulated upon mutation of MLL2. One of the well-established targets of MLL2 in ESCs is the *Magohb* gene (Glaser et al., 2009. *Epigenetics and Chromatin*; Ladopoulos et al., 2013. *Mol. Cell Biol*). *Magohb* has been shown by several groups to be convincingly downregulated upon mutation of MLL2, therefore the authors should test if they can rescue *Magohb* expression with their constructs.

The authors use the Prdm9 PR/SET domain to implement H3K4me3 in most of their experiments. This is a strange choice because Prdm9 is an atypical H3K4me3 methyltransferase that functions specifically during meiosis in mammals to direct DNA recombination machinery to recombination "hotspot" sequences. In addition to H3K4me3, Prdm9 also catalyzes H3K36me3 (Powers et al., 2016. *Nature Genetics*; Huang et al., 2020. *Elife*; Wells et al., 2020.) which is a highly unusual feature and greatly complicates the authors' experiments. The authors do perform some experiments using the SET domain of Setd1a, however, the results they observe are inconsistent with the properties of Prdm9. For instance, Prdm9 induces a ~500 fold induction of *Cldn6*, whereas Setd1a causes a ~6 fold induction, similarly, Prdm9 induces a 10 fold activation of *Fgf5*, whereas Setd1a does not cause any statistically significant increase in expression. Are these differences due to different levels of H3K4me3 implementation by these 2 enzymes? In the current version of the manuscript the authors have not performed any experiments to look at potential differences in the levels of H3K4me3 deposited by Prdm9 and Setd1a at their target genes. The authors should perform *Drosophila* chromatin spike in normalized ChIP-seq for H3K4me3 for the sgRNAs targeting all of the genes in figure 2B,C,E for both the Prdm9 and Setd1a expressing constructs to quantitatively assess the levels of H3K4me3 implemented at these sites. This should also be done for the genes shown in Supplemental Figure 6A. This is an essential control experiment to determine 1) if H3K4me3 is actually being deposited at these targets and 2) if the levels of H3K4me3 correlate with the amount of gene activation.

The authors must also test if Prdm9 is implementing H3K36me3 at their target genes using the same approach outlined in the previous paragraph. This is an essential experiment because if Prdm9 is implementing both modifications the authors cannot conclude that it is H3K4me3 that is driving activation.

Since the authors are using MLL2 (KMT2B) mutant cells it would make sense to use the KMT2B SET domain in the H3K4me3 rescue experiments. As I mentioned in my summary statement this work would benefit from using a comprehensive panel of modifiers and also include KMT2C (MLL3) or KMT2D (MLL4) which implement H3K4me1 to study the effects of different H3K4 methylation states.

3. Many of the experiments are done after 7 days of expression of the modifier constructs. This seems like an extremely long time and raises strong concerns regarding off-target and secondary effects. The

authors should perform a time course to detect how quickly modifications are implemented. Particularly for H3K4me3, if this modification is truly instructive for transcription you would expect to see increased gene expression soon after H3K4me3 appears. If there are several days of lag between appearance of the modification and gene activation, it is difficult to imagine how this is a direct causal effect.

4. The authors do not provide any mechanistic insight into how H3K4me3 might lead to gene activation. The authors should explore the biochemical mechanism through which activation might occur.

5. For all experiments claiming changes in RNA levels the authors must determine if this is due to increased transcription or increased RNA stability. This could be performed through PRO-seq or alternatively RNA Polymerase II ChIP-seq.

6. The authors should characterize the proteins that associate with each of their modifier constructs by performing immunoprecipitation followed by mass spectrometry. This is important because in addition to possessing catalytic activity, the catalytic domains may also mediate protein-protein interactions and recruit co-regulators that influence gene expression. This should be done for both the wildtype as well as the catalytic mutants because the catalytic mutations may disrupt protein folding and lead to loss of interactions.

The authors should provide western blot experiments demonstrating similar levels of expression of the wildtype and catalytic mutants.

I was unable to find citations for the source of all the catalytic dead mutations in this paper. I did a web search as was able to find that some had been previously reported, but for others such as Prdm9, I could not find publications characterizing these mutants. The authors should cite the primary literature that characterizes these mutations.

7. For the experiments presented in Figure 4 involving the introduction of different transcription factor binding motifs into their reporter construct, the authors must demonstrate that the transcription factors of interest are actually binding to the motifs that they have introduced into their reporter by performing ChIP-seq. It is generally accepted that only a fraction of consensus TF motifs in chromatin are actually bound by their cognate TF, potentially due to variations in chromatin accessibility. The authors have made the assumption that simply by inserting TF-motifs that they are bound, however, this must be tested experimentally by ChIP-seq. In this case, *Drosophila* chromatin spike in is not absolutely necessary because unlike histone modifications, TF epitopes are likely not conserved. However, the authors could perform spike in using a kit such as the Active Motif spike-in normalization kit to normalize overall ChIP efficiency. Also, in the case of motifs that can be bound by multiple proteins of a TF class (E-box, GATA, OTX) it is essential to identify specifically which TFs are bound to which TF-motifs.

The authors should also perform quantitative ChIP-seq to measure the amount of each histone modification that is deposited on the various TF-motif containing constructs. Do the alterations in gene expression that they observe correlate with the levels of histone modifications that are implemented? This has been done for H3K36me3 in figure 4H, but should also be performed for all the TF-motif modifier combinations for which the authors claim to see an effect on gene expression (e.g. H2AK119 and YY1 motif; H3K9me3 and YY1 motif).

8. I found the most interesting result in this study to be the ability of CTCF motifs to strongly enhance transcriptional repression by H3K36me3. However, the authors do not explore the mechanism of how this occurs. H3K36me3 is generally associated with the gene bodies of actively transcribed genes and studies in yeast suggest that it may function to suppress cryptic intra-genic transcription (although this function may not be conserved in mammals). So the ability of H3K36me3 to suppress transcription is interesting but it is completely unexplained in the authors work and how this links to CTCF is not examined mechanistically. As stated above, the authors must experimentally demonstrate that CTCF is actually binding to their CTCF construct using ChIP-seq (looking at Cohesin binding would also be informative). Unlike the effects of H3K4me3 on gene activation which seem to be modest at best and not reproducible across multiple loci, the combination of H3K36me3 and CTCF does appear to induce strong silencing of their reporter construct. An interesting experiment would be to identify an endogenous region with multiple CTCF sites and attempt to silence it by targeting Setd2/H3K36me3. If this is successful, then knock-out the CTCF sites and demonstrate a reduction in repression.

Since H3K36me3 is associate with active gene bodies, what happens when the authors target it to a gene body instead of a promoter?

9. The authors should provide western blots confirming lack of cMyc and PCGF6 protein in their knockout cells. Also does knockout of cMyc cause general transcriptional changes in ESCs? If so the effects the authors observe could be secondary. The use of an acute depletion model (Auxin-AID or dTag) would be more mechanistically informative.

10. The experiments in this paper are performed entirely in mouse embryonic stem cells (mESCs), which greatly limits the relevance of the authors' findings as ESCs have a number of atypical features (lack of several cell cycle checkpoints, aberrant maintenance of DNA methylation particularly at imprinted genes). The generation of mice harboring the authors' chromatin editing constructs and performing experiments in real tissues would greatly enhance the impact of this study. If the authors can rescue an in vivo animal phenotype caused by lack of a histone modifier (for example PRC2 deficiency) I feel that would bring the paper to the level where it could be considered by a top-tier journal.

11. Recent experiments using histone H3K27 point mutants which completely ablate both H3K27me3 and H3K27ac (McKay et al., Sankar et al., 2015. Developmental Cell; 2022. Nature Genetics) have demonstrated that although H3K27me3 is essential for PRC2 mediated silencing, H3K27ac is dispensable for gene activation. This suggests that H3K27ac is likely not the crucial substrate for CBP/p300 which are known to acetylate a wide range of both histone and non-histone proteins (Weinert et al., 2018. Cell). Therefore, the authors should examine what other alterations in histone acetylation patterns are induced by their CBP modifier construct by ChIP-seq. An extremely informative experiment would be to obtain the H3K27R complete mutant ESCs generated by Sankar et al. and then test the patterns of both gene expression through RNA-seq and various sites of histone acetylation (e.g. H3K18ac, H2B-N-terminal acetylation, see Weinert et al. for validated CBP/p300 histone substrates) genome-wide by ChIP-seq.

12. I am not convinced that combined effects of H3AK119ub and H3K27me3 are synergistic as the authors claim, the effect seems to simply be additive based on the data presented.

Reviewer #3:

Remarks to the Author:

The use of fusion proteins to catalytically dead versions of CRISPR CAS proteins now makes it possible to place specific epigenetic marks at defined locations in the human genome. To date, such programmed epigenetic editors have been used primarily as a means to control gene expression rather than as tools to dissect the mechanisms underlying chromatin-based transcriptional regulation. In addition, only a small set of protein domains have been successfully engineered as epigenetic editors (DNA methyltransferase and KRAB domains being the most widely implemented so far).

Policarpi and colleagues fill both of these gaps in the literature and in doing so make an important contribution to the field of epigenetic editing. They build an impressive repertoire of new histone mark modulators and deploy them to understand poorly characterized and/or controversial aspects of chromatin-mediated transcriptional control. Their work on H3K4me3 is particularly thorough, including the identification of dose-dependent H3K4me3-driven gene activation and a clear link between H3K4me3 activation and promoter acetylation. Other important findings relate to the relationship between specific DNA motifs and the transcriptional impact of histone modifications made possible by their elegant allelic series experiments. The identified association between CTCF motifs and enhanced H3K36me3 silencing is striking, as is the uncovered role of PRC1.6 recruitment to MYC motifs in limiting H3K4me3 activation. Overall, the manuscript offers both a valuable set of tools for the community as well as impactful new insights into chromatin biology. As such the manuscript has the potential to make a strong addition to Nature Genetics. Below are a number of suggestions and concerns that should be addressed either through textual changes and/or experiments prior to publication.

Major Points

The epigenetic editing platform the authors developed has the potential to be broadly enabling for the community but the systems would greatly benefit by more careful delineation of its limitations. Below are some specific examples.

Toxicity. There is no mention of whether expressing any of the epigenetic editors is toxic to cells.

Off-target effects. While the authors carry out RNA-seq experiments to show that transcriptional off-targets are rare (with the major exception of p300-CD-scFV), they do not assess off-target deposition of histone modifications. ChIP-seq or CUT&RUN on cells expressing each editor would be of value for more broadly exploring such off target effects. The signal observed in the negative control of Fig. 1I suggests off-targets may be a concern.

Multiplexing potential. A major advantage of the CRISPR-dCas9 system is the ability to multiplex using sgRNAs against different targets in the same cell. The authors could explore this by targeting more than one gene at the same time and assessing the degree to which the intended histone mark is robustly established at multiple loci.

Targeting window. It would be helpful to provide a better idea of how flexible the targeting window of each editor is, and whether different target sites across a given promoter result in different patterns of histone modification deposition.

Generalizability. It is difficult to get a sense of how effective the editors are across different gene targets since most of the experiments are carried out on the Hbby gene or their engineered EF1a reporter. While assessing the generalizability of all the editors is likely outside the scope of this

manuscript, categorizing the chromatin marks into three groups based on their reporter read-out alone may be misleading since the effects could vary widely across different targets. The grouping also seems arbitrary at times: based on the data in Fig. 2C-K, DNA methylation is indistinguishable from the modifications in the subtle/partially repressing group.
Open access. The plasmids encoding each of the editors should be deposited to Addgene (or the equivalent) upon publication.

While the authors present compelling evidence that H3K4me3 deposition can activate gene expression, it is still unclear whether this has any functional significance. Their EpiLC differentiation experiment attempted to get at this but unfortunately only the most lowly expressed genes failed to activate in the absence of H3K4me3. Do the authors believe these low expression levels are biologically meaningful? Ideally they would show that H3K4me3-mediated activation of these genes is required for differentiation (or some other function). If not, the authors should acknowledge that their results are limited to lowly expressed genes and do not necessarily point to a functional role.

The fact that the recruitment of the YY1 motif can block both activating and repressing effects is intriguing, especially given the literature showing that YY1 can have both activating and repressing effects on gene expression. Could the authors elaborate on this further? Are these unintuitive effects of the YY1 motif generalizable to endogenous genes? Is it fair to think of YY1 as a “transcriptional buffer”?

Minor points

The manuscript contains a lot of data and I understand which makes it difficult to condense all of the experimental design details. However, more justification for why and how certain decisions were made would be helpful. More details on the following would be appreciated.

How were the nine histone and DNA modifications selected?

Do the five copies of GCN4 upon one copy? This is not always the case (e.g. steric hindrance).

Why were the full-length enzymes used for Ezh2 and Kmt5c while most editors were distilled down to catalytic domains?

How were positive and negative controls selected in Fig. 1D-I? Strongest and weakest peaks possible, respectively? Based on which datasets?

How was Hbby selected for the initial targeting experiments and why were different gene targets used for DNA methylation?

Most experiments use a 7-day time point. How was this time point chosen?

How were the permissive (chr9) and non-permissive (chr13) sites selected?

How were the 8 genes in the H3K4me3 targeting rescue experiments picked? What about the 8 endogenously silent genes? The paper claims the latter were random – how were they randomized?

Do the DNA motifs investigated in the study correspond to the consensus mouse motifs or consensus human motifs? It would be interesting to test if equivalent human motifs have the same effect (i.e. is the effect of a given TF motif conserved even if the sequence of the motif itself is not?).

Is their Dox-inducible system leaky?

The authors mention that using 20-fold lower Dox was used to mitigate the off-target effects of p300-CD-scFV but do not show the resulting improved data following Dox titration.

Minor line-referenced comments:

Line 97-99: 'peak... centered on the gRNA binding site' – seems like the peak is actually upstream of sgRNA site for most marks?

Line 112-113: cannot claim lack of off-target histone marks based on two loci.

Line 130: what is the CpG content of the chosen sequence? Could low CpG density explain underwhelming DNA methylation effects?

Line 221: surprising that 53 genes are upregulated upon loss of H3K4me3 – which genes are these?

Line 253-156: did the authors try successive targeting? i.e. target, sort off cells, target again, assess transcriptional activation.

Line 259: what is meant by 'productive'? Should comment that productive transcription is not necessarily functional.

Line 364: the authors make the important point that the function of a chromatin modification is "highly context dependent". An important caveat to add is that most of their experiments are done on an exogenous EF1a reporter, albeit integrated at two genomic regions.

Lines 406-411: it is surprising that no epigenetic memory was observed after DNA methylation targeting given previous reports. Do the authors think this is primarily a result of working in ESCs or more likely to be a reporter-specific effect?

Comments on figures:

In general the figures are well constructed and aesthetically pleasing. Below are some suggestions for each that would improve clarity.

General:

Could individual data points be added to all the bar charts? Sometimes they're present and sometimes they're not.

Figure 1:

C: Size of region assessed? Same as in J?

G and H: The split y-axes are misleading – edited mark does not reach levels observed at positive controls.

D-I: Missing gene names for positive and negative controls. How were positive and negative controls selected?

K: Surprising that there are a number of downregulated genes following H3K27ac targeting – are these genes targets of upregulated genes?

J: where are the sgRNA binding sites for these genes?

Figure 2:

B: Could a population of cells without the reporter be included in this graph and other flow cytometry data in the paper to get a sense of what a completely 'off' state looks like? It would be nice to directly compare 'off' cells to edited silenced cells to determine whether some of the editors are capable of completely turning off gene expression.

C-K: It took a while to figure out which reporter is being assessed for each graph. Could the label be moved from the y axis to somewhere more obvious? Perhaps to the left of each row of graphs?

E: which region of the contextual DNA sequence + promoter was assessed here?

Figure 3:

C-E: There is a very significant difference in activation of Cldn16 between Prdm9 and Sete1a targeting. Is this related to the degree of H3K4me3 deposited? Could H3K9me3 enrichment for this particular gene be added to panel D?

What are biological replicates in this context?

C: Data for other 4 targeted genes missing?

C and E: Genes should appear in the same order and aligned one on top of another for easy comparison.

G: missing 'C' in mCherry

Figure 4:

B: Colors are hard to distinguish

G-I: Beautiful results

Supp Fig. 1:

A: Add number of NLSs and construct diagrams for each editor

B: Show WT population to assess leakiness of Dox system

D: DNA methylation data missing

E: Gene names for controls?

E: Split y-axis misleading

Supp Fig. 3:

A and B: Axis labels missing

Supp Fig. 5:

B: Will a list of the up- and downregulated genes be provided in a supplementary table?

Supp Fig. 6:

B: Equalize axes of two graphs

Supp Fig. 7:

C and D: What are the sample sizes for these graphs?

Supp Fig. 8:

C: Confusing layout – in some cases two different clones are compared and in others different motifs. Could a separate panel be created for clonal comparisons?

B and C: Only select examples are shown. As a supplemental figure there should be enough space to include all the data.

Supp Fig. 9:

Statistics?

Reviewer #4:

Remarks to the Author:

The use of fusion proteins to catalytically dead versions of CRISPR CAS proteins now makes it possible to place specific epigenetic marks at defined locations in the human genome. To date, such programmed epigenetic editors have been used primarily as a means to control gene expression

rather than as tools to dissect the mechanisms underlying chromatin-based transcriptional regulation. In addition, only a small set of protein domains have been successfully engineered as epigenetic editors (DNA methyltransferase and KRAB domains being the most widely implemented so far).

Policarpi and colleagues fill both of these gaps in the literature and in doing so make an important contribution to the field of epigenetic editing. They build an impressive repertoire of new histone mark modulators and deploy them to understand poorly characterized and/or controversial aspects of chromatin-mediated transcriptional control. Their work on H3K4me3 is particularly thorough, including the identification of dose-dependent H3K4me3-driven gene activation and a clear link between H3K4me3 activation and promoter acetylation. Other important findings relate to the relationship between specific DNA motifs and the transcriptional impact of histone modifications made possible by their elegant allelic series experiments. The identified association between CTCF motifs and enhanced H3K36me3 silencing is striking, as is the uncovered role of PRC1.6 recruitment to MYC motifs in limiting H3K4me3 activation. Overall, the manuscript offers both a valuable set of tools for the community as well as impactful new insights into chromatin biology. As such the manuscript has the potential to make a strong addition to Nature Genetics. Below are a number of suggestions and concerns that should be addressed either through textual changes and/or experiments prior to publication.

Major Points

The epigenetic editing platform the authors developed has the potential to be broadly enabling for the community but the systems would greatly benefit by more careful delineation of its limitations. Below are some specific examples.

Toxicity. There is no mention of whether expressing any of the epigenetic editors is toxic to cells.

Off-target effects. While the authors carry out RNA-seq experiments to show that transcriptional off-targets are rare (with the major exception of p300-CD-scFV), they do not assess off-target deposition of histone modifications. ChIP-seq or CUT&RUN on cells expressing each editor would be of value for more broadly exploring such off target effects. The signal observed in the negative control of Fig. 1I suggests off-targets may be a concern.

Multiplexing potential. A major advantage of the CRISPR-dCas9 system is the ability to multiplex using sgRNAs against different targets in the same cell. The authors could explore this by targeting more than one gene at the same time and assessing the degree to which the intended histone mark is robustly established at multiple loci.

Targeting window. It would be helpful to provide a better idea of how flexible the targeting window of each editor is, and whether different target sites across a given promoter result in different patterns of histone modification deposition.

Generalizability. It is difficult to get a sense of how effective the editors are across different gene targets since most of the experiments are carried out on the Hbby gene or their engineered EF1a reporter. While assessing the generalizability of all the editors is likely outside the scope of this manuscript, categorizing the chromatin marks into three groups based on their reporter read-out alone may be misleading since the effects could vary widely across different targets. The grouping also seems arbitrary at times: based on the data in Fig. 2C-K, DNA methylation is indistinguishable from the modifications in the subtle/partially repressing group.

Open access. The plasmids encoding each of the editors should be deposited to Addgene (or the equivalent) upon publication.

While the authors present compelling evidence that H3K4me3 deposition can activate gene expression, it is still unclear whether this has any functional significance. Their EpiLC differentiation experiment attempted to get at this but unfortunately only the most lowly expressed genes failed to activate in the absence of H3K4me3. Do the authors believe these low expression levels are biologically meaningful? Ideally they would show that H3K4me3-mediated activation of these genes is required for differentiation (or some other function). If not, the authors should acknowledge that their results are limited to lowly expressed genes and do not necessarily point to a functional role.

The fact that the recruitment of the YY1 motif can block both activating and repressing effects is intriguing, especially given the literature showing that YY1 can have both activating and repressing effects on gene expression. Could the authors elaborate on this further? Are these unintuitive effects of the YY1 motif generalizable to endogenous genes? Is it fair to think of YY1 as a “transcriptional buffer”?

Minor points

The manuscript contains a lot of data and I understand which makes it difficult to condense all of the experimental design details. However, more justification for why and how certain decisions were made would be helpful. More details on the following would be appreciated.

How were the nine histone and DNA modifications selected?

Do the five copies of GCN4 upon one copy? This is not always the case (e.g. steric hindrance).

Why were the full-length enzymes used for Ezh2 and Kmt5c while most editors were distilled down to catalytic domains?

How were positive and negative controls selected in Fig. 1D-I? Strongest and weakest peaks possible, respectively? Based on which datasets?

How was Hbby selected for the initial targeting experiments and why were different gene targets used for DNA methylation?

Most experiments use a 7-day time point. How was this time point chosen?

How were the permissive (chr9) and non-permissive (chr13) sites selected?

How were the 8 genes in the H3K4me3 targeting rescue experiments picked? What about the 8 endogenously silent genes? The paper claims the latter were random – how were they randomized?

Do the DNA motifs investigated in the study correspond to the consensus mouse motifs or consensus human motifs? It would be interesting to test if equivalent human motifs have the same effect (i.e. is the effect of a given TF motif conserved even if the sequence of the motif itself is not?).

Is their Dox-inducible system leaky?

The authors mention that using 20-fold lower Dox was used to mitigate the off-target effects of p300-CD-scFV but do not show the resulting improved data following Dox titration.

Minor line-referenced comments:

Line 97-99: ‘peak... centered on the gRNA binding site’ – seems like the peak is actually upstream of sgRNA site for most marks?

Line 112-113: cannot claim lack of off-target histone marks based on two loci.

Line 130: what is the CpG content of the chosen sequence? Could low CpG density explain underwhelming DNA methylation effects?

Line 221: surprising that 53 genes are upregulated upon loss of H3K4me3 – which genes are these?
 Line 253-156: did the authors try successive targeting? i.e. target, sort off cells, target again, assess transcriptional activation.
 Line 259: what is meant by 'productive'? Should comment that productive transcription is not necessarily functional.
 Line 364: the authors make the important point that the function of a chromatin modification is "highly context dependent". An important caveat to add is that most of their experiments are done on an exogenous EF1a reporter, albeit integrated at two genomic regions.
 Lines 406-411: it is surprising that no epigenetic memory was observed after DNA methylation targeting given previous reports. Do the authors think this is primarily a result of working in ESCs or more likely to be a reporter-specific effect?

Comments on figures:

In general the figures are well constructed and aesthetically pleasing. Below are some suggestions for each that would improve clarity.

General:

Could individual data points be added to all the bar charts? Sometimes they're present and sometimes they're not.

Figure 1:

C: Size of region assessed? Same as in J?

G and H: The split y-axes are misleading – edited mark does not reach levels observed at positive controls.

D-I: Missing gene names for positive and negative controls. How were positive and negative controls selected?

K: Surprising that there are a number of downregulated genes following H3K27ac targeting – are these genes targets of upregulated genes?

J: where are the sgRNA binding sites for these genes?

Figure 2:

B: Could a population of cells without the reporter be included in this graph and other flow cytometry data in the paper to get a sense of what a completely 'off' state looks like? It would be nice to directly compare 'off' cells to edited silenced cells to determine whether some of the editors are capable of completely turning off gene expression.

C-K: It took a while to figure out which reporter is being assessed for each graph. Could the label be moved from the y axis to somewhere more obvious? Perhaps to the left of each row of graphs?

E: which region of the contextual DNA sequence + promoter was assessed here?

Figure 3:

C-E: There is a very significant difference in activation of Cldn16 between Prdm9 and Sete1a targeting. Is this related to the degree of H3K4me3 deposited? Could H3K9me3 enrichment for this particular gene be added to panel D?

What are biological replicates in this context?

C: Data for other 4 targeted genes missing?

C and E: Genes should appear in the same order and aligned one on top of another for easy

comparison.

G: missing 'C' in mCherry

Figure 4:

B: Colors are hard to distinguish

G-I: Beautiful results

Supp Fig. 1:

A: Add number of NLSs and construct diagrams for each editor

B: Show WT population to assess leakiness of Dox system

D: DNA methylation data missing

E: Gene names for controls?

E: Split y-axis misleading

Supp Fig. 3:

A and B: Axis labels missing

Supp Fig. 5:

B: Will a list of the up- and downregulated genes be provided in a supplementary table?

Supp Fig. 6:

B: Equalize axes of two graphs

Supp Fig. 7:

C and D: What are the sample sizes for these graphs?

Supp Fig. 8:

C: Confusing layout – in some cases two different clones are compared and in others different motifs. Could a separate panel be created for clonal comparisons?

B and C: Only select examples are shown. As a supplemental figure there should be enough space to include all the data.

Supp Fig. 9:

Statistics?

Author Rebuttal to Initial comments

Reviewer #1:

C1: The paper by Policarpi et al is an impressive and interesting showcase of a powerful set of epigenetic editing tools, which are applied toward resolving key challenges relating the functional hierarchy among epigenetic modulators and transcription control. While there is a growing body of literature on dCas9-based epigenetic editing, I find the current work is taking several key steps toward a reliable, integrated and potent

scheme for resolving mechanisms and causal relationship between the transcriptionally promoting and repressive machineries. Beyond the technical contributions (the system itself), there are several truly important results. In that context I find of particular interest: a) experiments targeting H3k4me3 and positioning it within the feedback loop of transcriptional activation, b) some insightful results regarding polycomb-mediated repression and its complex interactions with other systems (PRC1.6, YY1, possible H2K119ub/H3K27me3 synergism), c) CTCF, H3k36me3 and linked effects.

I would like to stress that in my opinion, this paper is important not just because of any of these results individually – but because it is approaching the problem holistically and set up a system and a conceptual framework that can lead to much better quantitative understanding than the modest and in some cases non-existent models we have today. So even though the paper leaves many open questions, is restricted to work on mESC and is for the most part not expanding its conclusion to a epigenome-wide scale – I think it is of very high interest and quality as is, and support its timely publication.

R1: We thank the reviewer for their balanced and constructive feedback.

C2: I have very few comments for improvement and further consideration. The major comment relates to the uniqueness of the pluripotent/mESC state and the need to better acknowledge it. It is clear from the text that the authors understand very well that mESCs represent a unique epigenetic state, but the text (intro, discussion) can (and should) make it much clearer to readers that the hierarchy of epigenetic factors that is studied here is likely to be specific to this state. There are many mechanisms contributing to epigenetic turnover and plasticity in mES, and while these are not truly understood yet, it is unlikely that the authors results would reproduce as is in differentiating embryonic lineages, homeostatic adult stem cells or other systems of interest. It is truly important to be open about this point and poise the paper as addressing questions of epigenetic function and hierarchy in the pluripotent state. The discussion can include a short analysis of the possible implications to other cellular states. For the epigenetic editing system itself, it is interesting to ask if its efficiency will be similar in less plastic cellular states. If there are preliminary results on this it will be important to include them. But I don't suggest this should delay a revision process.

R2: We fully agree that the underlying cellular identity is likely an important factor for the precise impact/memory of a chromatin mark at any given locus. This is particularly evident for mESC, which as the reviewer articulates, carry a unique cellular and chromatin state. Future work should experimentally dissect the three-way interactions between cellular state(s), epigenetic mechanisms and *cis*-genetic context. We have modified the manuscript to further emphasise the uniqueness of pluripotent mESC utilised

predominantly in our study, and to highlight that cell-type likely plays an important contextual role in functional responses at any given locus. For example, in the discussion we now state:

“...Indeed, cell identity is likely another important factor influencing response to specific marks at specific loci, with the pluripotent cells deployed here potentially conveying distinct responses to differentiated contexts owing to the unique constellation of chromatin states and TF milieus. Thus, whilst our data imply that chromatin marks have the potential to causally instruct transcription programmes, they also highlight they represent one regulatory layer within multiple nonlinear governing mechanisms.”

In specific relation to epigenetic memory, we now point out more clearly that the absence of memory we observe may be indicative of the pluripotent cell milieu, which is in line with our previous study (Carlini et al., 2022):

“Such lack of ‘epigenetic memory’ may reflect the cell-type specific context of ESC, since recent observations indicate that acquired heterochromatin domains do not propagate in naïve pluripotent cells, but do so in differentiated cellular contexts (Carlini et al., 2022).”

C3: A point in which some additional analysis may be insightful is around Fig 3H. Some attempt to characterize the H3k4me3 dependent genes is not described. Analysis of different genomic features can be helpful: CpG content on promoters, GC content, distance to nearby CTCF site, distance to nearby TAD boundary, distance to other active promoters. It also make sense to run a simple DNA motif screen on this set.

R3: A deeper understanding of why some genes appear at least partially dependent on H3K4me3 *per se*, whilst others not, is important. We decided to reproduce the entire *MI12^{CM/CM}* ESC to EpiLC transition experiment using independent lines, reading out both transcription and now chromatin profiles as well. We did this for two reasons: (1) Reviewer #2 raised concerns about

reproducibility of relatively small effect sizes. (2) In re-analysing the data, we observed that there was a mis-match between the number of male and female ESC lines used for WT vs *Mll2^{CM/CM}*. We note this situation arose because all ESC lines were independently derived, freshly from blastocysts, which ensures low-passage, high-quality, and ‘real’ biological independent replicates to compare. To address the above points, we have now repeated the ESC-EpiLC transition experiment with all female freshly-derived WT or *Mll2^{CM/CM}* ESC lines. The data generated not only reproduces our previous result, but extends the power to detect genes that robustly fail to activate upon cell fate transition. We now find 498 genes to be dependent on MLL2 catalytic activity in EpiLC, of which 313 are completely OFF in ESC, indicative of failure to *initiate* transcription without H3K4me3 (see new Fig 3I-J, new Extended Data Fig 8D-E). EpiLC differentiation itself is normal as judged by a range of markers (new Extended Data Fig 8B-C). We have now also investigated the features associated with genes that fail to appropriately activate during differentiation of *Mll2* catalytic-mutant cells. We observe some gene ontology and motif enrichments amongst DEGs in cells that specifically lack H3K4me3. For example, SRF motifs are particularly prominent in the group that fails to activate from an initial ‘OFF state’ in *Mll2^{CM/CM}* EpiLC. In contrast, in general CpG density changes are very subtle amongst genesets. This data is shown right.

C4: Another possible genome-wide follow up is relating the observation on YY1 and E-BOX motifs to context specific epigenetic states. Are YY1 motifs appears at lower frequencies in endogenous polycomb domains of mES? For MYC and PRC1.6 binding to E-BOX, are loci (promoters?) with a strong E-box more likely to be repressed in mES and transition to activation in MYC expression lineages? (e.g. hematoendothelial progenitors and primitive erythrocytes?). I would like to stress that I find the mechanistic and reductionist approach of this paper very useful and appropriate. If the above suggestion for follow up are leading to some useful insight it can only strengthen the paper (and if such results are difficult to derive, given the numerous confounders that are affecting such analyses, it will not undermine the relevance of the experiments reported here).

R4: We have performed a number of integrative analyses to understand the (epi)genomic features and enrichments associated with specific motifs. For example, we find that EBOX (Myc) motifs are associated with an increased level of polycomb deposition (H3K27me3), consistent with our observation that E-box motifs impact the extent of induced gene activation, by promoting recruitment of polycomb (PRC1.6) to counteract activation. However, given the complexity of the inputs into endogenous gene regulation we have been unable to derive clear relationships elsewhere. For example, whilst YY1 motifs occur at lower frequency in polycomb marked promoters, the effect size is very small and results are exquisitely dependent on analysis parameters. Indeed, analysis shows that in general, promoters (1kb—200bp) that contain at least one E-box motif(s) are expressed at a similar overall level to those without, which likely reflects complexity

and redundancy, whereby an E-Box motif can be bound by many TF that act both positively or negatively, and are frequently co-associated with other local factors/motifs that further influence outcome. In contrast our reductionist system can isolate specific quantitative relationships that are otherwise obscured. Thus, whilst correlating (epi)genomic features is extremely powerful, many confounders affect such analysis, and we are therefore cautious in the interpretation. To emphasise the power of our reductionist system to identify new biology we now focussed on the interaction between H3K36me3 and CTCF motifs. We have generated an entirely *new* Figure 5 exploring this, with a full description found in R13 (reviewer #2) below.

C5: Typo: label in Fig S3A: H4K9me3 -> H3K9me3

R5: Thank you, corrected.

Reviewer #2:

Remarks to the Author:

C1: In this manuscript, Policarpi et al. use a catalytically dead Cas9-SunTag construct to target a set of 9 chromatin modifiers to a fluorescent reporter gene as well as selected endogenous loci and then measure the effects on gene expression. The authors make the conclusions that H3K4me3 is instructive for transcriptional activation, that DNA motifs influence the effects of chromatin modifications and that H2AK119ub (mediated by PRC1) and H3K27me3 (mediated by PRC2) function synergistically. While this work reports a few interesting but albeit unexplained phenomena, overall, I found it to be preliminary, lacking in mechanistic detail and missing a large number of essential control experiments. I also don't see the potential for a broad and sustained impact for the field of genetics and feel this work would be better suited to a journal focused on chromatin biochemistry.

R1: In this study we developed a novel set of precision epigenome perturbation tools that enable the *functional* impact of acquiring a wide range of chromatin modifications to be interrogated. The scale of the approach - and the number of controls - are we believe, unprecedented. As a result, we were able to determine the quantitative, single-cell and context-dependent impacts of installing each of nine key chromatin modifications at specific loci, thus unpicking the cause and consequence relationship with genome regulation that is crucial towards interpretation of (epi)genomics data. Whether this study has a

broad impact of course remains to be determined by our colleagues. We do however note that the pre-printed abstract was accessed >10,000 times in the first few weeks, that Altmetric placed the article ‘attention score’ within the top 1500 articles *ever* submitted to BioRxiv (>99th percentile), and that many colleagues have requested the reagents/tools we developed herein. This suggests that there may well be a significant potential for “*a broad and sustained impact*”.

C2: While the authors test multiple modifiers, this approach is not particularly novel as the dCas9-SunTag system has already been used to edit chromatin by a number of groups (For a review of this see: Pulecio et al., 2018. “CRISPR/Cas9-based engineering of the epigenome” Cell Stem Cell). Additionally, the authors have not performed a comprehensive analysis of chromatin modifiers and in some cases their choices of which enzymes to examine seems arbitrary. For instance, they use G9a to implement H3K9me2, but do not examine the activity of Suv39 or SETDB1. Similarly, they primarily use the PR/SET domain of Prdm9 to implement H3K4me3 and perform a few validation experiments with the SET domain of Setd1a, however, they do not examine the well characterized KMT2/MLL class of H3K4 methyltransferases. I feel that extending this study to include all known histone methyltransferase and acetyltransferase domains would make it more broadly relevant, although it would still be best suited as a resource in a chromatin-centric specialty journal.

R2: In order to build a new and broad system, we **comprehensively tested the activity of >30 catalytic cores**, and expended several years in optimising the domain, linkers and design of the system. As can be imagined, we came across many enzymes domains which, despite extensive optimisation, did not install their modification sufficiently owing to myriad regulatory requirements (such as allosteric activation) and/or broader challenges within programmable synthetic systems. This included testing H3K9me3 writers such as SETDB1, and H3K4me3 writers including MLL2, as suggested by the reviewer. In these cases, the domains were not amenable to engineering with sufficient activity as to pass our quality control for ON-target activity (see **R6** for more information). Thus, in order to develop our epigenetic editing system – which considerably surpasses the scale, specificity and flexibility of any effort to date – we had already tested the role of a broad fraction of known “*histone methyltransferase and acetyltransferase domains*” upstream. As a consequence of this testing regime, we engineered a unified toolkit that specifically programmes nine of the most prominent and key chromatin modifications – or their combinations – with a high magnitude effect-size, enabling their role to be evaluated. We also built catalytic-mutant versions, designed the system to be dynamic and multiplexable, and rigorously tested their activity/specificity. The technology represents a major advance in the tools available to interrogate chromatin function and it is not clear to us how adding more domains would make the already comprehensive study “*more broadly relevant*,” especially given the novel observations already discerned. To address the reviewer point, we have now added information in the supplementary information (also appended below) detailing the full range of domains that we tested in order to assemble a validated and optimised toolkit:

EFFECTOR	HISTONE MODIFICATION	DOMAIN	ON-TARGET QC
PRDM9 ^{scFV}	H3K4me3	aa 110-417	+++
SETD1A ^{scFV}	H3K4me3	aa 1424-1716	++
MLL2 ^{scFV} long	H3K4me3	aa 2199-2713	+
MLL2 ^{scFV} short	H3K4me3	aa 2573-2713	-
DOT1L ^{scFV}	H3K79me2/3	aa 1-415	++
P300 ^{scFV}	adds Ac	aa 1047-1663	++
P300 ^{scFV} short	adds Ac	aa 1299-1613	+++
ELP3 ^{scFV}	adds Ac	aa 341-547	-
KAT2A ^{scFV}	adds AC	aa 440-699	-
HDAC1 ^{scFV}	removes Ac	aa 1-372	-
SIRT6 ^{scFV}	removes Ac	aa 1-334	-
SETD2 ^{scFV}	H3K36me3	aa 1392-1688	+++
EZH2 ^{scFV}	H3K27me3	aa 1-746	++
EZH2 ^{scFV} short	H3K27me3	aa 531-704	-
FOG1 ^{scFV}	H3K27me3	aa 1-31	+
G9A ^{scFV}	H3K9me2/3	aa 954-1263	+++
G9A ^{scFV} human	H3K9me2/3	aa 886-1267	+
SUV39H ^{scFV}	H3K9me3	aa 43-412	-
KRAB ^{scFV}	H3K9me/DNAme	aa 2-62	+++
SETDB1 ^{scFV}	H3K9me3	aa 744-1307	-
RING1B ^{scFV}	H2A119Kub	aa 1-200	+++
KMT5C ^{scFV}	H4K20me3	aa 1-468	++
KMT5C ^{scFV} short	H4K20me3	aa 1-350	-
KMT5B ^{scFV}	H4K20me3	aa 126-365	-
KMT5A ^{scFV}	H4K20me1/3	aa 208-350	-
DNMT3A3L ^{scFV}	DNAme	aa 608-908/ aa 207-421	+++

TET1^{scFV}	removes DNAm	aa 1367-2039	+++
PADI4^{scFV}	H1R54ci	aa 1-666	-
OGT^{scFV}	O-GlcNAc	aa 545-1034	+
KDM4C^{scFV}	Lysine demethylase	aa 1-385	+
PRMT5^{scFV}	H4R3me2	aa 169-637	-

Table 1 showing initial catalytic core domains tested for epigenetic editing within the Cas9^{5xGCN4} system. Further optimisations were undertaken using differential nuclear localisation sequences (NLS), alternative linker parameters and/or by extending the domain utilised. Greater '+' number indicates higher level of deposition, '-' indicates no activity detected.

C3: It is also worth considering that when histone modifications are implemented in cells, their activity is generally restricted by multiple histone “reader” domains present in histone modifiers and/or regulation of the enzymes at the level of catalytic activity. Thus, the system that the authors use is quite artificial and may be prone to artefacts that have no real relevance in vivo. This can be seen by the very broad distribution of H3K4me3 mediated by Prdm9, as this modification is generally observed as narrow peaks associated with gene promoters. I found the experiments relating to H3K4me3 and transcription to be unconvincing. More problematically, this study lacks many essential control experiments, and therefore, I feel the data does not support the authors' conclusions.

R3: As detailed above, we extensively confirmed our system programmes *de novo* chromatin modifications to target loci to comparable levels as endogenously marked regions in both amplitude (enrichment) and breadth (genomic distribution) (see Fig 1D-I), suggesting it largely recapitulates normal chromatin domains. Like most perturbation strategies however, the system is inherently artificial – it is a *programmable epigenetic editing* toolkit. Our strategy however provides a number of major advantages over existing perturbation approaches, thus enabling novel insights into the fundamental principles of chromatin function. For example: (1) Precision – we can examine the impact of establishing a specific mark at a specific locus at endogenous genes. This avoids the major confounding impact(s) of global perturbations (i.e. secondary/pleiotropic changes), which typically manipulate chromatin-modifying enzymes or histone residues. (2) Gain-of-function – we can interrogate the impact of acquiring a modification, which may be functionally very different from the impact of deleting a modification, as has been standard in field. (3) Context-dependency - we can examine specific interplay between various chromatin marks and/or DNA sequence with reductionist strategies. (4) Dynamics - we can test temporal and memory responses with precision. Along with other advantages, this enables discovery of functional effects/interactions that were previously intractable.

With respect to the exemplar given of H3K4me3, we would like to point out that this modification is routinely found in relatively broad TSS peaks that span 2-5kb across cell types, as well as the narrower peaks mentioned (<1kb). Indeed, cumulative evidence indicates that these ‘broad’ H3K4me3 promoter peaks are most tightly linked with transcriptional activity, and essential genes (Benayoun et al., 2014; Chen et al., 2015; Liu et al., 2016). In addition, our rescue experiment shows that upon restoration of H3K4me3 via our epigenetic editing system, genes that had lost the mark and undergone downregulation, are efficiently re-activated (see Fig 3C). Thus, contrary to the assertion, our system recapitulates endogenous H3K4me3 domains (and other marks) that have an extremely high “biological relevance *in vivo*”. We make this point about H3K4me3 breadth in the discussion (*see* line 522-525).

Specific Comments

C4: A major flaw of this work is that it fails to adequately address off-target effects caused by overexpression of histone modifier catalytic domains. Throughout the paper the authors use a GFP-scFV or catalytic dead enzyme as a control. In my opinion, these are not appropriate controls because they do not control for off-target effects mediated by overexpression of the active catalytic domains, which could implement their respective modifications at aberrant sites throughout the genome. The authors should perform experiments expressing all of their active modifier machinery (dCas9-SunTag + Catalytic Domain-scFV + Doxycycline) but using non-targeting sgRNAs and/or sgRNAs that target to an inert region of the genome (e.g. Rosa26). It is very clear from the data presented in Figure 1D through 1I and 1K that several of the modifiers used have off-target effects. For instance, in Fig. 1I it is apparent that the Setd2 construct implements H3K36me3 at the off target *Zic4* and *HoxD13* loci. It is also clear from figure 1K that the p300 construct induces massive transcriptional changes. The *Ring1b* and *Prdm9* construct also result in widespread changes in gene expression, although the authors downplay point this in the manuscript.

In addition to performing experiments using non-target or off-target sgRNAs as controls, the authors must quantify genome-wide off-target activity of their constructs by performing normalized ChIP-seq using *Drosophila* chromatin spike in. Throughout the paper the authors use CUT&RUN followed by qPCR for selected loci. This is an atypical approach as ESCs can be easily proliferated to cell numbers that allow for ChIP-seq which provides much more reliable quantification. It should also be noted that the authors have not included any type of spike in normalization in their CUT&RUN experiments to control for efficiency and specificity. In addition, at least 2 of the antibodies that they use (H3K9me3 and H3K27ac) failed quality control testing in a large scale assay for antibody specificity (<https://chromatinantibodies.com/antibodies/e0531-5> and <https://chromatinantibodies.com/antibodies/e0512-1>). Performing a genome-wide quantitative ChIP-seq for the modifications implemented by each of the modifier constructs in the context of both on-target and

off-target sgRNAs is of crucial importance to the interpretation to all of the authors' experiments and is a standard in the chromatin field for measuring histone modification levels.

R4: In this study we have used a reductionist approach to determine how a specific chromatin modification at a specific *cis*-DNA sequence, within a specific genomic locus causally impacts transcription. We have applied this approach at scale (e.g. by systematically altering the underlying DNA sequence, locus, or chromatin mark) to discern general principles using precision strategies. Because we are employing a locus-specific readout and a locus-specific perturbation, we attempted to design the maximum number of controls at the locus to ensure robust conclusions. This includes (i) targeting each locus with a catalytic point-mutant effector that recapitulates all aspects of the system except deposition of the mark, (ii) recruiting a non-functional GFP effector, (iii) utilising exactly the same cell line without inducing the system (-DOX), and (iv) confirming the system has little, if any, off-target effects. The reviewer is suggesting that our results could be a consequence of the epigenetic editing system affecting an OFF-target secondary locus, which feeds back to exert specific effects on our targeted reporter *and* the various tested endogenous genes, across marks. Whilst if this were the case, it would still support regulatory functions for each chromatin mark, several lines of original - and newly generated - evidence argue against this scenario:

- 1.) We tested the OFF-targeting potential of each active CD^{scFV} effector by quantitative Cut&Run-qPCR for the cognate chromatin mark and observed **negligible impact at non-targeted (negative) regions** examined (*see Fig 1D-I*). Moreover, it is also clear that regions adjacent to the targeted site (>5kb) exhibit no statistically significant enrichment of each mark, further arguing against general OFF-targeting (*see Fig 1D-I*). The exception was Setd2-CD^{scFV} (depositing H3K36me3), which we point out. Notably, despite clear off-targeting of this specific effector, there was almost no transcriptional effect on the reporter suggesting any secondary effects are negligible for our reductionist strategy. Of note, for quantitative analysis of a specific locus, qPCR is preferable to sequencing since it provides better data depth at that locus, and is less susceptible to mapping and normalisation artefacts.
- 2.) We performed RNA-seq analysis in all cell lines to examine the general effect of over-expressing each CD^{scFV} effector. In all cases but one we found a high level of transcriptome concordance with control that targeted inert GFP^{scFV} (R>0.98) (*see Fig 1K*), with 99.9% of global genes unaltered. This is in line with the variation found between technical replicates of the same cultured cell lines. The lack of widespread DEGs upon induction of epigenetic editing extends to effectors with high transcriptional impact, such as G9a-CD^{scFV} (n=1 DEG; p<0.05 FC>2), **implying there is minimal intrinsic OFF-targeting** within the overall system design. As the reviewer points out, and as is made clear in the text, the p300-CD^{scFV} effector does however have a major general impact on the transcriptome, with a relatively small but discernible change also for Ring1b-CD^{scFV} and Prdm9-CD^{scFV}. This however is not responsible for the ON-target changes we observe (*see point #3 below*).

- 3.) Our study identified several endogenous silent genes that exhibited strong activation in response to ON-target *de novo* H3K4me3 epigenome editing, including *Fgf5* (>10-fold) and *Cldn16* (>100-fold) (see Fig 3D). We also identified genes such as *Calb2* and *Setmar* that are (re)activated by epigenome editing with a targeting gRNA (see Fig 3C).

We now show below in new analysis that when we induce effectors (+DOX) but with a gRNA against *Hbby*, **none of these genes exhibit activation**, as would be expected if “*off-target effects caused by overexpression of histone modifier catalytic domains*” were responsible for our observation. This data is shown specifically for *Prdm9* (left), and for all effectors (right). This strongly argues the response of target genes in the study is linked to ON-target epigenetic editing, since without a specific gRNA, no effect is detected, even in the presence of overexpressed effector domains. The specificity of response is further confirmed by catalytic-mutant and -DOX controls.

further investigate the reviewer concerns regarding

Testing OFF-target gene responses from the epigenetic editing system, using a *Hbby* gRNA. Left. Scatter plot showing expression of genes that respond to ON-target H3K4me3 editing (see Fig 3C-D) when Prdm9 is induced with a non-targeting *Hbby* gRNA. No major responses are observed. Right. Fold-change of the same responsive geneset shown left, when any of the CD^{scFV} effectors are induced with a non-targeting gRNA.

OFF-target effects, particularly around H3K4me3, we have now performed additional experiments with multiplexed gRNA and CUT&RUN:

- 4.) As suggested by the reviewer we performed a quantitative global analysis of H3K4me3 using spike-in normalisation following induction of *Prdm9*-CD^{scFV}. We observed 6 *de novo* (OFF-target) H3K4me3 peaks out of >900,000 genome-wide tiles upon *Prdm9*-CD^{scFV} expression, with no peaks overlapping a regulatory region. We further detected clear ON-target deposition at the targeted region (*Hbby*) (see new Extended Data Fig S2B, also shown right). This supports the previous qPCR-based analysis and transcriptome response (see #1-3 above). Given that we observe no transcriptional response at the target genes for catalytic mutant *Prdm9*-CD^{scFV}, and that OFF-target peaks are rare with *Prdm9*-CD^{scFV}, this suggests that the ON-target effects are direct.

Testing ON- and OFF-target H3K4me3 deposition following *Prdm9*-CD^{scFV} induction.

Left. Scatter plot showing *de novo* H3K4me3 peaks genome-wide. Six off-target peaks (in red) are detected in addition to the ON-target *Hbby* peak (in green) amongst >900,000 tested genomic tiles using calibrated CUT&RUN-seq. Right. Genome view of the *Hbby* locus, showing a clear *de novo* H3K4me3 peak around the

- 5.) To functionally interrogate potential OFF-targeting of *Prdm9*, we introduced a pooled mini-library of 39 gRNA into ESC. These targeted the H3K4me3-responsive gene *Cldn16*, as well as eleven other control gene promoters in triplicate. We then induced *Prdm9*-CD^{scFV} to target H3K4me3, and assayed effects by via single-cell expression. We de-multiplexed cells according to which gRNA was (randomly) present in each cell and assessed transcriptional response of *Cldn16* in each scenario. In cells that received *Cldn16* gRNA we observed strong activation, independently confirming H3K4me3-mediated effects (see Figure, right). In contrast and importantly, *Cldn16* was undetectable with 33/36 gRNA targeting other promoters (i.e. not activated), and only weakly detectable in the remaining. These data strongly argue against OFF-target or secondary effects of *Prdm9*-CD^{scFV} induction underlying the ON-target responses.

Taken together, cumulative lines of evidence support the notion the transcriptional responses primarily reflect ON-target epigenetic editing. Whilst we cannot completely rule out indirect influences and/or OFF-targeting, we find little evidence to support this as a major underlying mediator. We conclude our precision strategy demonstrates the impact of a chromatin mark at the targeted locus.

As a side note, the reviewers' point about antibody specificity needs an additional clarification. Of the ten antibodies against the primary modifications used in this study, the reviewer has chosen to point out that two failed one part of several specificity tests. The first of these was for H3K9me3, which *marginally* also detects H3K9me2 above background in 1 of 2 assays, but no other modifications. Since our effector triggers deposition of both these marks, which we point out (see Fig 2D), this minor effect has no bearing on the interpretation of data on H3K9 methylations. The second antibody against H3K27ac, also *marginally* detects other hydrophobic acylations on H3K27, such as butylation. Given our effector does not catalyse this mark, that the results of H3K27ac programming are broadly in line with expectations, and that this antibody is the most predominantly reagent used in the field for H3K27ac, we do not feel this has an impact on the conclusions.

C5: The experiments used to make the claim that H3K4me3 is instructive for transcription activation are not convincing. Many of the genes that the authors claim to be downregulated in the MLL2-CM/CM cells are expressed at extremely low levels in wildtype ESCs. The authors should provide RNA-sequencing tracks for the changes that they are observing as opposed to normalized qPCR data, which can be misleading when quantifying genes that are expressed at almost undetectable levels. Since these genes are barely expressed to begin with it's difficult to know how to interpret them being further downregulated. Given the extremely low expression of the genes shown in Figure 3B and Supplemental Figure 6A, I'm concerned that what they're looking at is random experimental variation due to either tissue culture, or during the multiple steps involved in RT qPCR. The authors should demonstrate they can rescue the expression of genes that are expressed at convincing levels (by RNA-seq) and downregulated upon mutation of MLL2. One of the well-established targets of MLL2 in ESCs is the *Magohb* gene (Glaser et al., 2009. Epigenetics and Chromatin; Ladopoulos et al., 2013. Mol. Cell Biol). *Magohb* has been shown by several groups to be convincingly downregulated upon mutation of MLL2, therefore the authors should test if they can rescue *Magohb* expression with their constructs.

R5: The central reviewer concern is that genes downregulated in our genetic LoF model - ESC with homozygous catalytic point-mutations in *Mll2* – are expressed at low levels in WT cells and thus potentially subject to experimental variation. We now provide new representative genome tracks demonstrating that the expression of these H3K4me3-dependent genes in WT is rigorously and reproducibly detected across replicates (see below examples). Indeed, the goal of statistical analysis of independent biological replicates is to ensure results do not represent “*random experimental variation*”, which applies to our data (all (adjusted) $p < 0.05$; see Fig 3A-3D, 3F). Moreover, the reviewer points out that *Magohb* is a well-established target. It is therefore noteworthy that the majority of H3K4me3-dependent genes (*Mll2*^{CM/CM}) that we identify and ‘rescue’ (see Fig 3C) are actually expressed at equivalent or higher levels than *Magohb* in our naïve ESC lines (freshly derived, low passage FVB in 2i/L), as shown in examples below - see scale in WT tracks. **This highlights that the expression effects we report intrinsically reflect “convincing levels”, as judged by the reviewer’s own benchmark (cont overleaf).**

It may also be noteworthy that *Magohb* is amongst the significant downregulated genes in *Mll2^{CM/CM}* ESC, confirming that H3K4me3 *per se* plays a role in its regulation as expected. However, the downregulation is only ~2-fold, far less than in *Mll2* knockout models previously deployed, suggesting this particular gene may depend in part on MLL2 protein itself in naïve ESC. This could be important for studies that have used *Magohb* as a reference gene to draw conclusions on H3K4me3-

Examples of genes that lose H3K4me3 and are downregulated in *Mll2^{CM/CM}* ESC consistently over replicates.

RNA-seq and CUT&RUN-seq tracks showing gene expression and H3K4me3 enrichment at *Dazl*, *Ttl4* and *Magohb* gene

based regulation.

C6: The authors use the Prdm9 PR/SET domain to implement H3K4me3 in most of their experiments. This is a strange choice because Prdm9 is an atypical H3K4me3 methyltransferase that functions specifically during meiosis in mammals to direct DNA recombination machinery to recombination “hotspot” sequences. In addition to H3K4me3, Prdm9 also catalyzes H3K36me3 (Powers et al., 2016. Nature Genetics; Huang et al., 2020. Elife; Wells et al., 2020.) which is a highly unusual feature and greatly complicates the authors’ experiments. The authors do perform some experiments using the SET domain of Setd1a, however, the results they observe are inconsistent with the properties of Prdm9. For instance, Prdm9 induces a ~500 fold induction of *Cldn6*, whereas Setd1a causes a ~6 fold induction, similarly, Prdm9 induces a 10 fold activation of *Fgf5*, whereas Setd1a does not cause any statistically significant increase in expression. Are these differences due to different levels of H3K4me3 implementation by these 2 enzymes? In the current version of the manuscript the authors have not performed any experiments to look at potential differences in the levels of H3K4me3 deposited by Prdm9 and Setd1a at their target genes. The authors should perform Drosophila chromatin spike in normalized ChIP-seq for H3K4me3 for the sgRNAs targeting all of the genes in figure 2B,C,E for both the Prdm9 and Setd1a expressing constructs to quantitatively assess the levels of H3K4me3 implemented at these sites. This should also be done for the genes shown in Supplemental Figure 6A. This is an essential control

experiment to determine 1) if H3K4me3 is actually being deposited at these targets and 2) if the levels of H3K4me3 correlate with the amount of gene activation. The authors must also test if Prdm9 is implementing H3K36me3 at their target genes using the same approach outlined in the previous paragraph. This is an essential experiment because if Prdm9 is implementing both modifications the authors cannot conclude that it is H3K4me3 that is driving activation. Since the authors are using MLL2 (KMT2B) mutant cells it would make sense to use the KMT2B SET domain in the H3K4me3 rescue experiments. As I mentioned in my summary statement this work would benefit from using a comprehensive panel of modifiers and also include KMT2C (MLL3) or KMT2D (MLL4) which implement H3K4me1 to study the effects of different H3K4 methylation states.

R6: As noted in previous responses, we designed and tested a large repertoire of epigenetic effectors (*see Table 1*, and updated methods). This includes three independent H3K4me3 catalytic domains (CD) derived from *Prdm9*, *Setd1a*, and *Mll2*. We found Prdm9-CD^{scFV} to be highly effective at deposition (typically >10-fold enrichment), Setd1a-CD^{scFV} to be moderately effective (~5-fold), and Mll2-CD^{scFV} to only deposit very low levels (<2-fold). The *Mll2* system therefore did not pass our QC steps.

We proceeded with two independent H3K4me3 effectors, and showed that both are capable of triggering transcription upregulation of reporters and multiple endogenous genes. Catalytic point-mutants for either led to complete loss of transcriptional responses. This provides independent lines of evidence that programming H3K4me3 *per se* triggers transcriptional changes, at least at a subset of loci, and that responses are not dependent on the identity/function of any one effector. The reviewer queries why the transcription responses were quantitatively different for *Prdm9-CD^{scFV}* and *Setd1a-CD^{scFV}* (e.g. *Cldn16* = >100-fold and 6-fold respectively). As shown in the current manuscript we demonstrate that **the level of transcriptional response is correlated with the quantitative level of H3K4me3 deposition**, revealing a dose-dependent effect (*see Fig 3E & Extended data Fig 7D*). This thus establishes a H3K4me3 titration experiment, which further supports our conclusions that H3K4me3 can trigger activation.

To extend this further and address the request to include an Mll2 effector, we have now targeted Mll2-CD^{scFV} and its catalytic mutant, and read-out single-cell transcription effects. This **new** data reveals a small but discernible transcription response that is well correlated with the low level of H3K4me3 programming, and which is not detected in the mutant version. The entire data series is shown right to highlight the exquisite dose-dependency of H3K4me3 impact.

H3K4me3 triggers transcriptional changes in a dose-dependent manner.

Dot plots showing expression at the OFF reporter after targeting with distinct H3K4me3 effectors Prdm9^{scFV} (left) or Set1a^{scFV} (center) and Mll2^{scFV} (right). The level of transcriptional upregulation correlates with the amount of epigenetic

Taken together, we show (i) independent H3K4me3 effectors can instruct transcriptional responses, (ii) that this is dose-dependent, (iii) that specific loss of H3K4me3 *per se* impairs transcriptional activation (*Mll2^{CM/CM}*) and, (iv) that programming H3K4me3 back to these promoters robustly rescues their expression defect. Taken with recent complementary loss of function reports that H3K4me3 *per se* is linked with RNA Pol2 pause-release control (Wang et al., 2023), this provides compelling lines of evidence that the H3K4me3 mark itself can contribute to modulating transcription level from at least some target promoters.

C7: Many of the experiments are done after 7 days of expression of the modifier constructs. This seems like an extremely long time and raises strong concerns regarding off-target and secondary effects. The authors should perform a time course to detect how quickly modifications are implemented. Particularly for H3K4me3, if this modification is truly instructive for transcription you would expect to see increased gene expression soon after H3K4me3 appears. If there are several days of lag between appearance of the modification and gene activation, it is difficult to imagine how this is a direct causal effect.

R7: In this study we systematically tested the transcriptional responses to epigenetic editing at both early (**day 2**) and later (**day 7**) timepoints. This information is available to the reviewer within the original manuscript for each programmed chromatin modification, and across multiple *cis* contexts (see Figure 4C & Figure 6A). In brief we found that the impact at day 2 was coherent with day 7, and for most modifications - including H3K4me3 - the magnitude of effect was almost indistinguishable at the early and late timepoints. Thus, we do observe “*increased gene expression soon after H3K4me3 appears*” further ruling out secondary effects, as enquired by the reviewer. An interesting exception is H2AK119ub, whereby the effect at day 7 was notably stronger than day 2, implying a progressive acquisition of PRC1 repressive activity and/or deposition. The reviewer makes a good point and to make these results clearer, we have now added an entirely new Extended Data Fig 4, which shows the temporal dynamics of transcriptional responses to each modification at single-cell resolution.

C8: The authors do not provide any mechanistic insight into how H3K4me3 might lead to gene activation. The authors should explore the biochemical mechanism through which activation might occur.

R8: Whilst the study is a broad dissection of the causal impact of many chromatin modifications across *cis* contexts, we chose to explore the mechanistic basis of a number of interesting and novel observations that arose. Amongst these were experiments that explore the mechanism through which H3K4me3 elicits a functional effect, and thus we do not recognise the reviewer comment as an issue. Specifically, we find that (i) precision programming of H3K4me3 to a variety of promoters instigates transcription upregulation (*see Fig 3C-F*) (ii) this is directly linked with downstream recruitment of H3K27ac (and loss of H3K27me3) specifically from this locus, indicating H3K4me3 itself can remodel the local epigenomic landscape (*see Fig 3G*), and (iii) that functionally blocking this H3K27ac recruitment results in a complete block of the downstream transcription upregulation, without affecting H3K4me3 deposition (*see Fig 3H*). We further show that H3K4me3 interacts with repressive epigenetic pathways via genetically encoded motifs to tune transcription responses (*see Fig 6A, D-G*). Taken together with evidence that H3K4me3 may recruit the pre-initiation complex (Vermeulen et al., 2007) and impacts RNA Pol2 pause-release (Wang *et al.*, 2023), our functional experiments provide evidence for “*the biochemical mechanism through which activation might occur*”. Whilst we do not claim to have deciphered fully the underlying mechanisms, we discuss our evidence and broader possibilities extensively in the discussion. For example, we suggest that *de novo* deposition of H3K4me3 likely overcomes repressive epigenetic systems, since we observe a co-occurring loss of H3K27me3. Such a scenario is consistent with recent publications (e.g. *see (Douillet et al., 2020)*). Given the extensive data and compound insights already included in the manuscript, and initial mechanistic observations, the full dissection of H3K4me3 mechanism is warranted in future.

C9: For all experiments claiming changes in RNA levels the authors must determine if this is due to increased transcription or increased RNA stability. This could be performed through PRO-seq or alternatively RNA Polymerase II ChIP-seq.

R9: A number of independent lines of evidence support the conclusion that the effects we observe reflect transcription regulation. (1) We performed ATAC-seq following deposition of each modification. We observe promoter accessibility changes are highly correlated with transcription changes elicited by each mark (either up- or down- regulation). This strongly implies regulation at the transcription level (*see Fig S5A*). (2) Secondly, in all experiments ESC carry both a BFP reporter (that marks gRNA activity) as well as the epigenetically-targeted mCherry2 reporter. Despite the BFP sequence having only a few nucleotide changes relative to mCherry2, its expression is unaltered in contexts where mCherry is either up- or down-regulated due to locus-specific epigenetic editing. Effects on RNA stability of the reporter sequence would

be predicted to affect both transcripts, whereas transcriptional difference would only impact mCherry2, as observed. (3) We report differential responses to the same chromatin mark whereby the coding RNA remains identical, but the upstream (non-transcribed) *cis* genetic sequence is manipulated. This strongly implies that regulation is at the level of transcription rather than RNA. Nevertheless, to address the reviewer concern a line is included in the discussion raising the possibility that chromatin modifications operate across many regulatory layers. “Thus, whilst our data imply that chromatin marks have the potential to causally instruct transcription programmes, they also highlight this may represent one regulatory layer within multiple nonlinear governing mechanisms.”

C9: The authors should characterize the proteins that associate with each of their modifier constructs by performing immunoprecipitation followed by mass spectrometry. This is important because in addition to possessing catalytic activity, the catalytic domains may also mediate protein-protein interactions and recruit co-regulators that influence gene expression. This should be done for both the wildtype as well as the catalytic mutants because the catalytic mutations may disrupt protein folding and lead to loss of interactions. The authors should provide western blot experiments demonstrating similar levels of expression of the wildtype and catalytic mutants.

R9: All effector domains (CD^{scFV}) used throughout this study were coupled to superfolder GFP (sfGFP) to enhance their correct folding, to track their dynamic expression, and to purify cells based on the activity of the epigenetic editing system at the protein level. We can also exploit this to *quantitatively* compare the expression/stability of each induced WT and mutant effector protein, at single-cell resolution. We found mutant CD^{scFV} effectors to be present at comparable levels to their WT counterparts at the protein level, with strong levels of induction relative to -DOX conditions (see new Extended data Fig 1C, also right). The suggestion to perform mass spectrometry for all effectors is beyond the scope of this study, represents years of work, and in our

Levels of induced WT- and mut- CDscFV epigenetic effectors at the protein level, confirming their stability and relative expression. Bar plots showing the expression level of each WT and corresponding catalytic mutant effector, relative to -DOX conditions at the protein level. Expression correlates with intensity of tagged sfGFP coupled to each effector and measured by means of quantitative single-cell flow cytometry.

opinion is not necessary to make robust conclusions.

C10: I was unable to find citations for the source of all the catalytic dead mutations in this paper. I did a web search as was able to find that some had been previously reported, but for others such as Prdm9, I could not find publications characterizing these mutants. The authors should cite the primary literature that characterizes these mutations.

R10: We now provide the citations in the methods.

C11: For the experiments presented in Figure 4 involving the introduction of different transcription factor binding motifs into their reporter construct, the authors must demonstrate that the transcription factors of interest are actually binding to the motifs that they have introduced into their reporter by performing ChIP-seq. It is generally accepted that only a fraction of consensus TF motifs in chromatin are actually bound by their cognate TF, potentially due to variations in chromatin accessibility. The authors have made the assumption that simply by inserting TF-motifs that they are bound, however, this must be tested experimentally by ChIP-seq. In this case, *Drosophila* chromatin spike in is not absolutely necessary because unlike histone modifications, TF epitopes are likely not conserved. However, the authors could perform spike in using a kit such as the Active Motif spike-in normalization kit to normalize overall ChIP efficiency. Also, in the case of motifs that can be bound by multiple proteins of a TF class (E-box, GATA, OTX) it is essential to identify specifically which TFs are bound to which TF-motifs.

R11: The aim of the experiments in Fig 4 is to investigate the functional relationship between a specific chromatin mark and the underlying DNA sequence. We therefore systematically introduced sequence motifs/variants associated with structural effects (e.g. G-quadruplex) and/or transcription factors PWMs and examined the impact on chromatin mark function. Because many TF within a family can bind the same TF motif, and because we are fundamentally interested in the principles of a *cis*-genetic x epigenetic interaction, we did not focus on the TF themselves but rather on understanding the quantitative effects of different DNA sequences. Indeed, we do not claim our observations are due to a specific TF binding, in part because multiple TF can bind the same motif, and in part because binding *per se* does not indicate functional relevance. Instead we chose to perform functional experiments on selected motifs by genetically perturbing the candidate TFs. For example, to investigate the interaction between E-box motifs and H3K4me3 function, we deleted *Pcgf6*, a factor known to bind this motif. This enabled us to identify a

genetically-encoded interaction between *Pcgf6* (PRC1) and H3K4me3 with a quantitative impact. We have now extended this with additional data to investigate the interaction between CTCF DNA motifs and H3K36me3 (see entirely new Fig 5) by deleting endogenous sites known to attract CTCF occupancy along with compound *Setd2* perturbation, and observing striking effects (see R13 below for full discussion). In summary, we do not make “*the assumption that simply by inserting TF-motifs that they are bound*”. Instead the study is focused on understanding the functional interactions between *DNA sequence* and *epigenetic marks*, rather than the specific TF (or many) that bind each sequence. We now make this point more clearly in the discussion and text.

C12: The authors should also perform quantitative ChIP-seq to measure the amount of each histone modification that is deposited on the various TF-motif containing constructs. Do the alterations in gene expression that they observe correlate with the levels of histone modifications that are implemented? This has been done for H3K36me3 in figure 4H, but should also be performed for all the TF-motif modifier combinations for which the authors claim to see an effect on gene expression (e.g. H2AK119 and YY1 motif; H3K9me3 and YY1 motif).

R12: As pointed out by the reviewer, we have performed this experiment on the most interesting interactions to investigate whether *cis*-context impacts initial deposition (see Fig 5B-D). In these cases, we found no difference in deposition across motifs suggesting the sequence motif triggers a functional interaction rather than impediment to programming the histone modification (see Fig 5B). Moreover, we have now extended this by performing considerable additional experiments which highlight this interaction is functionally relevant at endogenous loci (see new Fig 5 and R13 below). To perform replicate Cut&Run for all marks across all the *cis* contexts is an experiment on a scale that is not feasible. Moreover, any result would not change the conclusion. For example, if indeed we were to find lower levels of chromatin mark ‘X’ installation in the context of TF motif ‘Y’, this could reflect either lower activity of editing in this context, or that an equally-deposited mark is more rapidly turned-over/antagonised within a specific context. The latter option represents real biology but cannot, in the proposed experiment, be distinguished from a technical effect of differential initial deposition, thus negating any further conclusions.

C13: I found the most interesting result in this study to be the ability of CTCF motifs to strongly enhance transcriptional repression by H3K36me3. However, the authors do not explore the mechanism of how this

occurs. H3K36me3 is generally associated with the gene bodies of actively transcribed genes and studies in yeast suggest that it may function to suppress cryptic intra-genic transcription (although this function may not be conserved in mammals). So the ability of H3K36me3 to suppress transcription is interesting but it is completely unexplained in the authors work and how this links to CTCF is not examined mechanistically. As stated above, the authors must experimentally demonstrate that CTCF is actually binding to their CTCF construct using ChIP-seq (looking at Cohesin binding would also be informative). Unlike the effects of H3K4me3 on gene activation which seem to be modest at best and not reproducible across multiple loci, the combination of H3K36me3 and CTCF does appear to induce strong silencing of their reporter construct. An interesting experiment would be to identify an endogenous region with multiple CTCF sites and attempt to silence it by targeting *Setd2*/H3K36me3. If this is successful, then knock-out the CTCF sites and demonstrate a reduction in repression. Since H3K36me3 is associated with active gene bodies, what happens when the authors target it to a gene body instead of a promoter?

R13: We are glad that the reviewer recognises the potential interest around a functional interaction between H3K36me3 and CTCF motifs. We believe this observation exemplifies the discovery power of precision epigenetic editing intersected with a reductionist approach, as also deployed for H3K4me3. The reviewer asks whether the impact of *promoter* H3K36me3 is relevant at endogenous genes, since as correctly stated, H3K36me3 is typically preferentially found within gene bodies, in most cell types. We agree that this is important to follow up and have now performed a series of experiments to investigate endogenous settings using complex triple (epi)genetic perturbations strategies. These data now form the basis of entirely new Fig 5 in the revised manuscript, and are summarised below.

- 1.) We investigated the relevance of CTCF motif orientation for the functional interaction with H3K36me3, given the known importance of motif arrangement for 3D looping. We engineered multiple knock-in ESC lines with different CTCF motif orientations, and observed H3K36me3 can repress all of them. This suggests the interaction is CTCF motif orientation-independent (*see Fig 5C*).
- 2.) We next freshly derived (low-passage) homozygous *Setd2*^{-/-} ESC, which largely lack H3K36me3. We performed transcriptome analysis identifying 1,671 differentially expressed genes, of which the majority were upregulated (*see Fig 5F*).
- 3.) We then generated H3K36me3 Cut&Run-seq profiles and searched for genes that exhibited a high-confidence ‘peak’ of H3K36me3 at their promoter in WT ESC. As expected, there were a relatively small number, but their identification nonetheless demonstrates that H3K36me3 can, and does, occur at endogenous gene promoters.
- 4.) We intersected the promoter H3K36me3-marked genes with genes that were upregulated in *Setd2*^{-/-} cells. The top hit was *Xist*, the master regulator of X-inactivation. *Xist* exhibits a clear promoter H3K36me3 peak, and is amongst the top 5 de-repressed genes upon loss of H3K36me3 (*see Fig 5G*).
- 5.) Using *Xist* as a paradigm, we first asked whether targeting H3K36me3 back to the promoter in *Setd2*^{-/-} ESC, via epigenetic editing, reimposes silencing. We found an almost complete rescue,

strongly suggesting that promoter H3K36me3 is responsible for silencing this key endogenous gene (see Fig 5H-I).

- 6.) The *Xist* promoter region (10kb) is associated with several CTCF motifs, including one directly adjacent to the TSS. To investigate whether the CTCF site interacts with H3K36me3 to mediate silencing, we homozygously deleted the motif in *Setd2*^{-/-} ESC. Remarkably, when we programme H3K36me3 back to *Xist* without the CTCF site, there is no detectable silencing re-imposed. In contrast, there is strong silencing with the CTCF motif (see #5) (see Fig 5J).

Thus, using a compound triple genetic-perturbation strategy (*Setd2* knockout - CTCF motif knockout - induced H3K36me3 epigenetic editing), we extend the observations we made using a synthetic system to show they are directly relevant for endogenous loci. Indeed, the H3K36me3/CTCF interplay proved to be a regulatory mechanism for the master regulator gene of a critical developmental process (X-inactivation), suggesting it has major physiological relevance. Furthermore, H3K36me3 is found overlying many gene promoters in cell types aside from ESC, such as extraembryonic tissues, where its functional interaction with CTCF motifs may have even broader relevance, as well as at many potentially cryptic promoters in gene bodies. We thank the reviewer for suggesting this important line of experiments.

C14: The authors should provide western blots confirming lack of cMyc and PCGF6 protein in their knockout cells. Also does knockout of cMyc cause general transcriptional changes in ESCs? If so the effects the authors observe could be secondary. The use of an acute depletion model (Auxin-AID or dTag) would be more mechanistically informative.

R14: To confirm the genotypes of mutant (knockout) ESC clones we took three complementary approaches. (i) Our knockout strategy is to remove an entire critical 5' exon, leading a phase-shift and multiple stop codons early in the transcript. This is the strategy as recommended by KOMP and generates a null allele. Moreover, this is a published and validated strategy for *Pcgf6* KO (Zhao et al., 2017). We confirmed homozygous exon deletion unambiguously by PCR amplification from flanking introns revealing the expected reduced size upon exon deletion. (ii) We sequenced across the mutant alleles to confirm the precise mutations and homozygosity. (iii) To functionally confirm null status, we examined the effect on target gene expression in mutant cells. For example, PRC1 mutants have previously been linked with upregulation of *Slc25a31*, *Tcam1*, and *Tdrkh*, and we found robust

Confirmation of *Pcgf6* mutant clone genotype.

(Left) PCR amplification of a *Pcgf6* critical 5' exon in WT ESC and single clones transfected with wtCas9 and two targeting gRNAs. PCR shows the expected product size in *Pcgf6* KO clones. (Right) Bar plots showing expression of selected PRC1 target genes in WT and *Pcgf6* KO cells. (Bottom) Sanger sequencing views confirming the presence of large genomic

upregulation of these in our mutant lines.

C15: The experiments in this paper are performed entirely in mouse embryonic stem cells (mESCs), which greatly limits the relevance of the authors' findings as ESCs have a number of atypical features (lack of several cell cycle checkpoints, aberrant maintenance of DNA methylation particularly at imprinted genes). The generation of mice harboring the authors' chromatin editing constructs and performing experiments in real tissues would greatly enhance the impact of this study. If the authors can rescue an *in vivo* animal phenotype caused by lack of a histone modifier (for example PRC2 deficiency) I feel that would bring the paper to the level where it could be considered by a top-tier journal.

R15: Generating transgenic mouse models *de novo* for each epigenetic editing system (+targeting gRNAs) is beyond the scope of this manuscript, which already contains vast experimental data, and would mean at least 2-4 years revision, as surely the reviewer well understands. Whilst we perform our experiments

primarily in ESC, we make this clear, and point out future directions with other cell types. Furthermore, epigenetic editing enables locus-specific precision changes in chromatin modifications. This is extremely powerful to dissect functional and mechanistic relationships. It is not however clear that the suggestion to “*rescue an in vivo animal phenotype caused by lack of a histone modifier (for example PRC2 deficiency)*” with such a tool is a viable suggestion, given the pleiotropic and genome-wide nature of the defects caused by PRC2 deficiency. The reviewer is suggesting that the phenotypes in chromatin-modifying mutants (e.g. PRC2 deficiency) may be rescued by modulating just a few target loci. Whilst this is unlikely (albeit possible), it is entirely opaque which loci these might be in any given cell/tissue/age mouse and how this would relate to our core conclusions.

C16: Recent experiments using histone H3K27 point mutants which completely ablate both H3K27me3 and H3K27ac (McKay et al., Sankar et al., 2015. Developmental Cell; 2022. Nature Genetics) have demonstrated that although H3K27me3 is essential for PRC2 mediated silencing, H3K27ac is dispensable for gene activation. This suggests that H3K27ac is likely not the crucial substrate for CBP/p300 which are known to acetylate a wide range of both histone and non-histone proteins (Weinert et al., 2018. Cell). Therefore, the authors should examine what other alterations in histone acetylation patterns are induced by their CBP modifier construct by ChIP-seq. An extremely informative experiment would be to obtain the H3K27R complete mutant ESCs generated by Sankar et al. and then test the patterns of both gene expression through RNA-seq and various sites of histone acetylation (e.g. H3K18ac, H2B-N-terminal acetylation, see Weinert et al. for validated CBP/p300 histone substrates) genome-wide by ChIP-seq.

R16: It is now well established that p300 catalyses acetylation at several histone residues (Weinert et al., 2018), as cited in the manuscript. In discussing the functional impacts of epigenetic editing by p300-CD^{scFV}, we therefore refer to ‘histone acetylation’ generally, without reference to a specific residue. As a readout to confirm the activity of epigenetic editing by p300-CD^{scFV} we use H3K27ac, since this is the most well characterised of its acetylation sites. Thus, our data reveals the functional impact of locus-specific chromatin acetylation, at a quantitative and single-cell level.

C17: I am not convinced that combined effects of H3AK119ub and H3K27me3 are synergistic as the authors claim, the effect seems to simply be additive based on the data presented.

R17: In terms of the fraction of cells that mediate *complete* silencing, the combined effects of H2AK119ub and H3K27me3 produce 41%, whereas 27% and 7%, are found with singleplex marks. Because the combined epigenetic effect surpasses the percentage of adding the two single marks, we considered the impact synergistic rather than additive. However, we do recognise that this distinction of additive vs synergistic is not clear cut. We have therefore softened the manuscript in places to highlight this possibility, in line with the reviewer's comment. For example, line 477: "*with H3K27me3 and H2AK119ub together exemplifying effects that are at least additive, and potentially synergistic.*"

Reviewer#3:

C1: The use of fusion proteins to catalytically dead versions of CRISPR CAS proteins now makes it possible to place specific epigenetic marks at defined locations in the human genome. To date, such programmed epigenetic editors have been used primarily as a means to control gene expression rather than as tools to dissect the mechanisms underlying chromatin-based transcriptional regulation. In addition, only a small set of protein domains have been successfully engineered as epigenetic editors (DNA methyltransferase and KRAB domains being the most widely implemented so far).

Policarpi and colleagues fill both of these gaps in the literature and in doing so make an important contribution to the field of epigenetic editing. They build an impressive repertoire of new histone mark modulators and deploy them to understand poorly characterized and/or controversial aspects of chromatin-mediated transcriptional control. Their work on H3K4me3 is particularly thorough, including the identification of dose-dependent H3K4me3-driven gene activation and a clear link between H3K4me3 activation and promoter acetylation. Other important findings relate to the relationship between specific DNA motifs and the transcriptional impact of histone modifications made possible by their elegant allelic series experiments. The identified association between CTCF motifs and enhanced H3K36me3 silencing is striking, as is the uncovered role of PRC1.6 recruitment to MYC motifs in limiting H3K4me3 activation. Overall, the manuscript offers both a valuable set of tools for the community as well as impactful new insights into chromatin biology. As such the manuscript has the potential to make a strong addition to Nature Genetics. Below are a number of suggestions and concerns that should be addressed either through textual changes and/or experiments prior to publication.

R1: We thank the reviewer for both their time spent in critiquing our manuscript and their constructive comments and suggestions. We further note that the reviewer is balanced in placing our manuscript within the broader scientific context and suggests the “*work on H3K4me3 is particularly thorough*”.

Major Points

C2: The epigenetic editing platform the authors developed has the potential to be broadly enabling for the community but the systems would greatly benefit by more careful delineation of its limitations. Below are some specific examples.

R2: We fully take on board the reviewer’s suggestion and have attempted to increase the information available to the reader and community in both the revised manuscript and additional Extended data, for example adding information on >30 catalytic cores that we tested and optimised for epigenetic editing. We further address each subpoint below:

C2.2: Toxicity. There is no mention of whether expressing any of the epigenetic editors is toxic to cells.

R2.2: We observed no toxicity upon induction of the epigenetic editing system (+DOX), as judged by cell survival and proliferation rates. The major exception was p300-CD^{scFV}, where strong induction had a clear impact on cell proliferation and survival (*see right panel*). This is consistent with our observation that exceptionally, this effector had indirect effects on global transcription profiles. To mitigate against both the toxicity and OFF-targeting, we used reduced levels of DOX to minimise levels specifically of the p300-CD^{scFV} effector. Whilst we allude to this in the text, we have now made it clearer that there is also a toxicity effect for p300-CD^{scFV}, and added data demonstrating that this can be rescued by titrating the level of DOX-induction (*see new Extended Data Fig 2C-D*)

DOX titration mitigates p300-CD^{scFV} toxicity effects,

Cell count of p300-CD^{scFV}-transfected ESC either left untreated (No Dox) or treated with increasing amounts of DOX for either one (d1), two (d2) or three (d3) days. While higher concentration of DOX are toxic, 5ng/ml dosage largely preserves cell viability.

C2.3: Off-target effects. While the authors carry out RNA-seq experiments to show that transcriptional off-targets are rare (with the major exception of p300-CD-scFV), they do not assess off-target deposition of histone modifications. ChIP-seq or CUT&RUN on cells expressing each editor would be of value for more broadly exploring such off target effects. The signal observed in the negative control of Fig. 1I suggests off-targets may be a concern.

R2.3. We tested for OFF-targeting at the functional level by transcriptome profiling and at the direct level by quantitative CUT&RUN of non-targeted loci and target-adjacent regions, finding that for most

effectors there are negligible OFF-target effects. To address this further, we have now examined the genome-wide impact of H3K4me3 locus-specific programming. This was chosen since the Prdm9-CD^{scFV} effector is the most active, and the H3K4me3 antibody amongst the most sensitive, thereby enabling examination of even moderate OFF-target effects. Using spike-in CUT&RUN-seq we only detected six sites of H3K4me3 upon induction of the epigenetic editing system, except the targeted locus (*Hbby*) (*see new Extended data Fig 2A-B, and below, left*). Moreover, by introducing a pool of gRNAs into cells, we observed that target gene responses only occurred in cells that received the cognate gRNA, suggesting both

minimal OFF-targeting and specific ON-targeting responses (*see R4, reviewer #2*). Since the system is modular, this suggests that the inherent core design does not result in major OFF-target effects. Nevertheless, as pointed out by the reviewer, the Setd2-CD^{scFV} and p300-CD^{scFV} effectors did have detectable OFF-target activity. Importantly however, because we designed a reductionist system that specifically focusses on the *cis* response to a specific mark at the specific ON-target locus, with myriad controls, even low-level indirect effects are unlikely to impact our conclusions.

Testing ON- and OFF-target H3K4me3 deposition following Prdm9-CD^{scFV} targeting to the *Hbby* locus.

Left. Scatter plot showing de novo H3K4me3 peaks genome-wide. 6 off-target peaks (in red) are detected in addition to the ON-target *Hbby* peak (in green). Right. Genome view of the *Hbby* locus, showing a clear *de novo* H3K4me3 peak around the gRNA binding site, following Prdm9-CD^{scFV} targeting, relative to control GFP^{scFV}.

C2.4: Multiplexing potential. A major advantage of the CRISPR-dCas9 system is the ability to multiplex using sgRNAs against different targets in the same cell. The authors could explore this by targeting more than one gene at the same time and assessing the degree to which the intended histone mark is robustly established at multiple loci.

R2.4: Thank you for this comment, and we agree that multiplexed epigenetic editing across loci is both important to confirm (technically) and potentially of great utility. To test the potential for this in our toolkit we have now programmed three endogenous genes with H3K4me3 simultaneously, with independent pools of gRNA. Based on singleplex editing, two genes were expected to respond to multiplexed H3K4me3 (*Fgf5* and *Wnt8a*), with the third expected to be recalcitrant. Multiplexed targeting robustly led to upregulation of expected target genes (see right figure). This demonstrates proof of principle that indeed, epigenetic editing is feasible and efficacious at multiple loci simultaneously. Moreover, as shown in the manuscript multiplexed editing of multiple chromatin marks at single-locus is also possible, potentially opening up the possibility of compound epigenetic editing of several marks at multiple target loci.

Multiplexed epigenetic editing is enabled by the presented dCas9-effector system.

Bar plots showing expression of the indicated genes in ESC prior (-DOX) and following (+DOX) programmed deposition of H3K4me3 on gene promoter via pools (gRNA mix1 and gRNA mix2) of three different gRNAs (one for each targeted gene). *Fgf5* and *Wnt8a* are concomitantly upregulated, indicating successful multiplexed epigenetic editing.

C2.5: Targeting window. It would be helpful to provide a better idea of how flexible the targeting window of each editor is, and whether different target sites across a given promoter result in different patterns of histone modification deposition.

R2.5: To investigate the influence of gRNA targeting window we have performed *new* assays that tile across our knock-in reporter with gRNA(s) that span ~3kb around the TSS. We used CUT&RUN and found that all gRNA were sufficient to deposit significant levels of H3K27me3 (*see* below and Extended data Fig X). Notably, the programmed modifications spanned the entire promoter region (>2kb) irrespective of where the gRNA was targeted, albeit with differential peaks of enrichment according to the precise location. This is consistent with our system programming physiologically relevant ‘domains’ of chromatin, likely

because of compound recruitment of 5x effectors and a flexible dCas9 tail which enables greater reach. In contrast direct dCas9 fusion systems typically generate limited modification peaks corresponding to just a few nucleosomes, likely owing to more limited catalytic flexibility at target sites. Future work should investigate the rules of targeting window and effects

across endogenous loci.

Programmed epigenetic marks spread across large DNA domains irrespectively of the gRNA binding site.

Schematic and bar plots showing H3K27me3 enrichment on the ON reporter promoter region after EZH2^{scFV} recruitment via the specific indicated gRNA, relative to GFP^{scFV} control. The programmed modification spreads across a ~3kb region regardless of which gRNA is being used for targeting.

C2.6: Generalizability. It is difficult to get a sense of how effective the editors are across different gene targets since most of the experiments are carried out on the Hbby gene or their engineered EF1a reporter. While assessing the generalizability of all the editors is likely outside the scope of this manuscript, categorizing the chromatin marks into three groups based on their reporter read-out alone may be misleading since the effects could vary widely across different targets. The grouping also seems arbitrary at times: based on the data in Fig. 2C-K, DNA methylation is indistinguishable from the modifications in the subtle/partially repressing group.

R2.6: We agree fully with the comment. Indeed, a central theme of our conclusions is that the impact of chromatin modifications is likely highly-context dependent, and therefore we anticipate variable effect across loci. Detailing this comprehensively is, as the reviewer acknowledges, beyond the scope of this study. We are however able to establish quantitative and single-cell resolution data on what each chromatin modification has the *potential* to do, with causality. Based on this, and in order to provide structure, we loosely grouped modifications based on response patterns. With respect to the specific point on DNA methylation, we agree that this could functionally be considered with the subtle/partially-penetrant group (#3). The decision to place it within group #1 was based on two non-arbitrary parameters. (i) Whilst the magnitude of DNA methylation effect was relatively small at this locus (1.9-fold), it was still greater than group #3 marks (typically <1.5 fold). (ii) Secondly, and more distinguishing, the penetrance of DNA methylation-mediated effects was high, with most cells in the population exhibiting some response (*see Extended Data Fig 3A*). This is in contrast to group #3 responses which were typically only evident in a minor fraction of cells. As noted, the responses and groupings are likely locus and context-specific, and we highlight this in the discussion.

C2.7: Open access. The plasmids encoding each of the editors should be deposited to Addgene (or the equivalent) upon publication.

R2.7: We have already shared the toolkit across continents and intend to make it fully available.

C3: While the authors present compelling evidence that H3K4me3 deposition can activate gene expression, it is still unclear whether this has any functional significance. Their EpiLC differentiation experiment attempted to get at this but unfortunately only the most lowly expressed genes failed to activate in the absence of H3K4me3. Do the authors believe these low expression levels are biologically meaningful? Ideally they would show that H3K4me3-mediated activation of these genes is required for differentiation (or some other function). If not, the authors should acknowledge that their results are limited to lowly expressed genes and do not necessarily point to a functional role.

R3: We investigated the cellular functional significance of H3K4me3 deposition by generating homozygous knock-in point-mutant ESC for *Mll2*, which specifically abrogated its catalytic activity. MLL2

is one of several active H3K4me3 methylases, and preferentially targets lowly expressed bivalent genes. Other H3K4me3-marked genes are often modified redundantly by several methylases (i.e. no loss of H3K4me3), making their study challenging. Thus, we are primarily only able to investigate expression changes among more lowly expressed genes in *Mll2^{CM/CM}*. Indeed, since our question is whether switching genes OFF->ON requires H3K4me3, studying initially lowly expressed /OFF genes is necessary.

Nevertheless, because of concerns around the subtlety of *Mll2^{CM/CM}* effects, we repeated all RNA-seq and performed CUT&RUN-seq for H3K4me3, H3K27me3, and H3K27ac, using independent *Mll2^{CM/CM}* ESC lines (freshly-derived, low passage) (*see new Fig 3A-B and Extended data Fig 6*). The new data aligns extremely closely with previous data. For example, we find absence of H3K4me3 impacts timely activation of a subset of genes during cell state transition, implying a functional role. Indeed, in the new data we find 458 genes exhibit a significant failure to activate in EpiLC despite normal differentiation. Loss of H3K4me3 also resulted nearly exclusively in downregulation in ESC, as expected of a functional impact for the mark (*see new Fig 3B*). This likely translates into an observable phenotype in an *in vivo* mouse setting (*in preparation*), pointing towards an important functional role. Notably, these analyses are (by necessity) based on catalytic disruption of only *one* methylase, with many other methylases contributing further to overall H3K4me3 across loci. We therefore greatly underestimate the functional impact of H3K4me3 disruption at the cellular level.

C4: The fact that the recruitment of the YY1 motif can block both activating and repressing effects is intriguing, especially given the literature showing that YY1 can have both activating and repressing effects on gene expression. Could the authors elaborate on this further? Are these unintuitive effects of the YY1 motif generalizable to endogenous genes? Is it fair to think of YY1 as a “transcriptional buffer”?

R4: As the reviewer notes, we observed some intriguing *cis* genetic x epigenetic interactions including apparent YY1 insulation of chromatin modification impact. We agree with speculation that such YY1 sites may therefore act as a transcriptional ‘buffer’. Indeed, in general, we found motifs that promote chromatin structural/conformation changes (YY1, CTCF, G4) had far greater impact than specific TF motifs. We have performed a genome-wide analysis of genes with YY1 motifs. However, such analysis are confounded by myriad interacting and confounding influences that make any interpretation challenging. Therefore we believe to understand such specific interplays, reductionist and/or precision perturbation strategies such as ours are the optimal strategy, and should be pursued in future.

Minor points

C5: The manuscript contains a lot of data and I understand which makes it difficult to condense all of the experimental design details. However, more justification for why and how certain decisions were made would be helpful. More details on the following would be appreciated.

R5: Please see below, for point by point responses:

C6: How were the nine histone and DNA modifications selected?

R6: As explained fully in **R2** (reviewer #2, page 5), we tested and optimised in excess of 30 effectors covering >20 modifications. The nine modifications that were deployed for this study passed QC for ON-target activity, with minimised OFF-targeting, and were also of the most biological interest. In contrast many of the effectors tested had negligible activity.

C7: Do the five copies of GCN4 upon one copy? This is not always the case (e.g. steric hindrance).

R7: The system is designed to recruit 5x copies of an effector. Our data indicates this greatly maximises the magnitude of deposited histone modification relative to a (single) direct fusion. Moreover, because recruitment is non-covalent and linked with a flexible tail, the system can also deposit modifications over physiological domains as shown.

C8: Why were the full-length enzymes used for Ezh2 and Kmt5c while most editors were distilled down to catalytic domains?

R8: Our objective was to deploy only catalytic domains to specifically isolate the effect of targeted chromatin modifications, by avoiding co-recruitment of non-catalytic regulatory complexes associated with full-length proteins. In the case of *Ezh2*, despite optimisation attempts we were unable to establish a minimal catalytic domain with ON-target activity, likely because allosteric activation from the N-terminal (non-catalytic) region of the protein is necessary (Brooun et al., 2016). Because of the general interest in H3K27me3, and because this is the only enzyme for that mark, we elected to use (active) full-length effector in this case. Note that because we also generated a catalytic point-mutant *Ezh2*^{scFV} effector, we could still draw causal conclusions about whether any observed effects were due to catalytic- (H3K27me3) or non-catalytic- activity. In the case of *Kmt5C*, the catalytic SET domain necessary for activity comprises almost the entire protein. Moreover, a truncated version of the protein (see table in R2, page 5) did not display any catalytic activity. We therefore elected to use full-length to maximise activity, along with a catalytic-point mutant control.

C9: How were positive and negative controls selected in Fig. 1D-I? Strongest and weakest peaks possible, respectively? Based on which datasets?

R9: Positive control loci were selected based on previous literature and our available datasets indicating they are significantly modified and above the top 90 percentile of promoters for the given chromatin mark. Negative control genomic regions were selected as carrying undetectable chromatin modifications.

C10: How was *Hbby* selected for the initial targeting experiments and why were different gene targets used for DNA methylation?

R10: *Hbby* was selected for three defined reasons. (i) Firstly, the *Hbby* locus is well characterised and transcriptionally inert in ESC. This makes it a relatively neutral locus to test baseline epigenetic editing without confounding effects of TF or PIC binding. (ii) Secondly, our epigenomics datasets indicate *Hbby* carries little, if any, pre-existing chromatin marks in naïve ESC. It is therefore a target for assessing *de novo* programming of every histone modification in our toolkit. (iii) Finally, *Hbby* has previously been used successfully as a neutral location to test genomic engineering (Lienert et al., 2011).

Whilst ideal to test deposition of histone modifications, *Hbby* is however highly DNA methylated, like the majority of the genome. It cannot therefore be used to test for *de novo* programming of DNAm. We thus selected alternative (unmethylated) regions to test DNAm editing, which generally correspond to promoter regions, and which consistently showed a highly significant DNAm acquisition.

C11: Most experiments use a 7-day time point. How was this time point chosen?

R11: We tested the dynamics of epigenetic editing outcomes at initial steady-state (t_0), at an early timepoint (day 2), and extended timepoint (d7) across marks and context. In general, the responses at early timepoints (d2) were highly comparable to latter response, in both penetrance and magnitude. There were however some exceptions. For example, PRC1 (H2AK119ub)-mediated repression exhibited slower dynamics than H3K9me2/3 (shown right). This can be seen in the heatmaps shown in Fig 4C and Fig 6A. To expand upon the reviewers comment, we have now generated an entirely *new* Extended Data Figure 4 that demonstrates the temporal dynamics of causal responses to each epigenetic modification.

Temporal dynamics of transcriptional changes following epigenetic editing.

Dot plots showing expression of the ON reporter in single cells after recruitment of either G9a-CD^{scFV} or Ring1b-CD^{scFV} at the indicated timepoints, relative to control GFP^{scFV}. H3K9me2 elicits its repressive activity faster than H2AK119ub.

C11: How were the permissive (chr9) and non-permissive (chr13) sites selected?

R11: Recent studies from Bas Van Steensel using TRIP, have comprehensively mapped position effects on a fixed reporter sequence within the ESC genome (Akhtar et al., 2013). We made use of this data to select an optimal ESC locus that is anticipated to (initially) be non-permissive for transcription (Chr13), whilst also being located away from any key genes or genomic features that may be impacted by genetic engineering at the locus. For the permissive location we made use of the well-established TIGRE locus, that is characterised as a neutral genomic landing site that can support transcription. We have updated the methods to make this selection process clearer.

C12: How were the 8 genes in the H3K4me3 targeting rescue experiments picked?

R12: We selected these targets based on the following criteria. (i) They must be H3K4me3 marked in WT cells and lose promoter H3K4me3 in *Mll2*^{CM/CM}. (ii) As a consequence of this, they must exhibit a significant reduction of transcription, to enable to test rescue with targeted H3K4me3. Of note >99% differentially expressed genes that lose H3K4me3 due to homozygous *Mll2* catalytic mutation are indeed downregulated. (iii) Finally, amongst this group, we selected eight candidates to test rescue essentially randomly, albeit with a tendency to use higher effect-size downregulated genes to provide more scope for unambiguous conclusions on the rescue. However, no specific criteria was applied and the genes used exhibit a range of genomic features, effect-size, and expression level. Moreover, of note a high degree of rescue was noted in all cases upon promoter-targeted H3K4me3 programming in *Mll2*^{CM/CM} ESC. Finally, it is important to emphasise that these are not just the loci that worked. These data correspond to all loci that we tested.

C13: What about the 8 endogenously silent genes? The paper claims the latter were random – how were they randomized?

R13: The only specific criteria that was applied to select these genesets was that they must be extremely lowly - or undetectably - expressed in WT ESC (RPKM<1). This is essential, since we are specifically testing the ability to promote transcription from genes that are not expressed within a given cell type. Data from all tested genes is included. We found approximately 35% of these genes (3/8) exhibited some degree of transcriptional response to targeted H3K4me3, with around 65% being recalcitrant to this modification. We have modified the manuscript to remove mention of 'random' since whilst no criteria (other than obligatory low expression) were applied, we did not formally randomise the process.

C14: Do the DNA motifs investigated in the study correspond to the consensus mouse motifs or consensus human motifs? It would be interesting to test if equivalent human motifs have the same effect (i.e. is the effect of a given TF motif conserved even if the sequence of the motif itself is not?).

R14: The motifs used correspond to the consensus sequences in mouse accessed from JASPAR. The idea of rigorously testing cross-species motif interactions is excellent, and could form the basis of a interesting study going forward. In our case, the consensus mouse motif typically corresponds to the human motif almost exactly. For example core MYC motif, CACGTG, mouse; CACGTG, human. We have therefore (inadvertently) tested both human and mouse motifs within our system.

C15: Is their Dox-inducible system leaky?

R15: The reviewer is correct in that DOX-inducible systems *can* show a degree of leakiness under sub-optimal designs. To avoid this we undertook several steps and validations. (i) First, we deployed the most recent TET3G-based inducible promoter. This is specifically designed to lack any cryptic TF sites and to be maximally repressed, thus reducing leaky activity relative to other inducible promoters. Second (ii), we introduced dCas9 into the genome and selected clones with minimal insertions, thus reducing the chance of leaky expression that can occur upon multiple genomic integrations. Thirdly (iii), we tested for leaky expression comprehensively. Because all our effectors are tagged with sfGFP, we can examine

The dCas9-effector toolkit is characterised by no leakiness.

Representative flow cytometry dot plots showing DOX-dependent induction of the epigenetic editing system. The enhanced gRNA is constitutively expressed and marked by tagBFP (y-axis). dCas9^{GCN4} and each CD^{scfV} effector are activated by DOX, leading to nuclear GFP signal. In -DOX conditions very few cells express detectable GFP across all effectors, indicating that our epigenetic editing system is not subjected to leakiness.

background levels at single-cell resolution using quantitative flow cytometry. As shown in the figure (right) and Extended Data 1B, we were unable to detect ‘leaky’ (GFP) expression within the population in minus (-) DOX conditions, relative to untransfected controls that completely lack the system (used to set gates). In contrast, +DOX conditions led to strong activation of epigenetic editing machinery tagged with GFP. Finally (iv), to further support absence of leaky expression, we observed no functional effect on target genes in -DOX conditions relative to untransfected cells, whereas addition of DOX typically induced a clear response. These data imply that if there is ‘leaky’ expression is below threshold detection levels, both directly and functionally.

C16: The authors mention that using 20-fold lower Dox was used to mitigate the off-target effects of p300-CD-scFV but do not show the resulting improved data following Dox titration.

R16: We were not precise in clarifying this and apologise. Higher levels of p300-CD^{scFV} induction (100ng/ul DOX) led to toxicity in ESC, albeit not in other tested cell types, and not for any other effectors. As shown in R2.2 (above, page 19) and now in a *new* Extended Data 2D, we found lower DOX concentrations (5ng/ul) are sufficient to enable robust cellular proliferation in ESC, presumably owing to reduced toxicity/OFF-targeting, and the lower concentration is thus used throughout the manuscript. However, some OFF-target effect is still apparent specifically with this effector. Because our analysis is specifically focussed on isolating the ON-target effects, because H3K27ac is sparsely investigated within our manuscript, and because we make it clear that p300-CD^{scFV} has some indirect effects, we believe it is important to retain these caveated results.

Minor line-referenced comments:

C17: Line 97-99: ‘peak... centered on the gRNA binding site’ – seems like the peak is actually upstream of sgRNA site for most marks?

R17: In our data examining the genomic distribution of each programmed chromatin mark, we used adjacent primersets to the gRNA target site, which sit approx 400-500bp flanking either side (with other sets then radiating further). Thus, whilst it does appear there is a slight preferential upstream enrichment in

some cases, these primers are actually ~2 nucleosomes away from the precise target site, which is likely even more modified. More generally, it is clear that a significant peak of modification occurs *around* the gRNA target site and then tapers off after around 2-3kb either side. To address the reviewers point we have therefore altered the text to “centered *around* the gRNA binding site” rather than “on...”.

C18: Line 112-113: cannot claim lack of off-target histone marks based on two loci.

R18: We have now generated additional data that suggests *de novo* OFF-target peaks arise rarely genome-wide upon H3K4me3 deposition. However, we take on board the reviewers comment about the generality of our claim and have softened our conclusions to “*OFF-target and/or indirect effects are minimised.*”

C19: Line 130: what is the CpG content of the chosen sequence? Could low CpG density explain underwhelming DNA methylation effects?

R19: This is an important point to interpret DNA methylation-mediated effects. Indeed, unlike for histone modifications, where the density of substrate is relatively similar across loci, the density of CpG residues varies far more greatly. This means the effect of DNA methylation is especially sensitive to underlying DNA sequence (i.e. CpG content). The reporter was designed to be relatively genomically neutral, i.e. no TE, minimal TF binding sites, and average GC. In effect this means the locus is CpG depleted, since CG dinucleotides are greatly diminished within the ‘average’ genome, with the exception of CpG islands. This low CpG density likely does explain the underwhelming effect. It is noteworthy however that the core region of the promoter (EF1 minimal) does have a higher CpG density that recapitulates aspects of a weak CpG island over ~180bp. Thus, whilst we agree with the reviewer, one cannot unambiguously draw the conclusion that the effect is attenuated due to CpG-depletion. Future work should systematically investigate the linear and threshold effects of CpG density in respect to the functional effect of programmed DNA methylation.

C20: Line 221: surprising that 53 genes are upregulated upon loss of H3K4me3 – which genes are these?

R20: We anticipate that these changes represent indirect/secondary effects. In this scenario, these genes lose promoter H3K4me3, since they are targets of *Mll2*. At the same time a global change in gene expression and TF networks shift the cells such that a number of targets become aberrantly activated (independently of H3K4me3 – which may or may not play a role under normal circumstances at these loci). A supplementary dataframe file including all gene expression changes is now available for perusal.

C21: Line 253-156: did the authors try successive targeting? i.e. target, sort off cells, target again, assess transcriptional activation.

R21: This is an interesting idea. We sorted cells based on transcriptional response to various modifications, and then tracked the degree of modification each population received (*see Extended Data Fig 7E*). Moreover, we sorted cells and then tracked the memory of transcriptional response over time (*see Fig 7A-B*). However, we did not attempt the suggestion to then re-target the same or a different modification. This is something worth pursuing in the future.

C22: Line 259: what is meant by ‘productive’? Should comment that productive transcription is not necessarily functional.

R22: We have removed the word ‘productive’. It was erroneously used to emphasise active transcription.

C23: Line 364: the authors make the important point that the function of a chromatin modification is “highly context dependent”. An important caveat to add is that most of their experiments are done on an exogenous EF1a reporter, albeit integrated at two genomic regions.

R23: The message that the functional impact of any given modification is exquisitely dependent on myriad contextual parameters is a central theme of our study. These functional ‘modifiers’ include TF motifs within a broader promoter, the pre-existing chromatin state, and/or the genomic location. Other contexts that are

likely important include the cell type, the starting expression level, and the type of core promoter (e.g +/- TATA-box, precise sequence etc) as noted by the reviewer. Whilst we cannot test all of these, we have now modified the text to re-emphasise the various contexts that likely feed into functional outcomes.

C24: Lines 406-411: it is surprising that no epigenetic memory was observed after DNA methylation targeting given previous reports. Do the authors think this is primarily a result of working in ESCs or more likely to be a reporter-specific effect?

R24: We strongly suspect this is a phenomenon specifically associated with naïve pluripotent cells. In a previous study we programmed large domains of heterochromatin (H3K9me3, DNA methylation, and H4K20me3) using KRAB, specifically to examine the capacity for ‘epigenetic memory’. We found ESC rapidly erased even major heterochromatin domains when ectopically programmed, but in contrast, differentiated cells could propagate at least some memory, both *in vitro* and importantly *in vivo* (Carlini *et al.*, 2022). Whilst this study is distinct from the present manuscript, in that it does not assess the role of chromatin marks, but rather a combination of several modifications and direct transcriptional repression, it reinforces the idea that pluripotent cells have specific systems in place to antagonise epigenetic memory. Indeed, we identified *Dppa2*, as one of these factors (Carlini *et al.*, 2022, *EMBO*). Amongst other changes in the discussion to highlight this more clearly in the manuscript, we include a line in the results to: “*Such lack of ‘epigenetic memory’ may reflect the cell-type specific context of ESC*”

Comments on figures:

In general the figures are well constructed and aesthetically pleasing. Below are some suggestions for each that would improve clarity.

C25: Figure 1:C: Size of region assessed? Same as in J?

R25: Fig 1C captures the average level of DNA methylation across the locus whereas, Fig1J shows the effect at individual CpGs across the locus. Moreover these are independent experiments.

C25: G and H: The split y-axes are misleading – edited mark does not reach levels observed at positive controls.

R25: The split axes are necessary to capture the scale of the key point, that is, that epigenetic editing is present relative to control levels. We acknowledge in the manuscript text that these two marks do not reach the level of the (strong) positive controls. Of note, the positive controls used for these marks are exceptionally high; amongst the highest in the genome.

C26: D-I: Missing gene names for positive and negative controls. How were positive and negative controls selected?

R26: Where no gene name is indicated, this reflects that the locus is an intergenic region or not associated with a specific gene, selected based on inspection of genome tracks to ensure absence (negative) or presence (positive) of the mark in question. In the case of positive controls (e.g. for H3K9me2), these are exceptionally highly modified regions. The primers and locations are provided in the supplementary tables.

C27: K: Surprising that there are a number of downregulated genes following H3K27ac targeting – are these genes targets of upregulated genes?

R27: As discussed above, expression of p300-CD^{scFV}, but not other effectors, appears to effect cell state, thus leading to widespread indirect effects, that likely account for downregulated genes.

C28: J: where are the sgRNA binding sites for these genes?

R28: gRNA binding sites for DNA methylation programming are located adjacent (upstream), but not overlapping, the region assayed by pyrosequencing. This emphasises the ability to programme DNA methylation over larger distances than fusion based approaches.

Figure 2:

B: Could a population of cells without the reporter be included in this graph and other flow cytometry data in the paper to get a sense of what a completely ‘off’ state looks like? It would be nice to directly compare ‘off’ to edited silenced cells to determine whether some of the editors are capable of completely turning off gene expression.

R29: We agree that a better understanding of how a completely ‘OFF’ looks like is important. We have added a *new* Fig 2B to demonstrate the levels of silencing across each landing locus relative to an absolute negative.

Expression of the REF reporter at the permissive and non-permissive locus.

Dot plots from quantitative flow cytometry showing single-cell activity of the Reference reporter when integrated into either the permissive or non-permissive locus, relative to expression in cells carrying no reporter at all.

C30: C-K: It took a while to figure out which reporter is being assessed for each graph. Could the label be moved from the y axis to somewhere more obvious? Perhaps to the left of each row of graphs?

R30: Whilst we tried different configurations (such as by row), this led to other issues, for example did it correspond to all plots in that row? Therefore indicating the genomic context of the reporter on *each plot* on the Y-axis appears most accurate. Moreover, this information can be determined by observing the starting expression level (GFP^{scFV}) on each plot. Finally, the effect of every modification on the reporter at both integration sites is shown in full in Extended Data Fig 3.

C31: E: which region of the contextual DNA sequence + promoter was assessed here?

R31: The region corresponds to a 71bp long sequence located around the initial part of the reporter promoter. Specifically, we assayed 27bp upstream of the promoter and 44bp into the 5’ end of the EF1a

promoter. As explained above (R19), this region has a CpG density that recapitulates aspects of a weak CpG island.

Figure 3:

C32: C-E: There is a very significant difference in activation of *Cldn16* between *Prdm9* and *Set1a* targeting. Is this related to the degree of H3K4me3 deposited? Could H3K9me3 enrichment for this particular gene be added to panel D? What are biological replicates in this context?

R32: As detailed in R6 (reviewer #2, page 11) the difference between effectors is, we believe, related to level of H3K4me3 deposited. Specifically, we found *Prdm9*-CD^{scFV} to be highly effective at deposition (typically >10-fold enrichment), *Setd1a*-CD^{scFV} to be moderately effective (~5-fold), and *Mll2*-CD^{scFV} to only deposit very low levels (<2-fold). The reviewer queries why the transcription responses were quantitatively different for *Prdm9*-CD^{scFV} and *Setd1a*-CD^{scFV} (e.g. *Cldn16* = >100-fold and 6-fold respectively). We demonstrate that the level of transcriptional response is correlated with the quantitative level of H3K4me3 deposition, revealing a dose-dependent effect (see Fig 3E & Extended data Fig 7D). The replicates for Fig 3D are independent transfections and inductions of the epigenetic editing system into WT cells.

H3K4me3 triggers transcriptional changes in a dose-dependent manner.

Dot plots showing expression at the OFF reporter after programming H3K4me3 with distinct effectors: *Prdm9*^{scFV} (left), *Set1a*^{scFV} (center) and *Mll2*^{scFV} (right). The level of transcriptional upregulation correlates with the amount of epigenetic mark deposited by each effector, as indicated on top of each plot.

C33: C: Data for other 4 targeted genes missing?

C and E: Genes should appear in the same order and aligned one on top of another for easy comparison.

G: missing 'C' in mCherry

R33: The figure has been remodelled to include new data. For example, we repeated and extended analysis of *Mll2^{CM/CM}* and have added the new analysis (see *new Fig 3B*), which demonstrates a clearer effect of H3K4me3 loss towards gene downregulation. We further added new analysis to more clearly highlight the impact of *Mll2^{CM/CM}* during ESC->EpiLC (see *new Fig 3I-J*). Note all of this new analysis comes from entirely reproduced 'omics' datasets from the original manuscript, thus also validating previous results independently.

Figure 4:

C34: B: Colors are hard to distinguish

G-I: Beautiful results

R34: We have modified the colours to be the same. Indeed, there is actually no reason for a difference given each plot is labelled as 'Reference' or '+CTCF' so we fully agree with the reviewer.

Supplementary

C35: Supp Fig. 1:

A: Add number of NLSs and construct diagrams for each editor

B: Show leakiness of Dox system

D: DNA methylation data missing

E: Gene names for controls?

E: Split y-axis misleading

Supp Fig. 3:

A and B: Axis labels missing

Supp Fig. 5:

B: Will a list of the up- and downregulated genes be provided in a supplementary table?

Supp Fig. 6:

B: Equalize axes of two graphs

Supp Fig. 7:

C and D: What are the sample sizes for these graphs?

Supp Fig. 8:

C: Confusing layout – in some cases two different clones are compared and in others different motifs.

Could a separate panel be created for clonal comparisons?

B and C: Only select examples are shown. As a supplemental figure there should be enough space to include all the data.

Supp Fig. 9:

Statistics?

R35: We have been through and made the suggested modifications, where appropriate and not responded to previously. Lists of DEGs are now provided as supplementary tables.

Akhtar, W., de Jong, J., Pindyurin, A.V., Pagie, L., Meuleman, W., de Ridder, J., Berns, A., Wessels, L.F., van Lohuizen, M., and van Steensel, B. (2013). Chromatin position effects assayed by thousands of reporters integrated in parallel. *Cell* 154, 914-927. 10.1016/j.cell.2013.07.018.

Benayoun, B.A., Pollina, E.A., Ucar, D., Mahmoudi, S., Karra, K., Wong, E.D., Devarajan, K., Daugherty, A.C., Kundaje, A.B., Mancini, E., et al. (2014). H3K4me3 breadth is linked to cell identity and transcriptional consistency. *Cell* 158, 673-688. 10.1016/j.cell.2014.06.027.

Brooun, A., Gajiwala, K.S., Deng, Y.L., Liu, W., Bolanos, B., Bingham, P., He, Y.A., Diehl, W., Grable, N., Kung, P.P., et al. (2016). Polycomb repressive complex 2 structure with inhibitor reveals a mechanism of activation and drug resistance. *Nat Commun* 7, 11384. 10.1038/ncomms11384.

Carlini, V., Policarpi, C., and Hackett, J.A. (2022). Epigenetic inheritance is gated by naive pluripotency and Dppa2. *EMBO J*, e108677. 10.15252/embj.2021108677.

Chen, K., Chen, Z., Wu, D., Zhang, L., Lin, X., Su, J., Rodriguez, B., Xi, Y., Xia, Z., Chen, X., et al. (2015). Broad H3K4me3 is associated with increased transcription elongation and enhancer activity at tumor-suppressor genes. *Nat Genet* 47, 1149-1157. 10.1038/ng.3385.

Douillet, D., Sze, C.C., Ryan, C., Piunti, A., Shah, A.P., Ugarenko, M., Marshall, S.A., Rendleman, E.J., Zha, D., Helmin, K.A., et al. (2020). Uncoupling histone H3K4 trimethylation from developmental gene expression via an equilibrium of COMPASS, Polycomb and DNA methylation. *Nat Genet* 52, 615-625. 10.1038/s41588-020-0618-1.

Lienert, F., Wirbelauer, C., Som, I., Dean, A., Mohn, F., and Schubeler, D. (2011). Identification of genetic elements that autonomously determine DNA methylation states. *Nat Genet* 43, 1091-1097. ng.946 [pii] 10.1038/ng.946.

Liu, X., Wang, C., Liu, W., Li, J., Li, C., Kou, X., Chen, J., Zhao, Y., Gao, H., Wang, H., et al. (2016). Distinct features of H3K4me3 and H3K27me3 chromatin domains in pre-implantation embryos. *Nature* 537, 558-562. 10.1038/nature19362.

- Vermeulen, M., Mulder, K.W., Denissov, S., Pijnappel, W.W., van Schaik, F.M., Varier, R.A., Baltissen, M.P., Stunnenberg, H.G., Mann, M., and Timmers, H.T. (2007). Selective anchoring of TFIIID to nucleosomes by trimethylation of histone H3 lysine 4. *Cell* *131*, 58-69. 10.1016/j.cell.2007.08.016.
- Wang, H., Fan, Z., Shliha, P.V., Miele, M., Hendrickson, R.C., Jiang, X., and Helin, K. (2023). H3K4me3 regulates RNA polymerase II promoter-proximal pause-release. *Nature* *615*, 339-348. 10.1038/s41586-023-05780-8.
- Weinert, B.T., Narita, T., Satpathy, S., Srinivasan, B., Hansen, B.K., Scholz, C., Hamilton, W.B., Zucconi, B.E., Wang, W.W., Liu, W.R., et al. (2018). Time-Resolved Analysis Reveals Rapid Dynamics and Broad Scope of the CBP/p300 Acetylome. *Cell* *174*, 231-244 e212. 10.1016/j.cell.2018.04.033.
- Zhao, W., Tong, H., Huang, Y., Yan, Y., Teng, H., Xia, Y., Jiang, Q., and Qin, J. (2017). Essential Role for Polycomb Group Protein Pcgf6 in Embryonic Stem Cell Maintenance and a Noncanonical Polycomb Repressive Complex 1 (PRC1) Integrity. *J Biol Chem* *292*, 2773-2784. 10.1074/jbc.M116.763961.

Decision Letter, first revision:

17th Oct 2023

Dear Dr Hackett,

First, please accept my sincere apologies for the delay in returning this decision to you. I am honestly mortified that I kept you waiting for so long and am very grateful for your patience.

Thank you for submitting your revised manuscript "Systematic Epigenome Editing Captures the Context-dependent Instructive Function of Chromatin Modifications" (NG-A60906R). We tried without success to contact Reviewer #2 and in the end decided to move forward without their feedback. I'm delighted to say that we'll be happy in principle to publish it in *Nature Genetics*, pending minor revisions to satisfy the referees' final requests and to comply with our editorial and formatting guidelines.

Sincerely,

Safia Danovi
Editor
Nature Genetics

Reviewer #1 (Remarks to the Author):

The revision is improving several aspects of the paper, which remains in my mind a strong candidate for nature genetics.

The important contribution of this paper is in bringing together flexible epigenetic editing tools (whose robustness and functionality remain quite difficult to understand), with quantitative readout through a reductionist approach. This is really impactful – especially given the interesting and positive results for some of the perturbations.

After many years of epigenomic profiling and knockouts, we still know very little about the mechanisms and mode-of-action of the various complexes when working in actual logic in-vivo. This is mostly because in epigenomics everything is part of a mega feedback loop. Observing a regulatory effect for a mutation in an epigenetic factor genome wide is always open to questions of direct vs. indirect mechanism. And when many of the effects are of very small magnitude, but occur over hundred or thousands of loci, it is almost impossible to make progress using more ChIP-seq, C&R, C&T and even single cell versions of these approaches.

What the authors are doing here is to use a quantitative readout and compare distributions over single cells given perturbation, in a highly standardized and reductionist fashion. This will not explain how H3K4me3 or PRC1/PRC2 works in general – and we should not expect this work to resolve all questions in the field. In systems that are not ESCs, or in genomic context that are different from those tested, the complex epigenetic interplay may hold additional surprises. But it is, even given the need to continue and validate lines, perturbations, side effects, a big step in the right direction and a message to the epigenetic community regarding how real progress can and should be made.

Reviewer #3 (Remarks to the Author):

We thank the authors for their detailed responses to our questions. They have added important controls and have performed new experiments that further support their conclusions. Notable additions were a spike-in CUT&RUN experiment to quantify off-target H3K4me3 programming (showing only 6 de novo off-target peaks) and RNA- & CUT&RUN-seq experiments on independent freshly derived MII2CM/CM cell lines, which provided validation of the more subtle effects observed in

their first batch of experiments related to H3K4me3 function. Overall, they have addressed our major concerns and we are happy to recommend the manuscript for publication without further delay

Minor comment: For some of their responses to our concerns, it was unclear whether the answer was for the reviewers' benefit or whether changes were made to the manuscript text and/or figures. We understand the journal's need for space limitation but given that an important aspect of this paper is a resource to the community, we think it would be helpful to include them. For example, the results from their proof-of-principle multiplexing experiment and the experiment tiling guides across the promoter to assess targeting window. For the latter, it would be helpful to equalize the y-axis ranges to 0-20 to better highlight that guides 2 and 3 are superior, possibly suggesting that H3K4me3 targeting is optimal at or immediately upstream of the TSS.

Reviewer #4 (Remarks to the Author):

Remarks to the Author:

We thank the authors for their detailed responses to our questions. They have added important controls and have performed new experiments that further support their conclusions. Notable additions were a spike-in CUT&RUN experiment to quantify off-target H3K4me3 programming (showing only 6 de novo off-target peaks) and RNA- & CUT&RUN-seq experiments on independent freshly derived MII2CM/CM cell lines, which provided validation of the more subtle effects observed in their first batch of experiments related to H3K4me3 function. Overall, they have addressed our major concerns and we are happy to recommend the manuscript for publication without further delay

Minor comment: For some of their responses to our concerns, it was unclear whether the answer was for the reviewers' benefit or whether changes were made to the manuscript text and/or figures. We understand the journal's need for space limitation but given that an important aspect of this paper is a resource to the community, we think it would be helpful to include them. For example, the results from their proof-of-principle multiplexing experiment and the experiment tiling guides across the promoter to assess targeting window. For the latter, it would be helpful to equalize the y-axis ranges to 0-20 to better highlight that guides 2 and 3 are superior, possibly suggesting that H3K4me3 targeting is optimal at or immediately upstream of the TSS.

Author Rebuttal, first revision:

Policarpi et al

Response to reviewers (2nd round)

REVIEWERS #3,4:

C1: For some of their responses to our concerns, it was unclear whether the answer was for the reviewers' benefit or whether changes were made to the manuscript text and/or figures. We

understand the journal's need for space limitation but given that an important aspect of this paper is a resource to the community, we think it would be helpful to include them. For example, the results from their proof-of-principle multiplexing experiment and the experiment tiling guides across the promoter to assess targeting window. For the latter, it would be helpful to equalize the y-axis ranges to 0-20 to better highlight that guides 2 and 3 are superior, possibly suggesting that H3K4me3 targeting is optimal at or immediately upstream of the TSS.

R1: Whilst the vast majority of new data/figures provided to reviewers are included in the revised manuscript, the specific data mentioned is omitted for two key reasons. First, we are deeply limited by space having already reduced the number of extended data figures in line with journal policy, and cut the word count considerably. Second, since it was an indicative answer for a reviewer enquiry, we would prefer not to include it within the main body without further replications, across many marks. However, it does clearly provide answers to the specific reviewer question and will be fully available to the community in the co-published peer-review transcripts.

Final Decision Letter:

In reply please quote: NG-A60906R1 Hackett

5th Mar 2024

Dear Dr Hackett,

I am delighted to say that your manuscript "Systematic Epigenome Editing Captures the Context-dependent Instructive Function of Chromatin Modifications" has been accepted for publication in an upcoming issue of Nature Genetics.

Your paper will be published online after we receive your corrections and will appear in print in the next available issue. You can find out your date of online publication by contacting the Nature Press Office (press@nature.com) after sending your e-proof corrections.

Please note that *Nature Genetics* is a Transformative Journal (TJ). Authors may publish their research with us through the traditional subscription access route or make their paper immediately open access through payment of an article-processing charge (APC). Authors will not be required to make a final decision about access to their article until it has been accepted. Find out more about Transformative Journals

Authors may need to take specific actions to achieve compliance with funder and institutional open access mandates. If your research is supported by a funder that requires immediate open access (e.g. according to Plan S principles) then you should select the gold OA route, and we will direct you to the compliant route where possible. For authors selecting the subscription publication route, the journal's standard licensing terms will need to be accepted, including <https://www.nature.com/nature-portfolio/editorial-policies/self-archiving-and-license-to-publish>. Those licensing terms will supersede any other terms that the author or any third party may assert apply to any version of the manuscript.

To assist our authors in disseminating their research to the broader community, our SharedIt initiative provides you with a unique shareable link that will allow anyone (with or without a subscription) to

read the published article. Recipients of the link with a subscription will also be able to download and print the PDF.

If you have not already done so, we invite you to upload the step-by-step protocols used in this manuscript to the Protocols Exchange, part of our on-line web resource, natureprotocols.com. If you complete the upload by the time you receive your manuscript proofs, we can insert links in your article that lead directly to the protocol details. Your protocol will be made freely available upon publication of your paper. By participating in natureprotocols.com, you are enabling researchers to more readily reproduce or adapt the methodology you use. [Natureprotocols.com](http://natureprotocols.com) is fully searchable, providing your protocols and paper with increased utility and visibility. Please submit your protocol to <https://protocolexchange.researchsquare.com/>. After entering your nature.com username and password you will need to enter your manuscript number (NG-A60906R1). Further information can be found at <https://www.nature.com/nature-portfolio/editorial-policies/reporting-standards#protocols>

Sincerely,

Safia Danovi, PhD
Senior Editor, Nature Genetics
ORCID: 0009-0007-7822-5479